# Dorsal hippocampus to nucleus accumbens projections drive reinforcement via activation of accumbal dynorphin neurons

Khairunisa Mohamad Ibrahim [1,2,9], Nicolas Massaly[1,2,3,9], Hye-Jean Yoon [1,2], Rossana Sandoval [1,2], Allie J. Widman[1,2], Robert J. Heuermann [1,2,4], Sidney Williams[1,2], William Post[1,2], Sulan Pathiranage [1,2], Tania Lintz[1,2], Azra Zec[1,2], Ashley Park[1,2], Waylin Yu[5,6], Thomas L. Kash[5,6], Robert W. Gereau IV [1,2,7] & Jose A. Morón [1,2,7,8] ✉

The hippocampus is pivotal in integrating emotional processing, learning, memory, and reward-related behaviors. The dorsal hippocampus (dHPC) is particularly crucial for episodic, spatial, and associative memory, and has been shown to be necessary for context- and cue-associated reward behaviors. The nucleus accumbens (NAc), a central structure in the mesolimbic reward pathway, integrates the salience of aversive and rewarding stimuli. Despite extensive research on dHPC→NAc direct projections, their sufficiency in driving reinforcement and reward-related behavior remains to be determined. Our study establishes that activating excitatory neurons in the dHPC is sufficient to induce reinforcing behaviors through its direct projections to the dorso-medial subregion of the NAc shell (dmNAcSh). Notably, dynorphin-containing neurons specifically contribute to dHPC-driven reinforcing behavior, even though both dmNAcSh dynorphin- and enkephalin-containing neurons are activated with dHPC stimulation. Our findings unveil a pathway governing reinforcement, advancing our understanding of the hippocampal circuit's role in reward-seeking behaviors.

Despite years of groundbreaking research, the exact circuits and brain structures involved in governing reinforcement and goal-directed behaviors are still not fully uncovered. The hippocampus is a heterogenous brain structure, in which the dorsal subdivision (dHPC) is critically involved in spatial and contextual memory while its ventral subdivision (vHPC) integrates the emotional value of salient stimuli and guides goal-directed behavior[1–5]. Indeed, vHPC activity modulates both rewarding stimuli, drug relapse, and threat

avoidance behaviors[6–13], confirming its role as a key structure in emotional encoding[2,14,15]. On the other hand, the dHPC plays a crucial role in the integration, maintenance, and retrieval of contextual, spatial, and reward-associated memories[16–25]. We and others have demonstrated that dHPC activity is necessary to initiate goal-directed behavior to avoid stressors/threats or engage in reinforcement in response to cues/contexts previously associated with an aversive or rewarding stimulus, respectively[1,19,20,25–29]. These findings

[1]Department of Anesthesiology, Washington University Pain Center, St. Louis, MO 63110, USA. [2]Washington University in St. Louis, School of Medicine, St. Louis, MO 63110, USA. [3]Department of Anesthesiology and Perioperative Medicine, University of California, Los Angeles, CA 90095, USA. [4]Department of Neurology, Washington University Pain Center, St. Louis, MO 63110, USA. [5]Bowles Center for Alcohol Studies, University of North Carolina at Chapel Hill School of Medicine, Chapel Hill, NC 27599, USA. [6]Department of Pharmacology, University of North Carolina at Chapel Hill School of Medicine, Chapel Hill, NC 27599, USA. [7]Department of Neuroscience, Washington University in St. Louis, St. Louis, MO 63110, USA. [8]Department of Psychiatry, Washington University in St. Louis, St. Louis, MO 63110, USA. [9]These authors contributed equally: Khairunisa Mohamad Ibrahim, Nicolas Massaly. ✉e-mail: jmoron-concepcion@wustl.edu

highlight the critical role of the dHPC in reinforcing behavior and reward processing.

The nucleus accumbens (NAc), a central structure of the meso-limbic dopaminergic pathway, plays a critical role in integrating dopaminergic reinforcement signals from the ventral tegmental area (VTA)[30–32]. Excitatory calcium-calmodulin (CaM) dependent protein kinase II (CaMKII[+]) afferents to the NAc including inputs from the amygdala, the prefrontal cortex, and the vHPC are sufficient to drive reinforcement[33]. While the vHPC is sufficient to reinforce instrumental behavior, more recent findings have demonstrated that CaMKII+ afferent from the dHPC to the NAc Shell (NacSh) are necessary for the expression of contextual memories induced by salient stimuli[23]. In addition, dHPC electrical stimulation is sufficient to elicit reinforcement independently of associative learning[34,35]. However, the current literature still lacks clear evidence on the sufficiency of selective dHPC → NAc projection to drive reinforcement, a critical process that shapes reward learning and reward-seeking behavior to facilitate and guide goal-directed behaviors.

Considering the well-defined role of the NAcSh in processing reward and positive reinforcement[36–40], and particularly the dorsomedial subregion of the NAcSh (dmNAcSh)[41–43], this study used a combination of optogenetics, chemogenetics, in-vivo electrophysiology, and fiber photometry approaches in freely moving mice to investigate the sufficiency of dHPC[CaMKII+] neuronal inputs selectively to the dmNAcSh to drive reinforcement. We also assessed whether dHPC[CaMKII+]-induced reinforcing behaviors were driven via local glutamatergic signaling that acts on medium spiny neurons (MSNs) within the dmNAcSh. Finally, we assessed the neuronal properties of dmNAcSh MSNs that receive inputs from dHPC[CaMKII+] using ex vivo recordings in brain slices. Using an optogenetic self-stimulation instrumental procedure, we demonstrate that activation of the dHPC[CaMKII+]→dmNAcSh pathway is sufficient to drive instrumental reinforcement. Interestingly, both dynorphin- and enkephalin-containing medium-spiny neurons (MSNs) in the dmNAcSh display significant increase in calcium transient, a proxy for neuronal activity, in response to dHPC[CaMKII+] neurons stimulation. Our ex vivo recordings in accumbal slices determine that the dHPC[CaMKII+] neurons connect directly to enkephalin-containing MSNs throughout the dmNAcSh, while the dynorphin-containing MSNs received both direct and indirect inputs mainly in the dmNAcSh. Finally, we show that dHPC[CaMKII+]-induced reinforcement is mediated by the activation of dynorphin-containing MSNs in the dmNAcSh. Collectively, our data suggest that the dHPC[CaMKII+]→dmNAcSh pathway is sufficient to drive reinforcement via the activation of dynorphin-containing MSNs found in the dmNAcSh. These findings underline the potential role of the dHPC[CaMKII+]→dmNAcSh pathway in other reward-related behaviors.

## Results
### Stimulation of excitatory neurons in the dHPC promotes reinforcement

To determine whether the activity of excitatory neurons in the dHPC is sufficient to promote and maintain reinforcement, we used the real-time place test (RTPT) paradigm (Fig. 1A). We injected a viral construct carrying channelrhodopsin (ChR2) under Ca[2+]-calmodulin (CaM)-dependent protein kinase II (CaMKII) promoter (AAV5-CaMKII-ChR2-eYFP) to selectively stimulate excitatory neurons in the dHPC of wild-type mice. Animals were implanted with an optic fiber secured above the dHPC, 3 weeks after viral injection to allow optimal opsin expression (Fig. 1B). A week after the last surgery, mice were placed in an unbiased custom-made RTPT apparatus composed of 2 black matte compartments with no external cues or context (Fig. 1A). During the RTPT procedure, mice's presence in the stimulation-paired compartment triggered laser stimulation while the presence in the non-stimulation-paired compartment ended laser stimulation[41]. During the

30-minute test, the control (Ctrl) mice group lacking ChR2 (AAV5-CaMKII-eYFP; Ctrl; Fig. 1C) showed similar time spent in each of the RTPT compartments. On the contrary, mice expressing ChR2 in the dHPC spent significantly more time in the stimulation-paired compartment throughout the session (Fig. 1D, E). To further assess if mice would actively seek dHPC[CaMKII+] neuron activation, we used an operant self-stimulation procedure (Fig. 1F). One week after the surgical procedures (as described above), mice were exposed to 10 daily operant sessions during which light stimulation was delivered upon every active (illuminated) inset interaction (Fixed Ratio 1; FR1) and followed by a 10-second time-out (TO). The daily session was terminated after one hour or upon reaching 100 light stimulations (Max stimulation), whichever came first, to prevent unwanted side effects produced by overstimulation of the dHPC[CaMKII+] neurons[44,45]. On the first training day, animals expressing ChR2 in dHPC[CaMKII+] neurons significantly sought optogenetic self-stimulation. This behavior was maintained throughout all acquisition sessions, as demonstrated by the significant difference in the number of nose pokes into the active inset versus the inactive inset (Fig. 1G). On the contrary, animals injected with a control virus (control group) did not discriminate between the active and inactive insets, confirming the reinforcing nature of dHPC[CaMKII+] neuronal activation (Fig. 1G). Furthermore, ChR2-injected mice steadily sought self-stimulation throughout each session, earning a significantly higher number of optogenetic self-stimulations compared to the control mice (Fig. 1H). The same cohort of mice then underwent extinction sessions during which all cues remained the same, but interaction with either nose inset had no further consequence (laser turned OFF). During extinction sessions, mice expressing ChR2 in dHPC[CaMKII+] neurons significantly decreased their interaction with the active inset, uncovering the necessity for dHPC[CaMKII+] neurons activation to maintain self-stimulation (Fig. 1I). Lastly, 24 h after the last extinction session, mice were exposed to a "reinstatement session" during which the laser was turned back ON and a nose poke in the active inset triggered light stimulation of dHPC[CaMKII+] neurons. During reinstatement, mice expressing ChR2 demonstrated a significantly higher number of interactions with the active inset than the inactive inset (Fig. 1J). In addition, the ChR2-expressing animals also exhibited a significantly higher number of interactions with the active inset as compared to control littermates (Fig. 1J). Overall, these findings uncover the sufficiency of dHPC[CaMKII+] neuron activation to trigger self-stimulation. To rule out the possibility that reinforcing behavior is maintained by the saliency of the cue light, we trained a separate cohort of animals to self-stimulate in the presence of cue light and then exposed them to 4 sessions additional sessions during which the cue light was switched off, but interaction with the active inset still provided light stimulation (Supplementary Fig. 1B). The absence of cue light did not impact the ratio of active vs inactive pokes (Supplementary Fig. 1C), the average number of pokes on the active inset (Supplementary Fig. 1D), or the rate that mice sought for the light stimulation (Supplementary Fig. 1E). This finding demonstrates that the maintenance of self-stimulation is not driven by the salience of the cue light but rather by the activation of dHPC[CaMKII+] neurons themselves. Additionally, when removing the time out period after laser stimulation (TO) and the maximum number of stimulations that mice can achieve per session (100 Max stimulations; Supplementary Fig. 1A), mice significantly increased their interactions with the active inset (Fig. 1K) to a level reported in other studies aiming to dissect the role of other brain areas driving reinforcing behavior[46,47], and steadily sought stimulation of dHPC[CaMKII+] neurons stimulation throughout the session (Fig. 1L). Lastly, to further confirm that cue light saliency was not triggering self-stimulation behavior, we assessed if mice would engage in self-stimulation without the presence of cue lights, timeout (TO) period, or a maximum limit on the number of stimulations, starting from the first day of acquisition (Supplementary Fig. 1F). Similar to our previous results, mice expressing ChR2 in the dHPC exhibited a

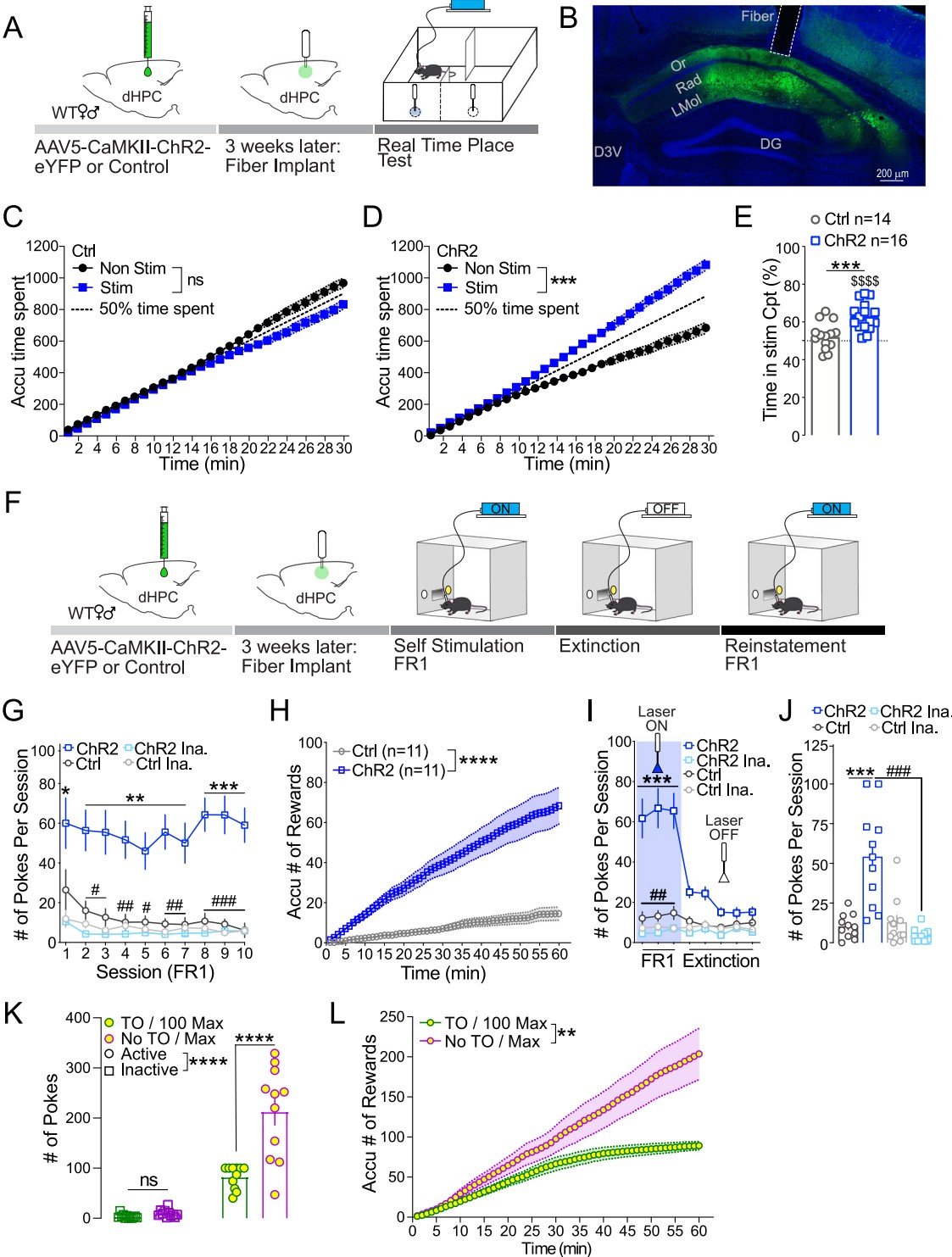

significantly higher number of interactions with the active versus inactive nose inserts. In comparison, control mice did not differentiate between the active and inactive nose inserts (Supplementary Fig. 1G). To fully assess the robustness of the self-stimulation protocol in the absence of cue lights, we switched the active and inactive nose inserts (active becoming inactive and inactive becoming active) during sessions 6-8. Despite this switch and the absence of cue lights to signal the availability of stimulation, ChR2-expressing mice maintained a steady number of self-stimulations obtained during the session and a significantly higher number of active nose pokes compared to inactive nose pokes (Supplementary Fig. 1H). Collectively,

these findings provide strong evidence that the activation of CaMKII+ neurons within the dHPC is sufficient to reinforce instrumental behavior.

## dHPC^CaMKII+ neurons are anatomically and functionally connected to the dorso-medial NAcSh

Recent evidence has uncovered direct dHPC inputs to the NAc and its role in the behavioral manifestation of spatial appetitive memory[23]. In line with those results, we also observed that dHPC^CaMKII+ neurons project preferentially to the rostral part of the NAc[23] and that most of those projections are localized within the NAcSh (Fig. 2 and

**Fig. 1 | Stimulation of dHPC CaMKII⁺ neurons drives reinforcing behavior.**
**A** Schematic illustrating real-time place test (RTPT) protocol. **B** Representative 10x magnification image of the viral expression and fiber placement in the dHPC. **C**, **D** ChR2 mice ($n = 16$) increased their preference for the stimulation-paired compartment within the session (Two-way ANOVA, $F_{1,15} = 21.56$, $p = 0.0003$), while control did not ($n = 14$; Two-way ANOVA, $F_{1,13} = 4.386$, $p = 0.0564$). **E** ChR2 mice (blue) demonstrated a significant preference for (two-tailed $t$-test, ChR2 vs 50% of time: $t_{15} = 6.967$, \$\$\$\$ $p < 0.0001$) and spent more time in the stimulation-paired compartment than control (grey) (two-tailed $t$-test: $t_{28} = 4.372$, \*\*\*$p = 0.0002$). **F** Schematic outlining self-stimulation protocol. **G** ChR2 mice (blue; $n = 11$), but not control (grey; $n = 11$), sought and maintained dHPC stimulation for 10 acquisition sessions (Two-way ANOVA, ChR2 vs Ctrl: $F_{3,40} = 28.35$, $p < 0.0001$ with Tukey's multiple comparisons; \* Active vs Inactive ChR2 groups; # ChR2 vs Ctrl Active groups). **H** ChR2 mice (blue) sought dHPC stimulation throughout the last session at a significantly higher level

than control (grey; Two-way ANOVA, ChR2 vs Ctrl: $F_{1,19} = 33.84$, $p < 0.0001$ with Sidak's multiple comparisons). **I** Uncoupling active inset with dHPC stimulation (laser turned OFF) resulted in ChR2 (blue; $n = 11$) mice to extinguish their seeking behavior while control mice (grey; $n = 11$) were unaffected (Two-way ANOVA, ChR2 vs Ctrl: $F_{3,42} = 27.08$, $p < 0.0001$ with Tukey's multiple comparisons indicating no significance during extinction sessions; \* Active vs Inactive ChR2 groups; # ChR2 vs Ctrl Active groups). **J** ChR2 mice reinstate behavior (laser ON) with significantly higher active pokes compared to the inactive and control active pokes ($n = 11$; One-way ANOVA, $F_{3,40} = 18.33$, $p < 0.0001$ with Tukey's multiple comparison). **K** Mice significantly increase the number of active pokes in the absence of the TO and max criteria ($n = 11$; Two-way ANOVA, $F_{1,11} = 31.10$, $p = 0.0002$ with Sidak's multiple comparisons). **L** The accumulation of rewards is significantly higher in the absence of TO and max stimulations (Two-way ANOVA, $F_{1,11} = 14.12$, $p = 0.0032$ with Sidak's multiple comparisons). Data are expressed as mean ± S.E.M.

Supplementary Fig. 2). To determine whether stimulation of dHPC$^{CaMKII+}$ neurons drives activation of the NAc neuronal population in freely moving animals, we combined local field potential (LFP) recording in the NAc with optogenetic stimulation in the dHPC. Briefly, after receiving CaMKII⁺ promoter-driven ChR2 (AAV5-CaMKII-ChR2-eYFP) in the dHPC, a fiber implant was permanently secured above the dHPC and 16 electrodes were implanted within the rostral part of the NAc. One week after surgical procedures, LFP was recorded in response to dHPC$^{CaMKII+}$ neurons stimulation (Fig. 3A). Pre- and post-stim amplitude were calculated by subtracting the minimum amplitude from the maximum amplitude within a 0.1 s window pre- and post-dHPC stimulation (Fig. 3B). Stimulation of dHPC$^{CaMKII+}$ neurons evoked a significant increase in overall evoked LFP amplitude recorded in the NAc (Fig. 3C). By calculating the time between dHPC stimulation onset and the first evoked peak (Latency to peak; Fig. 3B), we observed a wide variability in latency to peak (Fig. 3D), suggesting the presence of monosynaptic inputs (latency <0.01 s) and polysynaptic inputs (latency > 0.01 s) from the dHPC[48]. The well-defined role of the NAcSh in processing reward and positive reinforcement[36–40], together with our finding of denser dHPC$^{CaMKII+}$ projections into the NAcSh (Fig. 2C), suggest the possible involvement of this pathway to mediate dHPC$^{CaMKII+}$-driven reinforcing behavior. Interestingly, we uncovered two segregated areas of projections, namely in the dorso-medial (dmNAcSh) and ventro-lateral (vlNAcSh) areas of the NAcSh (Fig. 2C). Considering the involvement of the dmNAcSh in reward and reinforcement[41–43], we further assessed if dHPC projections can selectively trigger neuronal activation in the dmNAcSh. To achieve this goal, we combined in-vivo fiber photometry with an operant self-stimulation instrumental task (as described above). Briefly, we expressed a calcium indicator under a ubiquitous synapsin promoter within the dmNAcSh (AAV9-hSyn-GCamp6f-eYFP) or control (AAV9-hSyn-eYFP) and ChR2 in dHPC$^{CaMKII+}$ neurons before implanting fiber optics above dmNAcSh and dHPC. This allowed us to selectively record real-time calcium transients within the dmNAcSh during dHPC$^{CaMKII+}$ neurons self-stimulation. Mice were trained to acquire self-stimulation operant behavior prior to the start of the in-vivo fiber photometry recordings (Fig. 3E). During self-stimulation session in which dmNAcSh fiber photometry recording was done, mice remained actively engaged in seeking dHPC$^{CaMKII+}$ stimulation (Fig. 3F). The reinforcing behaviors mediated by dHPC$^{CaMKII+}$ stimulation coincide with a significant increase in calcium transients (Fig. 3G, H) and a greater area under the curve (AUC) 2 s pre- and post-onset of dHPC stimulation (Fig. 3I) within the dmNAcSh in ChR2-GCaMP expressing mice, but not in ChR2-eYFP control mice. Importantly, the presentation of the cue light alone was not sufficient to increase calcium transients in the dmNAcSh suggesting that the activity of dmNAcSh was contingent on the stimulation of dHPC$^{CaMKII+}$ neurons (Supplementary Fig. 3A–D). We observed a similar increase in evoked calcium transient in the dmNAcSh when dHPC CaMKII⁺ neuronal activation was triggered by the experimenter (non-contingent light delivery) (Supplementary Fig. 3E–G). These

findings further support the idea that dHPC$^{CaMKII+}$ neurons drive reinforcing behavior via the activation of dmNAcSh neurons.

## dHPC$^{CaMKII+}$→dmNAcSh neurons are sufficient to drive reinforcement through glutamatergic transmission

While stimulation of dHPC$^{CaMKII+}$ neurons is sufficient to evoke dmNAcSh activity, the sufficiency of dHPC$^{CaMKII+}$→ dmNAcSh neurons to drive reinforcement remains to be determined. To selectively activate dHPC$^{CaMKII+}$ neurons that project to the dmNAcSh, we injected a retrograde AAV2-pgk-Cre virus in the dmNAcSh and a cre-dependent ChR2 under the CaMKII promoter in the dHPC (Fig. 4A). A week after the last surgical procedure, mice were exposed to an FR1 schedule of self-stimulation reinforcement. Similar to our results obtained with stimulation of the overall dHPC$^{CaMKII+}$ neurons (Fig. 1G), selective stimulation of dHPC$^{CaMKII+}$→dmNAcSh neurons induced rapid and significant discrimination between the active and inactive insets of the operant box (Fig. 4C). The number of interactions with the active inset significantly decreased when the inset was no longer associated with stimulation of the dHPC$^{CaMKII+}$→dmNAcSh projecting neurons (Laser OFF-Fig. 4C). Importantly, mice significantly increased their interactions with the active inset when the laser was turned back on (reinstatement, Laser ON), compared to both the number of inactive inset interactions and the number of interactions with the active inset during the last extinction session (Fig. 4C, D). A similar set of results was also observed when ChR2 was expressed in CaMKII⁺ neurons within the dHPC and a fiber implant was placed above the dmNAcSh to selectively activate dHPC$^{CaMKII+}$ terminals in the dmNAcSh (Supplementary Fig. 4A–C). The number of interactions with active inset with dHPC$^{CaMKII+}$→dmNAcSh terminal stimulation is also comparable with those observed with dHPC$^{CaMKII+}$→dmNAcSh somatic stimulation (Fig. 4C, Supplementary Fig. 4D). This is also further illustrated in Supplementary Fig. 4E where there is no difference between somatic and terminal active pokes or inactive pokes. These findings demonstrate that activation of dHPC$^{CaMKII+}$→dmNAcSh neurons is sufficient to drive reinforcing behavior.

To investigate whether the local glutamatergic transmission in the dmNAcSh is mediating the reinforcing behavior induced by dHPC$^{CaMKII+}$ neurons activation, we combined opto-stimulation of dHPC CaMKII⁺ neurons with local pharmacology. After infusion of CaMKII-driven ChR2 in the dHPC, implantation of optic fiber above the dHPC, and guide-cannula above the dmNAcSh, mice were trained to self-stimulate (Fig. 4E). Once the mice exhibited stable self-stimulation behavior (Fig. 4F), animals were infused with either a cocktail of AMPA/NMDA antagonists (CNQX:0.7 mM/AP5:1.6 mM) or aCSF as a control in the dmNAcSh 30 min prior to an FR1 self-stimulation session[1]. Each animal received both treatments and tests were performed 3 days apart in a counter-balanced manner (half received CNQX/AP5 before the first test, while the other half received aCSF). Blockade of glutamatergic signaling within the dmNAcSh blunted self-stimulation performances, as shown by a significant decrease in interactions with the active inset

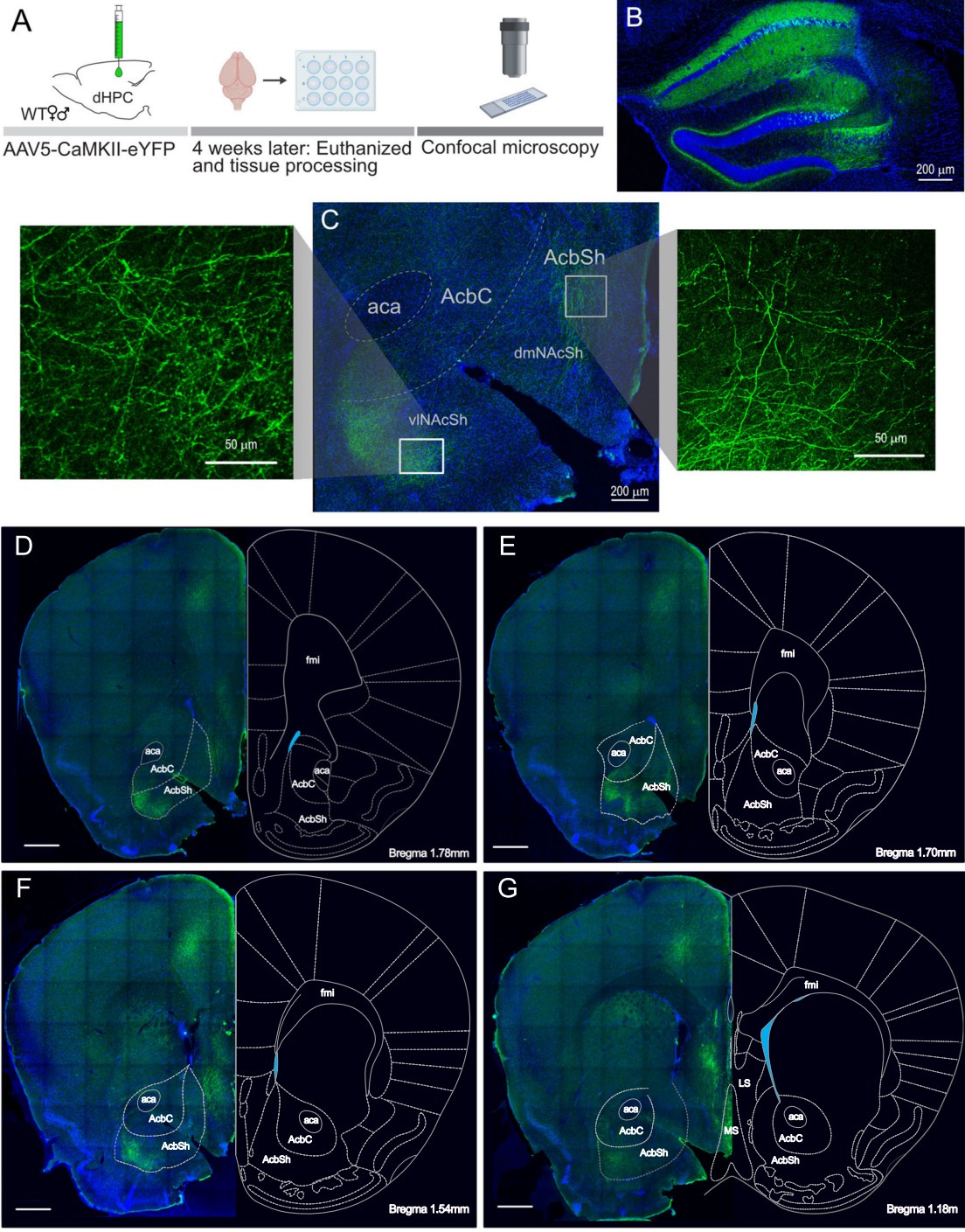

**Fig. 2 | Viral tracing of dHPC CaMKII⁺ fibers in the NAc and its surrounding areas. A** Schematic illustrating experimental protocol to determine the anatomical dHPC projections to the NAc (created with BioRender.com). **B** Representative images of viral expression in the dHPC (10x magnification), and (**C**) corresponding fiber terminals in the NAcSh at 10x and 20x magnification, highlighting fibers in the dorso-medial (dmNAcSh) and ventro-lateral (vlNAcSh) areas of the NAcSh. This viral expression was observed in all 5 mice that were injected. **D–G** Overall tracing of dHPC CaMKII⁺ fibers in brain regions that are 1.18 mm to 1.78 mm away from Bregma. Scale bars represent 500 μm.

after CNQX/AP5 infusion as compared to aCSF infusion (Fig. 4G). Altogether our data demonstrate that dHPC^CaMKII+→dmNAcSh neuronal activation promotes reinforcement mediated, at least in part, by local glutamatergic transmission in the dmNAcSh.

### dHPC^CaMKII+→dmNAcSh neurons contribute to natural reward seeking behavior

While previous evidence has shown that dHPC→NAc inputs are necessary for place-reward memory for food[23], the role of dHPC^CaMKII+→dmNAcSh in driving natural reward seeking using a goal-directed behavior procedure has yet to be determined. We selectively target dHPC^CaMKII+ neurons that project to the dmNAcSh by injecting a retrograde AAV2-pgk-Cre virus in the dmNAcSh and a cre-dependent Gi DREADDs under the CaMKII promoter (AAV9-CaMKII-DIO-hM4D(Gi)-mCherry) in the dHPC (Fig. 5A). Once trained to self-administer sucrose pellets under FR1, FR2, and FR5 schedules of reinforcement (Fig. 5D), mice received either CNO or saline for 4 consecutive days. This was

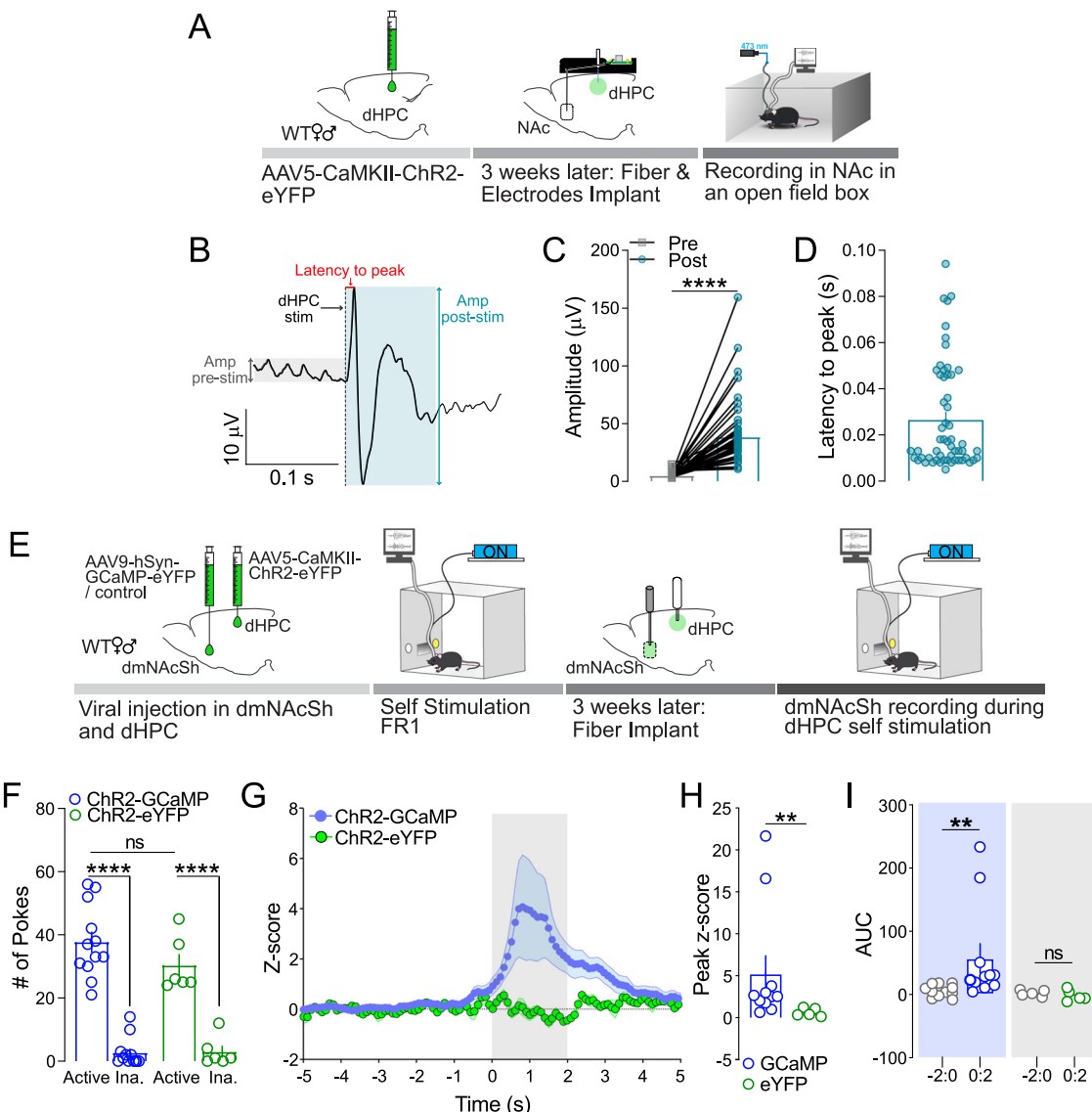

**Fig. 3 | Stimulation of dHPC excitatory neurons evoked neuronal activation in the dmNAcSh. A** Schematic outlining the protocol for evoked in-vivo local field potential (LFP) recording in the NAc. **B** Representative trace illustrating how the amplitude (amp) pre- and post-stimulation as well as latency to peak were calculated. The respective amplitude was calculated as the maximum voltage – minimum voltage within a 0.1 s window. The time between dHPC stimulation onset to the first evoked peak (Latency to peak) was calculated to determine direct input to NAc. **C** Stimulation of dHPC$^{CaMKII+}$ neurons evoked a significant increase in LFP amplitude in the NAc ($n = 56$ electrodes; two-tailed Wilcoxon matched-pairs signed rank test, $p < 0.0001$). **D** Variability in latency to peak suggests that recorded NAc cells received both direct and indirect inputs from dHPC$^{CaMKII+}$ neurons ($n = 56$ electrodes). **E** Experimental schematic of fiber photometry dmNAcSh recording with stimulation of dHPC$^{CaMKII+}$ neurons in self-stimulation operant boxes. **F** Both ChR2-GCaMP (($n = 11$) and ChR2-eYFP ($n = 5$) mice significantly sought for dHPC self-

stimulation (Two-way ANOVA, Active vs Inactive: $F_{1,16} = 184.7$, $p < 0.0001$ with Sidak's multiple comparisons). **G** An increase in calcium transients in the dmNAcSh upon optical self-stimulation of ChR2 expressing CaMKII+ neurons in the dHPC was only observed in ChR2-GCaMP mice ($n = 11$) but not in ChR2-eYFP mice ($n = 5$). **H** The peak $Z$-score obtained from mice expressing both ChR2 and GCaMP ($n = 11$) is significantly higher than control littermates expressing ChR2 and eYFP ($n = 5$) in the dmNAcSh (Peak z-score GCaMP vs eYFP: two-tailed Mann Whitney, $p = 0.0055$). **I** The measured calcium transients Area Under the Curve (AUC) in the dmNAcSh of ChR2-GCaMP mice upon self-stimulation (ChR2-GCaMP: 0:2) were significantly higher than their respective baseline (ChR2-GCaMP −2:0 vs 0:2 ($n = 11$): two-tailed Wilcoxon matched-pairs signed rank test, $p = 0.002$). Stimulation of the dHPC had no effect on the calcium transients AUC in the control group ChR2-eYFP −2:0 vs 0:2 ($n = 5$): two-tailed Wilcoxon matched-pairs signed rank test, $p = 0.3125$). Data are expressed as mean ± S.E.M.

followed by another 4 consecutive days where the treatments (CNO vs Saline) were switched (Fig. 5E, red-filled symbols represent groups that received CNO). Inhibiting dHPC$^{CaMKII+}$→dmNAcSh neurons significantly reduced the number of interactions with the active insets and sucrose pellets obtained (Fig. 5E–H). This finding demonstrates that activation of dHPC$^{CaMKII+}$→dmNAcSh CaMKII$^+$ neurons is not only sufficient to drive reinforcing behavior but also contributes to goal-directed behaviors for natural rewards.

## Stimulation of dHPC$^{CaMKII+}$ neurons increase dmNAcSh activity in a cell type-specific manner

The NAc is composed of 90 - 95% GABAergic MSNs organized in two distinct peptidergic populations, enkephalin (Enk)- and dynorphin (Dyn)-MSNs, which have been identified and studied for their role in rewarding and aversive behaviors[42,43]. We used transgenic mice models to assess whether CaMKII$^+$ neurons in the dHPC are preferentially recruiting one MSN population over the other to drive reinforcement.

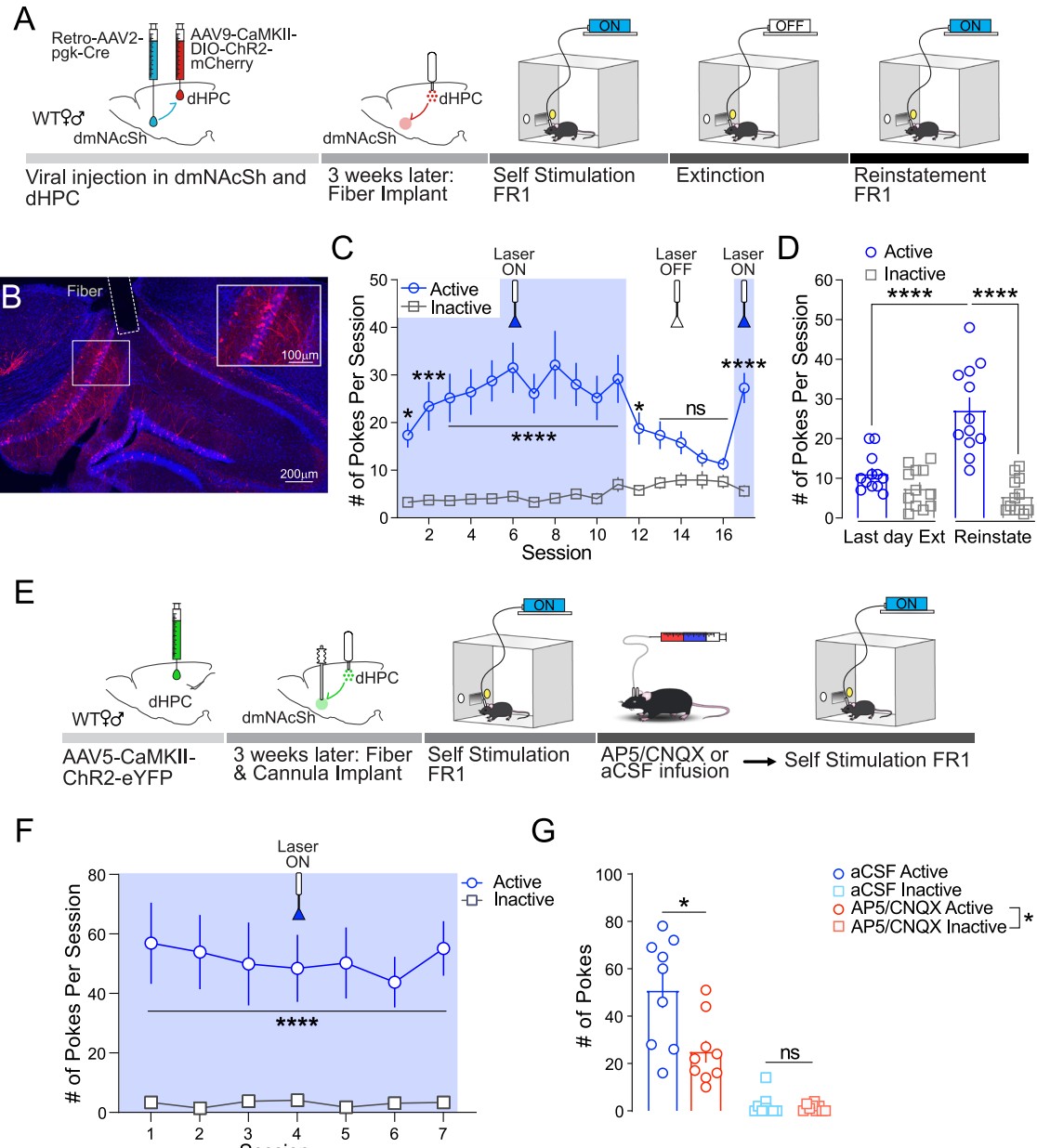

**Fig. 4 | Stimulation of dHPC-dmNAcSh projections is sufficient to drive reinforcement via glutamate transmission. A** Experimental schematic illustrating selective stimulation dHPC^CaMKII+ neurons that project to the dmNAcSh. Briefly, mice were injected with a retrograde virus carrying Cre recombinase in the dmNAcSh, Cre-dependent ChR2 in the dHPC, and optic fibers were secured above the dHPC. After recovery, mice were exposed to self-stimulation protocol. **B** Representative images of viral expression and fiber placement in the dHPC at 10x and 20x magnification (insert). **C** During the first FR1 session mice ($n = 12$) discriminated between the active and inactive nose insets, and this behavior was maintained throughout the 10 training sessions. The seeking behavior was abolished within the 5 days of extinction during which interaction with the active inset was not associated with dHPC→dmNAcSh stimulation (laser turned OFF). Self-stimulation behavior resumed when the laser was turned back ON during a reinstatement test performed 24 h after the last extinction session (Two-way ANOVA, *Active vs Inactive: $F_{1,11} = 32.14$, $p = 0.0001$ with Sidak's multiple comparisons). **D** Overall, the number of interactions with the active inset during the reinstatement test was

significantly higher than interactions with the inactive inset during reinstatement and the active inset on the last day of extinction ($n = 12$; Two-way ANOVA, Session x Active vs Inactive: $F_{1,11} = 29.70$, $p = 0.0002$, Sidak's multiple comparisons: Active vs Inactive during reinstatement: $p < 0.0001$; Last day Ext vs Reinstate active: $p < 0.0001$). **E** Experimental schematic to assess the necessity for dmNAcSh glutamatergic signaling in dHPC-induced reinforcing properties. **F** Within the first self-stimulation session, mice ($n = 9$) sought dHPC stimulation through persistent active nose pokes and maintained this behavior throughout the 7 training sessions (Two-way ANOVA, Active vs Inactive: $F_{1,8} = 24.09$, $p = 0.0012$ with Sidak's multiple comparisons). **G** Local micro-infusion of AP5/CNQX cocktail (AMPA/NMDA antagonists) within the dmNAcSh 30 min before self-stimulation session significantly decreased active inset interactions as compared to aCSF micro-infusion ($n = 9$; Two-way ANOVA, Treatment x Active vs Inactive: $F_{1,8} = 5.410$, $p = 0.0485$ with Sidak's multiple comparisons: Antagonist vs aCSF active: $p = 0.046$ and Antagonist vs aCSF inactive: $p > 0.9999$). Data are expressed as mean ± S.E.M.

Enk-IRES-Cre or Dyn-IRES-Cre mice were injected with a Cre-dependent calcium indicator (AAV9-Syn-Flex-GcAMP6f-eYFP) in the dmNAcSh and ChR2-CaMKII (AAV5-CaMKII-ChR2-eYFP) in the dHPC (Fig. 6A). Both Enk+ and Dyn+ neurons in the dmNAcSh exhibited a significant increase

in calcium transients upon light stimulation of the dHPC^CaMKII+ neurons as compared to their respective baseline (Fig. 6B). Although Enk+ neurons in the dmNAcSh exhibit a higher increase in calcium transient with light stimulation of dHPC^CaMKII+ neurons compared to Dyn+ neurons, no

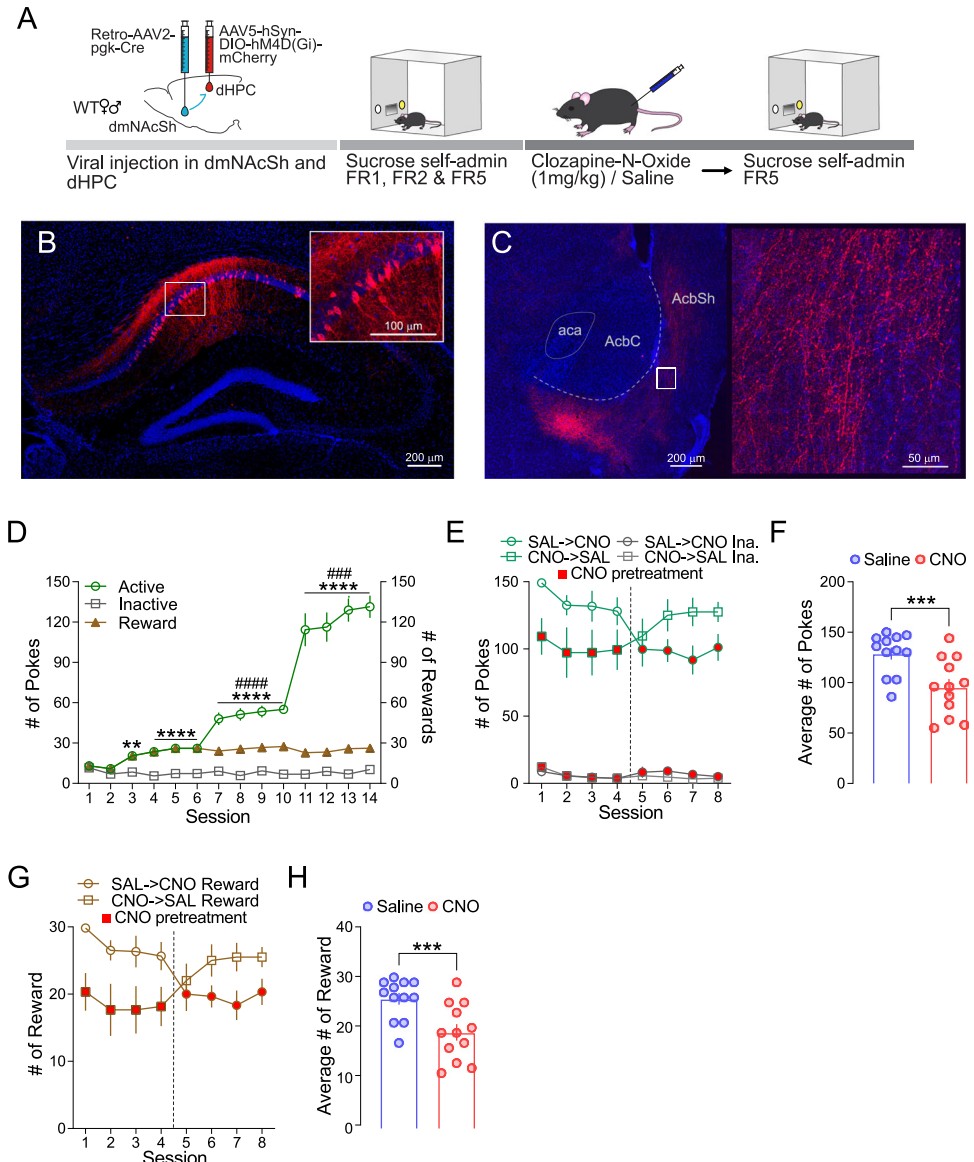

**Fig. 5 | dHPC-dmNAcSh projections modulate sucrose self-administration.**
**A** Experimental schematic illustrating selective inhibition of dHPC neurons projecting to the dmNAcSh during sucrose self-administration. Once mice have acquired FR5 sucrose self-administration, they received 4 sessions of CNO or SAL injections 30 min before the start of sucrose self-administration test (FR5). The treatment groups were counterbalanced such that mice receiving CNO in the first 4 sessions received SAL in the last 4 sessions and mice receiving SAL in the first 4 sessions received CNO in the last 4 sessions. **B** Representative images of viral expression in the dHPC at 10x and 20x magnification (insert). **C** Representative images of the terminal fiber expression in the NAcSh at 10x (left) and 20x magnification (right). **D** Mice rapidly discriminated active from inactive nose pokes during acquisition sessions. Mice increased the number of active nose pokes to maintain the number of sucrose pellets (rewards) received as the schedule increased from FR1 to FR5 ($n = 12$; Two-way ANOVA, Session x Active vs Inactive: $F_{2.415,26.57} = 78.66$, $p < 0.0001$ with Tukey's multiple comparisons * Active vs Inactive

and # Active vs Reward). **E** Mice ($n = 12$) that received CNO (red filled symbols) decreased the number of active nose pokes compared to mice that received SAL. When the treatment was switched (sessions 5–8), mice that previously received CNO, increased their number of active nose pokes while mice that previously received SAL decreased their number of active nose pokes. **F** Pretreatment with CNO before sucrose self-administration significantly decreases the average number of active nose pokes compared to their respective SAL pretreatment sessions ($n = 12$; CNO vs Sal: two-tailed Wilcoxon matched-pairs signed rank test, $p = 0.0005$). **G** The number of sucrose pellets obtained during a session was also attenuated when mice received CNO pretreatment before sucrose self-administration ($n = 12$). **H** Pretreatment with CNO significantly decreases the average number of sucrose pellets compared to SAL pretreatment before a sucrose self-administration session ($n = 12$; CNO vs Sal: two-tailed Wilcoxon matched-pairs signed rank test, $p = 0.001$). Data are expressed as mean ± S.E.M.

significant differences were observed between the two cell populations (Fig. 6C, D). To further dissect if dHPC$^{CaMKII+}$ projects to Dyn$^+$ and Enk$^+$ neurons in dmNAcSh, we examined the synaptic response of Dyn$^+$ and Enk$^+$ neurons to photo-stimulation of dHPC$^{CaMKII+}$ terminals in whole-cell patch clamp recordings. Mice were injected with AAV5-CaMKII-ChR2-eYFP in the dHPC and AAV9-hSyn-DIO-mCherry in the dmNAcSh of either Dyn-cre or Enk-cre mice before the experiments (Fig. 7A). mCherry-positive cells were visually identified and patched throughout

the dmNAcSh (Fig. 7B). Neurons were voltage-clamped at −70 mV, and the synaptic response to 1 ms wide-field photo-stimulation of dHPC$^{CaMKII+}$ projections was assessed. In the presence of GABA$_A$ receptor blockers, an inward current was observed in 9 out of 19 Enk$^+$ cells (−67.17 ± 16.83 pA) (Fig. 7C–F). This current was blocked by the glutamatergic antagonists NBQX (10 μM) and AP5 (50 μM, −3.35 ± 1.62 pA) indicating it is an excitatory connection (Fig. 7D–F). The latency and jitter of the evoked response were analyzed to determine if the input

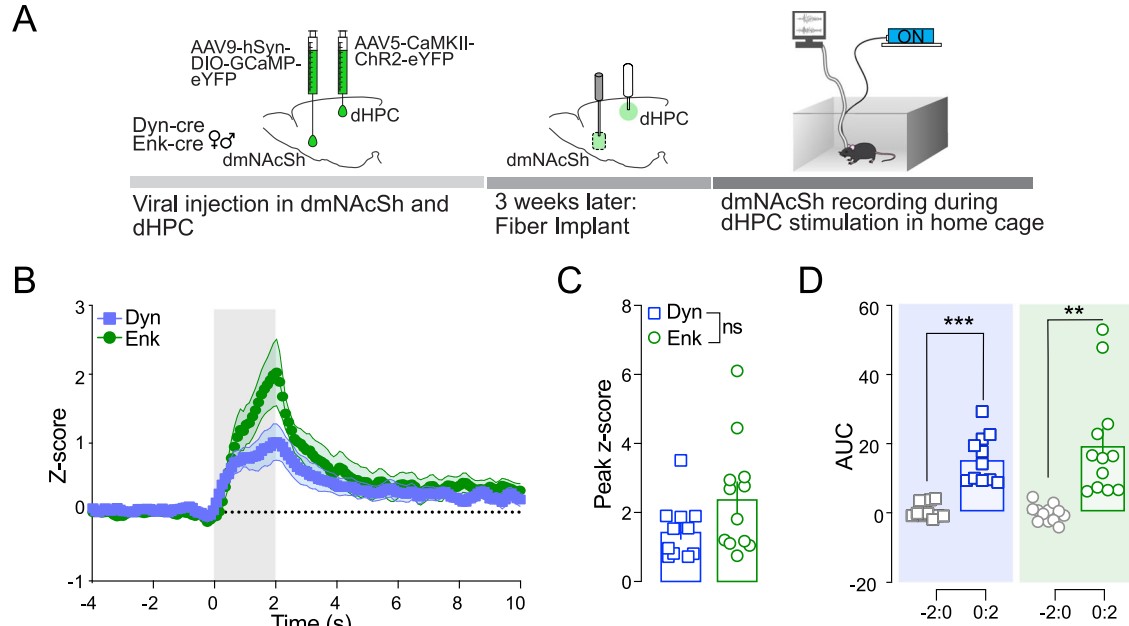

**Fig. 6 | dHPC CaMKII⁺ stimulation evoked response in both Dyn- and Enk-containing neurons in the dmNAcSh of behaving mice. A** Experimental schematic outlining fiber photometry recording in the dmNAcSh dynorphin (Dyn) or enkephalin (Enk) containing neurons with non-contingent (experimenter-induced) dHPC stimulation in the home cage. **B** Time course for the Z-scores of the evoked calcium transient in both Dyn and Enk containing neurons where the gray area represents dHPC stimulation. Stimulation of dHPC CaMKII⁺ neurons evoked increases in calcium transients in both Dyn ($n = 11$) and Enk ($n = 12$) containing neurons in the dmNAcSh. **C** The peak $Z$-score during the 2 s of dHPC stimulation was similar between dmNAcSh Dyn ($n = 11$) and Enk ($n = 12$) containing neurons (Dyn vs Enk: two-tailed Mann Whitney, $p = 0.1335$). **D** Area under the Curve (AUC) for $Z$-scores obtained in experimental groups demonstrate that both calcium transients recorded from Dyn ($n = 11$) and Enk ($n = 12$) containing neurons significantly increase during stimulation (0:2) compared to their respective baseline ($-2:0$) (two-tailed $t$-test; Dyn $-2:0$ vs 0:2: $t_{10} = 6.155$, $p = 0.0001$; Enk $-2:0$ vs 0:2: $t_{11} = 4.430$, $p = 0.001$). Data are expressed as mean ± S.E.M.

was monosynaptic. A mean latency of 4.65 ± 0.82 ms, and the synaptic onset variation, or jitter of 1.08 ± 0.19 ms (Fig. 7G) are consistent with previous studies of excitatory monosynaptic connections showing minimal variation in the onset of synaptic response[49–51]. Collectively, these findings suggest that dHPC^CaMKII+ makes a direct synaptic connection with Enk⁺ MSNs in the dmNAcSh with strong input observed in the ventral part of this subregion. In Dyn⁺ neurons, photo-stimulation of dHPC^CaMKII+ terminals yield terminals yield a response in 9 out of 64 cells (Fig. 7H). In contrast to Enk⁺ cells, this response was predominantly GABAergic, with a peak amplitude that dropped from −284 ± 99 pA at baseline to −47 ± 15 pA after applying the GABA_A blocker bicuculline (10 µM) (Fig. 7I–K). The small residual component was blocked by NBQX/AP5 (Fig. 7K). To determine whether the GABAergic response was still dependent on glutamatergic signaling following light stimulation, the order of drug application was reversed in one cell. When NBQX/AP5 was bathed applied first, the entire synaptic response was eliminated (Supplementary Fig. 5A) indicating both the direct and indirect synaptic response in Dyn+ cells is dependent on activation of glutamatergic dHPC inputs. The average latency in these Dyn⁺ cells was longer than that for Enk⁺ neurons (8.95 ± 1.30 ms, Fig. 7L), but still consistent with a monosynaptic response for most cells. However, 3 cells had latencies between 10–16 ms suggesting a possible polysynaptic connection (though still within the range that has been reported for monosynaptic light-evoked responses)[52]. Together, these data support a weak direct connection between the dHPC and Dyn+ neurons in the dmNAcSh, with a stronger indirect response likely mediated by local feedforward inhibitory circuits.

### Dynorphin-containing neurons within the dmNAcSh drive dHPC-mediated reinforcing behavior

To investigate the necessity of each type of MSN to drive dHPC^CaMKII+-mediated reinforcement within the dmNAcSh, we used a chemogenetic inhibitory approach. Enk-cre and Dyn-cre mice were injected with a cre-dependent Gi DREADD (AAV5-hSyn-DIO-hM4D(Gi)-mCherry) in the dmNAcSh and ChR2-CaMKII (AAV5-CaMKII-ChR2-eYFP) in the dHPC. Mice were first trained to self-stimulate dHPC^CaMKII+ neurons before receiving either CNO (1 mg.kg⁻¹) or saline (control) intraperitoneally 30 min before 4 consecutive self-stimulation sessions (Fig. 8A). For each animal, CNO or saline pretreatment was then switched for 4 additional consecutive self-stimulation sessions. Silencing the activity of Dyn⁺ neurons during the optogenetic self-stimulation sessions significantly reduced reinforcement (i.e., number of nose pokes in the active inset) (Fig. 8B, D, Supplementary Fig. 5C). However, silencing the activity of Enk⁺ neurons had no effect on optogenetic self-stimulation behavior (Fig. 8C, D, Supplementary Fig. 5D). Overall, while dHPC stimulation increases the activity of both Enk and Dyn MSNs within the dmNAcSh, our results uncover that the maintenance of dHPC reinforcing behavior is selectively mediated by the activation of Dyn⁺ neurons in the dmNAcSh.

## Discussion

In the current study, we used operant self-stimulation to demonstrate that activation of CaMKII⁺ neurons within the dHPC drives reinforcement behavior. Our findings uncover that dHPC-driven reinforcement, which previous studies had identified using electrical stimulation[34,35], is mediated, at least partially, via its glutamatergic inputs to the dmNAcSh. Despite many reports demonstrating the role of hippocampal pyramidal cells in encoding location, context, cues, and memory expression associated with appetitive and aversive behaviors[1,22,23,53,54], no study to date has assessed the sufficiency of dHPC^CaMKII+ neurons to drive reinforcement.

The NAc has been identified as a crucial brain area for the integration of behavioral responses to both positive and negative reinforcement[40,41,55]. Both positive and negative reinforcement maintain or increase the probability of specific behavior, and can contribute to many reward-associated disorders such as drug addiction[55,56].

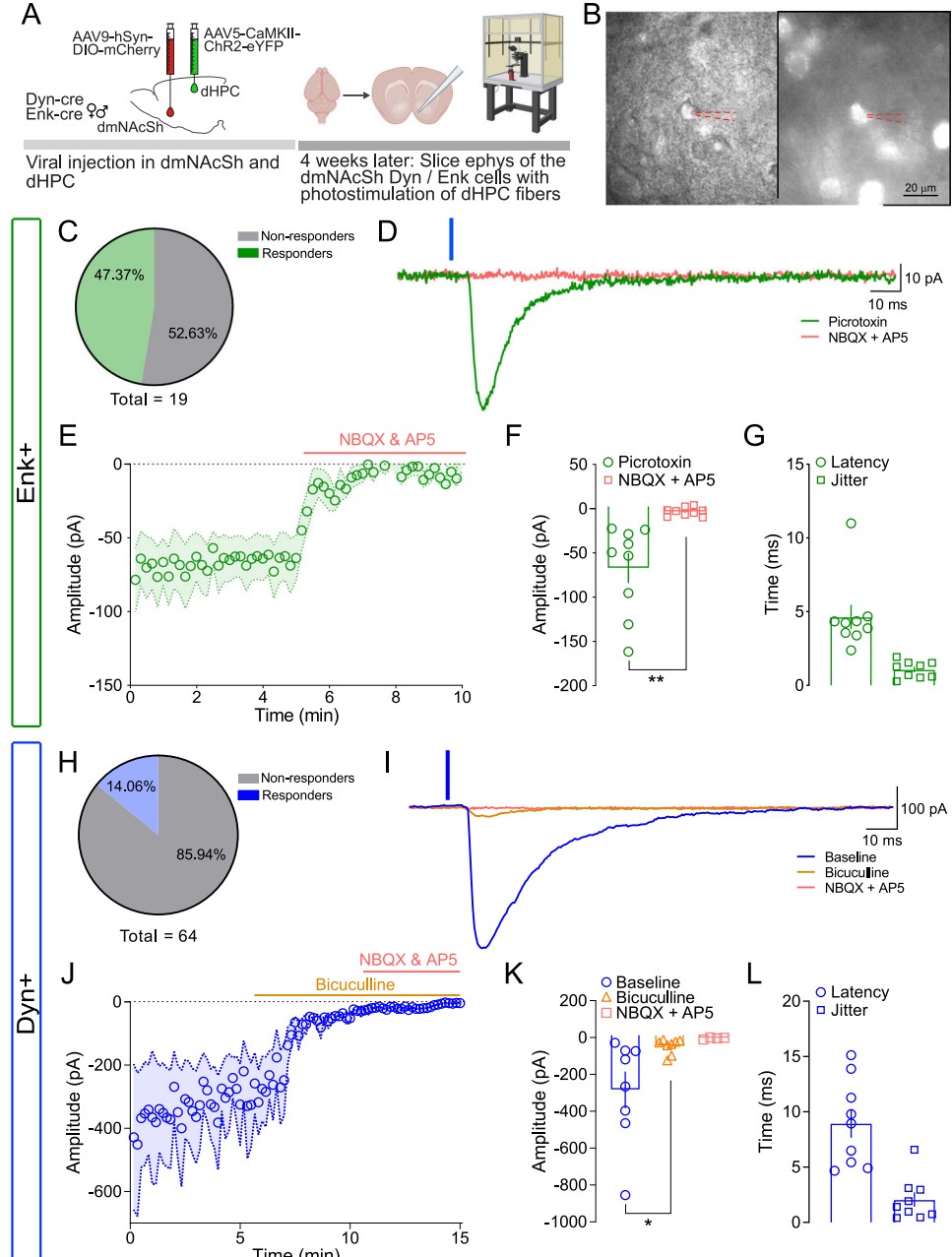

**Fig. 7 | Ex vivo electrophysiological stimulation of dHPC^CaMKII+ terminals in the dmNAcSh evoked responses in both Dyn- and Enk-containing neurons.**
**A** Experimental schematic outlining ex vivo electrophysiology experiments in Dyn+ and Enk+ cre mice (created with BioRender.com). **B** Representative images of a single cell patched under a light microscope (left) which is identified as mCherry positive using 570 nm LED fluorescence (right). The glass pipette is outlined by a dotted red line. **C** 47.4% of Enk+ cells patched responded to wide-field 470 nm stimulations, indicating that they received a monosynaptic input from dHPC^CaMKII+ neurons. **D** Representative trace of the Enk+ cells with 470 nm stimulation (indicated by the blue line above the traces) in the presence of picrotoxin in the bath. **E** The average evoked amplitude for all Enk+ cells was abolished when glutamatergic blockers (NBQX + AP5) were bath applied suggesting it is glutamatergic input. **F** The amplitude of evoked activation in the presence of NBQX + AP5 was significantly lower compared to the presence of only picrotoxin in the bath ($n = 9$ cells; two-tailed $t$-test, $t_8 = 3.993$, $**p = 0.004$). **G** The average latency of synaptic current

onset and jitter (variability in synaptic timing) show slight variation consistent with a monosynaptic connection ($n = 9$ cells). **H** 14.06% of Dyn+ cells that were patched responded to wide-field 470 nm stimulations and were predominately located in the dorsal region of the dmNAcSh. **I** Representative trace of the Dyn+ cells with 470 nm stimulation (indicated by the blue line above the traces) in the presence of GABAergic (bicuculline) and glutamatergic (NBQX and AP5) blockers. **J** The average evoked amplitude for all Dyn+ cells was mainly abolished when bicuculline were bath applied, and any remaining responses were subsequently blocked with NBQX + AP5 in the bath. This indicates that their activation is primarily driven by GABAergic signaling, with a limited contribution from glutamatergic input from the dHPC. **K** The amplitude of evoked activation in the presence of bicuculline was significantly lower compared to baseline ($n = 8$ cells; One-way ANOVA, $F_{2,17} = 4.634$, $p = 0.0248$ with Tukey's multiple comparison). **L** The average latency of the synaptic current onset and jitter suggest the presence of both monosynaptic and polysynaptic responses ($n = 9$ cells). Data are expressed as mean ± S.E.M.

Although there are a significant number of studies assessing the role of the NAc in reinforcing and reward-seeking behavior, dissecting the inputs to the NAc that drive these behaviors and the accumbal cell types involved warrants further study. Appetitive rewards such as sucrose not only trigger dHPC activity but also engage the NAc to encode guided appetitive behaviors[2–6]. However, these previous studies did not provide any insights into the potential role of excitatory transmission from dHPC terminals onto NAc synapses in driving

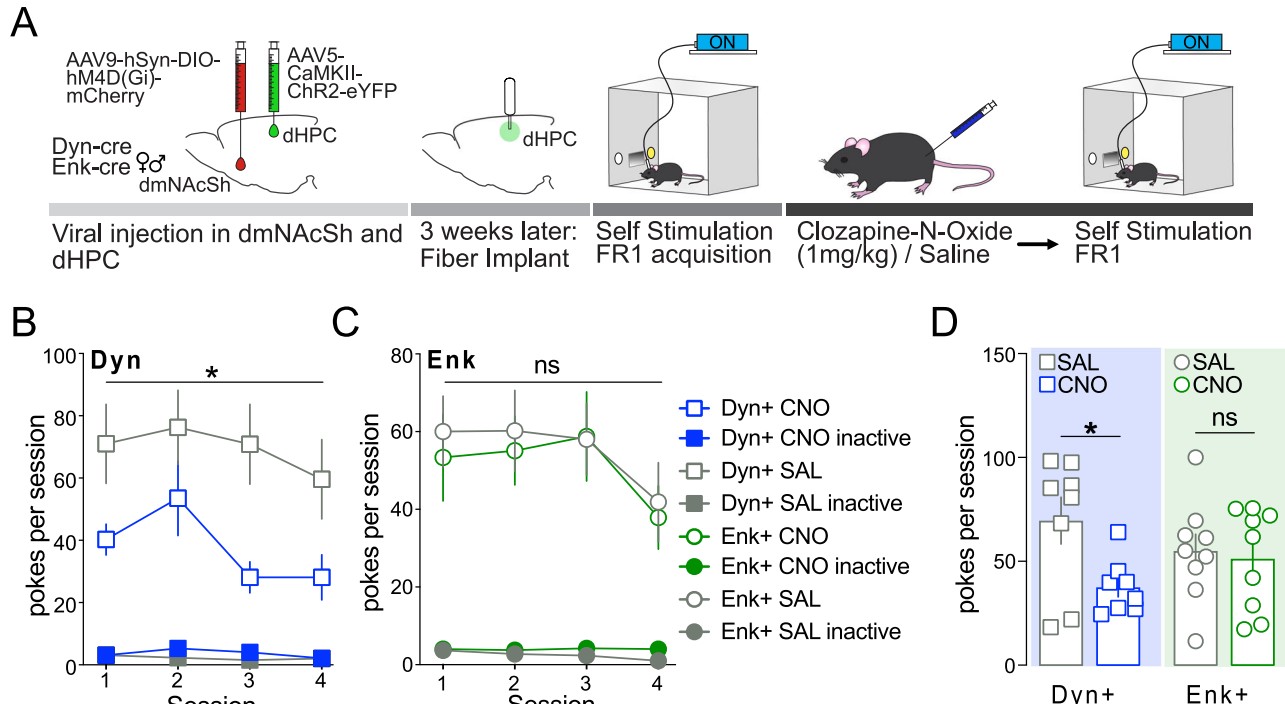

**Fig. 8 | Dynorphin-containing neurons in the dmNAcSh participate in the reinforcing properties of dHPC CaMKII⁺ stimulation. A** Experimental schematic outlining silencing Dyn or Enk containing neurons in the dmNAcSh during dHPC self-stimulation. All mice were exposed to 4 saline pretreatment and 4 CNO pretreatment sessions after acquiring the operant behavior. **B** Silencing Dyn-containing neurons within the dmNAcSh significantly reduces dHPC self-stimulation ($n = 8$; Dyn+ Sal vs Dyn+ CNO active: Two-way ANOVA, $F_{1,14} = 6.938$, $p = 0.0196$ with Sidak's multiple comparisons). **C** Silencing Enk containing neurons

within the dmNAcSh had no effect on dHPC self-stimulation ($n = 9$; Enk+ Sal vs Enk+ CNO active: Two-way ANOVA, $F_{1,16} = 0.1073$, $p = 0.7475$ with Sidak's multiple comparisons). **D** Overall, the number of stimulations is significantly attenuated when dmNAcSh Dyn containing neurons ($n = 8$) are selectively silenced, uncovering the role of those neurons in modulating dHPC CaMKII⁺-induced reinforcement (two-tailed $t$-test; Sal vs CNO: Dyn, $t_7 = 2.939$, $p = 0.0217$; Enk, $t_8 = 0.4428$, $p = 0.6696$; $n = 9$). Data are expressed as mean ± S.E.M.

reinforcing behavior. Our data demonstrate that self-stimulation of dHPC^CaMKII⁺ neurons is reinforcing and leads to significant increases in calcium transients in the dmNAcSh (Fig. 1, Fig. 3E–I). Thus, our findings extend the crucial role of dorsal hippocampal-accumbens functional connectivity in driving reward encoding[23,57,58] by uncovering its involvement in driving positive reinforcement.

The NAc receives dense excitatory projections including dopaminergic inputs from the VTA and glutamatergic excitatory inputs from cortical and subcortical structures such as the amygdala, the prefrontal cortex, and the vHPC[33,59–61]. Recently, Trouche et al. (2019) uncovered that a direct monosynaptic circuit between dCA1 CaMKII⁺ neurons and NAc parvalbumin fast-spiking interneurons (PV⁺ FSIs) and MSN neuronal populations is necessary for the expression of appetitive memory[23]. Additionally, they demonstrate that silencing those projections does not impact reward foraging in their conditioned place preference assay, confirming that silencing dCA1→NAc impairs the expression of CPP but this pathway is not necessary for reward-seeking itself. On the other hand, our study demonstrates that activation of dHPC^CaMKII⁺→dmNAcSh is sufficient to promote reinforcing behavior (Fig. 4A–D, Supplementary Fig. 4) and contributes to goal-directed behavior for natural rewards (Fig. 5), providing an additional level of complexity to the role of this pathway. We used in-vivo fiber photometry to determine the extent to which dHPC^CaMKII⁺ activation could recruit dmNAcSh activity. We demonstrate that the increase in dmNAcSh activity is contingent on stimulation of dHPC^CaMKII⁺ in the self-stimulation operant task (Fig. 3E–I) suggesting the engagement of the dmNAcSh in dHPC^CaMKII⁺ driven reinforcing behavior. However, infusion of AP5/CNQX into the dmNAcSh only partially decreased dHPC^CaMKII⁺-induced self-stimulation behavior, suggesting that the reinforcing behavior

driven by dHPC^CaMKII⁺ neurons may not be entirely reliant on the local glutamatergic transmission within the dmNAcSh. It is possible that dHPC^CaMKII⁺ projections to other regions or dHPC^CaMKII⁺→dmNAcSh collateral projections might still be active, thereby sustaining the reinforcing behavior.

The NAc MSNs are typically classified into two main populations according to their opioid peptidergic content (Dyn versus Enk) and dopamine receptor expression (D1 versus D2 receptor); Dyn⁺ MSNs vastly express D1 receptors while Enk⁺ MSNs express mostly D2 receptors[41,42,61]. The activity of Dyn⁺ (D1R) and Enk⁺ MSNs (D2R) has been well-established to play a critical role in rewarding or aversive experiences, respectively[42,43,55,62–67]. However, recent studies have reported conflicting findings, with the recruitment of Enk⁺ MSNs during appetitive/rewarding tasks[57,68] and Dyn⁺ MSNs necessary to drive aversive behavior such as negative affect and anhedonia[11,43]. Thus, considering the dichotomic roles of those two neuronal populations within the NAc, we pursued a cell-specific approach to further investigate the recruitment of each MSNs subtype in the dmNAcSh on dHPC^CaMKII⁺ driven reinforcement. Interestingly, our findings demonstrate that dHPC^CaMKII⁺ neuron activation triggers an increase in calcium transients in both Enk⁺ and Dyn⁺ MSNs within the rostral part of the dmNAcSh (Fig. 6).

To further characterize the function of dHPC^CaMKII⁺ inputs to the dmNAcSh in a cell-specific manner, we carried out ex-vivo electrophysiology studies in combination with optogenetics. Our finding provides further insight into the functional connectivity of dHPC^CaMKII⁺ projecting to the dmNAcSh MSNs by demonstrating that ~47.3% of Enk⁺ MSNs receive monosynaptic input whereas only ~14.1% of Dyn⁺ MSNs respond to input from the dHPC ^CaMKII⁺ neurons (Fig. 7). Furthermore, while dHPC^CaMKII⁺ inputs to Enk⁺ MSNs could be found

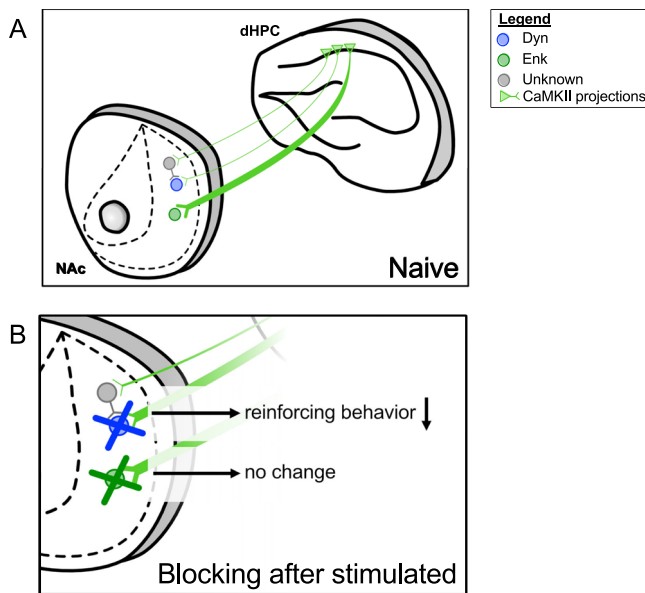

**Fig. 9 | Schematic detailing the projections from dHPC CaMKII[+] neurons to the Dyn- and Enk-containing neurons in the dmNAcSh. A** Our findings suggest that there is a differential connectivity pattern between the dHPC CaMKII[+] neurons and the MSNs in the dmNAcSh. A strong monosynaptic input from the dHPC CaMKII[+] neurons was mainly observed in the Enk neurons located in the ventral subregion of the dmNAcSh while a weaker input from the dHPC CaMKII[+] neurons was found in the Dyn neurons in the dorsal subregion of the dmNAcSh. **B** Inhibiting dmNAcSh Dyn neurons, but not dmNAcSh Enk neurons, decreases dHPC CaMKII[+] driven reinforcing behavior suggesting that the dHPC-dmNAcSh pathway's role in reinforcement is mediated, at least in part, by Dyn neurons.

throughout the dmNAcSh, Dyn[+] MSNs that respond to dHPC[CaMKII+] inputs were predominately found in the dorsal region of the dmNAcSh (only 1 of 9 Dyn[+] cells was located ventrally in the dmNAcSh; Figs. 7, 9). Those results are in accordance with a previous report[23] demonstrating that MSNs can either exhibit responses or not to direct dHPC[CaMKII+] neurons. Interestingly, while our results identify that dHPC[CaMKII+] neurons project to both subtypes of MSNs (Fig. 6) and similarly recruit both neuronal populations upon dHPC[CaMKII+] light stimulation (Fig. 6), only selective silencing of Dyn[+] MSNs was sufficient to dampen dHPC[CaMKII+]-induced reinforcing properties as indicated by a reduction in self-stimulation of dHPC[CaMKII+] neurons (Fig. 8). In contrast, silencing Enk[+] MSNs had no effect on dHPC[CaMKII+]-driven reinforcement. These results are somewhat in contrast with our ex vivo recordings showing a sparse, weak glutamatergic response in Dyn[+] neurons with a much stronger indirect GABAergic component. This suggests the dHPC[CaMKII+]-dmNAcSh connection may be dynamically regulated in vivo in a way that is not captured in slice experiments (for example, by dopaminergic signaling). Collectively, our findings illustrate that Dyn[+] MSNs located in the dmNAcSh not only receive input from dHPC[CaMKII+] neurons but also mediate the positive reinforcing behavior driven by these neurons. This outcome is consistent with previous literature describing that the activity of Dyn[+] MSNs in the medial NAcSh can both trigger appetitive and aversive behavioral outcomes depending on the dorsal/ventral anatomical distribution of the recruited neurons. Specifically, the stimulation of Dyn[+] MSNs in the dmNAcSh induced positive reinforcement behavior[41].

While the number of Dyn[+] MSNs responding to dHPC[CaMKII+] terminal stimulation in the dmNAcSh remains low (-14.1%), it is plausible that constant stimulation of dHPC[CaMKII+] input to dmNAcSh Dyn[+] MSNs, either through experience or training, would strengthen the pathway, making it key to modulate positive reinforcing behaviors. For example, constant exposure to drugs such as cocaine enhances excitatory synaptic connections into NAc MSNs[68].

The recent report of dHPC[CaMKII+] neurons recruiting FSIs, which in turn drives the activity of the MSNs assembly in the NAc[23], suggests the potential recruitment of the local circuitry in the NAcSh to regulate Dyn[+] neurons during dHPC[CaMKII+] driven reinforcing behavior. Therefore, further investigation is also necessary to fully determine the contribution of local circuitry in the NAcSh in dHPC[CaMKII+]-driven reinforcing behavior. Additionally, there is high heterogeneity in the anatomy and function of the NAc along the rostral/caudal axis[40], indicating the need for further investigation to fully evaluate the role of NAcSh MSNs in driving reinforcement, rewarding, and aversive behaviors.

In this study, we demonstrate that mice actively seek dHPC[CaMKII+] neuronal stimulation, regardless of the presence of discrete light cues (Fig. 1G, Supplementary Fig. 1). However, our findings do not negate the potential for dHPC[CaMKII+] neuronal stimulation to increase nose poking saliency. In this regard, further studies aimed at distinguishing between saliency-driven and reinforcement-based behavioral responses within the operant self-stimulation paradigm are warranted. Moreover, given the primary focus on the dmNAcSh in our current study, there is a clear need for future research to investigate the role of the observed dHPC[CaMKII+] → ventral lateral subregion of the NAcSh (Fig. 2, Supplementary Fig. 2), particularly regarding its potential involvement in reinforcement mechanisms.

Overall, our study brings an additional level of granularity to the role of dHPC in reward-seeking and memory processes. While previous studies have described the direct pyramidal dHPC→NAc projections[23,58,69] and their role in the expression of contextual memories[23], we demonstrate here that activating this pathway is also sufficient to drive reinforcement via the activity of Dyn[+] MSNs within the dmNAcSh. We propose that the reinforcing properties of dHPC[CaMKII+]→dmNAcSh projections, in conjunction with their involvement in cue and contextual memory, are critical in the expression of reward seeking. Hence, this study uncovers a role of the dHPC[CaMKII+]→dmNAcSh projections in mediating reward seeking and potentially drug use disorder and relapse.

## Methods
### Experimental model and subject details
All procedures were approved by the Washington University Institutional Animal Care and Use Committee (IACUC) in accordance with the National Institutes of Health Guidelines for the Care and Use of Laboratory Animals. Adult WT, PEnk-IRES-Cre, and PDyn-IRES-Cre C57BL/6 J male and female mice (20–30 g) were used for this study. All animals were 10 to 12 weeks at the beginning of the experiments. Four to five mice were housed together, given access to food pellets and water ad libitum, and maintained on a 12/12 h dark/light cycle (lights on at 7:00 AM). All animals were kept in a sound-attenuated, isolated holding facility in the lab 1 week before surgery, post-surgery, and throughout the duration of the behavioral assays to minimize stress. The housing temperature was maintained at 20–26 °C with humidity levels at 30–70%.

### Surgeries
All surgeries were performed under isoflurane (1.5/2 MAC) anesthesia under sterile conditions. Mice were anesthetized in an induction chamber (4 MAC Isoflurane) and placed into a stereotaxic frame (Kopf Instruments, Model 1900) where they were maintained at 1.5 MAC–2 MAC isoflurane. A craniotomy was performed and followed by a unilateral injection, using a blunt needle (86200, Hamilton Company), 500 nl of AAV5-CaMKII-ChR2-eYFP or, AAV5-CaMKII-eYFP (Hope Center Viral Vector Core, viral titer $2.8 \times 10^{13}$ vg/mL) into the dHPC (stereotaxic coordinates from Bregma: A/P: −1.70 mm, M/L: ±1.5 mm, D/V: −1.80 mm)[70,71]. For chemogenetic approaches, animals were injected with 250 nl of AAV5-DIO-hSyn-hM4Di-mCherry (Addgene, viral titer $2.8 \times 10^{13}$ vg/mL) in the dorsal-medial NAcSh (dmNAcSh; stereotaxic

coordinates from Bregma: A/P: +1.70 mm, M/L: ±0.5 mm, D/V: −4.50 mm) during the same procedure. Treatments were given bilaterally to ensure that any ChR2-induced recruitment of Enk⁺ or Dyn⁺ neurons would be silenced. For experiments involving retro-cre virus, 250 nl of Retro-AAV2-pgk-Cre (Addgene, viral titer $2.1 \times 10^{13}$ vg/mL) in the dmNAcSh (A/P: +1.70 mm, M/L: ± 0.5 mm, D/V: −4.50 mm) and 500 nl of either AAV9-CaMKII-DIO-ChR2-mCherry (Canadian Neurophotonics Platform Viral Vector Core, viral titer $5.3 \times 10^{12}$ vg/mL) or AAV5-hSyn-DIO-hM4Di-mcherry (Addgene, viral titer $2.5 \times 10^{13}$ vg/mL) in the dHPC (A/P: −1.70 mm, M/L: ±1.5 mm, D/V: −1.80 mm) of WT mice. For fiber photometry experiments, a calcium indicator AAV9-hSyn-GCAmp6f-eYFP (Addgene, viral titer $2.8 \times 10^{13}$ vg/mL) or a control virus AAV9-hSyn-eYFP (Addgene, viral titer $2.8 \times 10^{13}$ vg/mL) was injected in the dmNAcSh of WT mice while a cre-dependent sensor AAV9-Syn-Flex-GcAMP6f-eYFP (Addgene, viral titer $2.1 \times 10^{13}$ vg/mL) was injected in the dmNAcSh of Cre+ mice (stereotaxic coordinates from Bregma: A/P: +1.70 mm, M/L: ±0.5 mm, D/V: −4.50 mm). Three weeks after intracranial local injections, mice underwent a second surgery during which a fiber optic was implanted in the dHPC, and also above the dmNAcSh for fiber photometry experiments. Alternatively, a bilateral guide cannula was implanted 0.5 mm above the dmNAcSh (AP: +1.7 mm; ML: ±0.5 mm; DV: −4.25 mm) to allow local AMPA/NMDA antagonists infusion. Treatments were given bilaterally to ensure that any ChR2-induced dHPC→dmNAcSh glutamate signaling would be blocked. In local field potential experiments, a total of four holes were drilled into the skull, one above the left dorsal hippocampus (A/P: −1.70 mm, M/L: ±1.5 mm), two above the nucleus accumbens (A/P: +1.34 mm, M/L: ±0.8 mm) and one to secure the ground screw and wire. A unilateral injection, using a blunt needle (86200, Hamilton Company), of AAV5-CaMKII-ChR2-eYFP (500 nl, Hope Center Viral Vector Core, viral titer $2.3 \times 10^{13}$ vg/mL) into dHPC (D/V: −1.80 mm)[71]. After viral injection, the skull was gently scratched across and the custom-made head cap was carefully aligned before slowly lowered to the depth desired (1.80 mm DV for dHPC and 4.75 mm DV for dmNAcSh).

The implants or head caps were secured with Metabond dental cement using Radiopaque L-powder, B Quick Base, and C Universal TBB Catalyst (Parkell). Mice were allowed to recover for at least 1 week before conditioning or running any behavioral experiment. The efficiency of all constructs used in this manuscript has been tested in previous reports[36,48]. Furthermore, the experiments also started 4 weeks after any virus injection, permitting optimal viral expression in the targeted neurons.

### Chemogenetics, optogenetics, and behavioral assays
**Real-time place test (RTPT).** All behaviors were performed within a sound-attenuated room maintained at 23 °C after habituation to the holding room and the final surgery for at least 1 week. Movements were video recorded and analyzed using AnyMaze software version 7 (Stoelting Co). After recovery from both viral and fiber implant surgeries (see above section for the procedure), mice were gently placed in a custom-made unbiased and balanced two-compartment conditioning apparatus ($52.5 \times 25.5 \times 25.5$ cm) as described previously[41,43,72]. For Fig. 1A–E, the RTPT lasted for 30 min, where entry into one compartment triggered light stimulation (20 Hz, 5 ms pulse width, 4 mW). Light stimulation was delivered as long as the animal remained in this compartment and would only stop upon entry into the other compartment.

### Optogenetic self-stimulation
Mice operant-conditioning chambers (Med Associates version 5, Fairfax, VT) were equipped with nose insets accessible to animals. Mice were exposed to a Fixed Ratio 1 (FR1) session in which the fiber implant was connected to a 473 nm laser. A cue light was presented on the active inset and active interaction with this port resulted in a laser

activation (20 Hz, 15 ms pulses for 2 s, 4 mW) together with the cue light being turned off for 10 s (time out period; TO) and any interactions with the insets during this period had no consequences. Interaction with the inactive inset also had no consequences. Each daily session will terminate upon receiving 100 stimulations (Max stimulation) or after an hour, whichever comes first. Mice were trained to discriminate between active and inactive insets during the sessions. In Fig. 1F–J, Fig. 4A–D, and Supplementary Fig. 4A–C, animals underwent extinction sessions for 5 days during which all parameters were kept constant but interaction with the active inset did not result in light stimulation (laser was turned OFF). After animals extinguished their discrimination between the active and inactive insets, they underwent a single reinstatement procedure (FR1) during which interaction with the active inset led to the delivery of light stimulation.

In Fig. 1K–L, mice were first trained to self-stimulate under the FR1 schedule in the presence of 10 s TO and 100 Max stimulation sessions before undergoing 1 self-stimulation session without the 10 s TO and 100 Max. Similarly, in Supplementary Fig. 1A–D, after being trained to self-stimulate under the FR1 schedule in the presence of cue light to differentiate active from inactive insets, mice underwent 4 consecutive self-stimulation sessions in the absence of cue light in both active and inactive insets. In Supplementary Fig. 1F–H, mice were engaged in self-stimulation under the FR1 schedule in the absence of cue light, 10 s TO, and 100 Max stimulation from the first day of training.

In Fig. 5D–G, the animals were first trained to self-stimulate before receiving an intra-dmNAcSh infusion of either a cocktail of AMPA/NMDA antagonists (CNQX:0.7 mM/AP5:1.6 mM; 250 nl per side at 50 nl/min) or aCSF (control) 30 min before an FR1 self-stimulation session[73]. Treatments were given bilaterally to ensure that any ChR2-induced dHPC→dmNAcSh glutamate signaling would be blocked. The treatments were counterbalanced, where half of the animals received CNQX/AP5 30 minutes before the first test while the other half received aCSF. The second test was done 3 days after the first test where the local infusion of drugs was switched between the groups (i.e. previously received aCSF mice in the first test will receive CNQX/AP5 in the second test).

In Fig. 8, after training sessions, all animals were exposed to 8 days of 1-hour self-stimulation sessions. For 4 out of the 8 consecutive days mice received an intraperitoneal injection of CNO (1 mg.kg⁻¹) 30 min before each self-stimulation session. On the other 4 consecutive days, animals received an intraperitoneal injection of sterile saline 30 min before the self-stimulation. For both Enk-cre and Dyn-cre cohorts, the experiment was counterbalanced, with half of the animals receiving CNO on the first four days after training followed by 4 sessions with sterile saline pretreatment. The other half of the cohorts received intraperitoneal saline during the first four self-stimulation sessions post-training and CNO pretreatment 30 min before the last four self-stimulation sessions.

### Sucrose self-administration
The same mice operant-conditioning chambers (Med Associates version 5, Fairfax, VT) that were equipped with nose insets accessible to animals were used. Mice were exposed to an FR1 session at least 1 week after the final surgery. Within each self-administration session, the active inset was differentiated from the inactive inset by the presence of a cue light on the active inset. Interaction with the active inset resulted in 1 chocolate-flavored sucrose pellet being dispensed while interaction with the inactive inset had no consequences. Once mice had acquired the self-administration behavior, the schedule of reinforcement was changed to FR2 for 4 consecutive sessions and then FR5 for another 4 consecutive sessions. Once they have completed the training and it's at least 4 weeks of viral expression, mice were exposed to 8 days of 1-hour self-administration sessions. For 4 out of the 8 consecutive days, half of the mice received an intraperitoneal injection

of CNO (1 mg.kg⁻¹) while the other half received saline 30 min before each session. On the other 4 consecutive days, the treatments were switched between the groups (i.e. previously received CNO mice in the first test will receive saline in the second test).

## Immunohistochemistry

For tracing experiment in Fig. 2 and Supplementary Fig. 2, mice were anesthetized with 1–3% isofluorane and head-fixed in a stereotaxic apparatus (Stoelting). 500 nl of AAV5-CaMKII-eYFP (Hope Center Viral Vector Core, viral titer $2.8 \times 10^{13}$ vg/mL) virus was infused bilaterally into the dHPC (rate of infusion: 100 nl/min; Coordinates: A/P: −1.7 mm, M/L: ±1.5 mm, D/V: −1.8 mm). Mice were transcardially perfused with ice-cold 4% paraformaldehyde (PFA) in phosphate-buffered saline (PBS) while deeply anesthetized with isofluorane 4 weeks after surgery to allow maximum viral expression. To validate viral expression in Fig. 4A–D, Fig. 5, Supplementary Fig. 4, and Supplementary Fig. 5B–D, mice were perfused with 4% PFA in PBS after each behavioral experiment. The whole brain was extracted and post-fixed at 4 °C for 24 h in 4% PFA before the brains were transferred for equilibration in 30% sucrose in PBS for 72 h. Equilibrated brains were then flash-frozen using isopentane and the entire brain was sectioned into 40 μm slices using a cryostat (Leica CM 1950). Free-floating sections of the NAc and dHPC were washed in PBS and blocked with 4% normal donkey serum (Millipore, S30) and 0.3% Triton-0.01 M PBS (PBS-T). The sections are then incubated overnight in 3% normal donkey serum and 0.3% PBS-T containing either chicken anti-GFP (1:500, ab13970 Abcam) or chicken anti-mCherry (1:2000, ab205402 Abcam). The following day, sections were washed with 3 × 5 min PBS and incubated (2 h) with the appropriate secondary antibodies diluted in 3% normal donkey serum and 0.3% PBS-T (for GFP: donkey anti-chicken AF488 at 1:200, #730-545-155 Jackson ImmunoResearch and for mCherry: donkey anti-chicken Cy3 at 1:200, #730-165-155 Jackson ImmunoResearch). Slides were cover slipped with hardset antifade mounting media with DAPI (Vectashield) and imaged via confocal microscopy (Leica).

## Local field potential recordings

**Custom-made electrode and head cap.** The electrode was made in-house using 16 tungsten wires (0.0015", coated, California Fine Wire) for recording and 1 silver wire (0.010", coated, California Fine Wire) as ground. The ground wire was soldered on an 18-pin, 16-channel electrical interface board (EIB) (Triangle BioSystems Inc.) while the recording wires (8 on the left and 8 on the right side of the EIB) were connected using small gold EIB pins (Triangle Biosystems Inc.). The electrode was then secured to a 3D-printed head cap that was designed to hold both the electrode and optic fiber at the desired coordinates during implant. Optic fibers (Thorlabs) were cleaved at a length that allowed 2.1 mm to protrude out from the bottom of the head cap. The 8 tungsten wires on each side are then twisted and pulled to make them straight before being cut to the desired length (5 mm) for the implant.

**Experiment LFP protocol.** Mice were habituated to a black opaque testing chamber (17 cm width × 30.5 cm length × 25 cm height) and the electrophysiology recording setup for 3 consecutive days before test day. 1 day before the test day (3rd day of habituation), a recording of the evoked local field potential (LFP) was taken to ensure that there was a stable response in the NAc with stimulation of the dHPC$^{CaMKII+}$ neurons. On test day, mice were connected to a head-stage amplifier (Plexon model HST/16D Gen2) and PLEXON amplifier (Plexon Omniplex-D acquisition system) and allowed to habituate for 30 min before the start of recording. The neuronal recordings were collected using the OMNIPLEX PLEXON neurophysiology acquisition system and software (version 1.18; Plexon Inc). The dHPC was stimulated for a duration of 5 ms every 5 s while evoked LFP in the NAc was recorded. The peri histogram of evoked LFP recorded in the NAc was analyzed using Neuroexplorer version 5 (Plexon Inc).

## Fiber photometry

For fiber photometry studies, an optic fiber was attached to the implanted fiber using a ferrule sleeve (Doric, ZR_2.5). Two LEDs were used to excite GCaMP6f in WT or cre-dependent mice. A 531-Hz sinusoidal LED light (Thorlabs, LED light: M470F3; LED driver: DC4104) was bandpass filtered (470 ± 20 nm, Doric, FMC4) to excite GCaMP6f and evoke $Ca^{2+}$-dependent emission. A 211-Hz sinusoidal LED light (Thorlabs, LED light: M405FP1; LED driver: DC4104) was bandpass filtered (405 ± 10 nm, Doric, FMC4) to excite GCaMP6f and evoke $Ca^{2+}$-independent isosbestic control emission. Laser intensity for the 470 nm and 405 nm wavelength bands was measured at the tip of the optic fiber and adjusted to 50 μW before each day of recording. GCaMP6f fluorescence traveled through the same optic fiber before being bandpass filtered (525 ± 25 nm, Doric, FMC4), transduced by a femtowatt silicon photoreceiver (Newport, 2151), and recorded by a real-time processor (TDT, RZ5P). The envelopes of the 531 Hz and 211 Hz signals were extracted in real time by the TDT program Synapse (version 95) at a sampling rate of 1017.25 Hz.

For Supplementary Fig. 3E–H and Fig. 6, mice received sustained 2 s of non-contingent stimulation in dHPC (20 Hz, 5 ms, every 30 s, 4 mW) to stimulate ChR2 expressing CaMKII⁺ neurons within the dHPC. For Fig. 3E–I, a separate cohort of mice was first trained in operant conditioning (Med Associates Inc.) as described in the behavioral paradigm above (self-stimulation). When self-stimulation was consistent across sessions, fiber photometry recordings were made throughout one-hour operant conditioning sessions. All signals were aligned to light-stimulation delivery, or cue light turning back on after time out (TO) period as a control. The resulting signal recorded by TDT Synapse (version 95) program was analyzed using custom-written MATLAB scripts (using MATLAB version 2019b). In brief, signals from the isosbestic 405 nm control channel were smoothed and regressed on the smoothed GCaMP 470 nm signal to generate a predicted 405 nm channel using a linear model generated during the regression. Changes in fluorescence across the experiment session (ΔF) were then calculated by subtracting the predicted 405 nm from the raw GCaMP signal to remove photo-bleaching and fiber-bending artifacts. Signals from the GCaMP channel were then divided by the control signal to generate the ΔF/F. Peri-event histograms (Z-score) were then created by averaging changes in fluorescence (ΔF/F) across repeated trials using a duration window that encompassed events of interest (stimulation or cue)[74].

## Slice electrophysiology

Nucleus Accumbens (NAc) slice preparation: Mice were anesthetized and transcardially perfused with cold N-methyl-D-glucamine (NMDG) solution containing (in mM): 93 NMDG, 2.5 KCl, 1.25 NaH2PO4, 30 NaHCO3, 20 HEPES, 25 glucose, 5 ascorbic acid, 2 thiourea, 3 Na⁺pyruvate, 5 MgSO₄, 0.5 CaCl₂, 12 N-acetylcysteine and pH to 7.3 with HCl. Brains were rapidly removed and embedded in 2% low-melt agarose. Coronal slices (250 μm) of NAc were cut in cold NMDG solution equilibrated with 95% O₂ and 5% CO₂ using a Compresstome (Precisionary Instruments, cat # VF210-0Z). Slices were transferred to a recovery chamber containing NMDG at 32 °C for 10 min before transfer to a holding chamber containing artificial cerebral spinal fluid (aCSF) containing (in mM): 124 NaCl₂, 2.5 KCl, 1.25 NaH₂PO₄, 24 NaHCO₃, 5 HEPES, 12.5 glucose, 1 MgCl₂, 2 CaCl₂. The slices were kept in the dark and allowed to recover for a minimum of 1 h before recording.

Whole-cell patch clamp recordings: Slices were placed in a submersion chamber and continuously perfused (1–2 mL/min) with room temperature aCSF. Recordings were performed using pClamp software (version 11.1) controlling a Molecular Devices 700B amplifier. Neurons were visualized using infrared DIC (IR-DIC 770 nm) illumination, and mCherry positive cells were identified with a custom 570 nm LED coupled to the back fluorescence port of the microscope (Olympus, BX-51). Voltage clamp recordings were performed at −70 mV

using glass pipettes with resistances of 3–5 MΩ when filled with CsCl intracellular solution containing (in mM): 140 CsCl, 10 EGTA, 0.1 CaCl$_2$, 3 MgATP, 0.3 NaGTP, 10 HEPES, pH to 7.29 with CsOH, 290 mOsm. Wide-field 470 nm photo-stimulation was delivered through the 40x objective using a LED triggered by TTL pulses from the amplifier (Molecular Devices, 700B). To determine if GABAergic signaling contributed to the response, picrotoxin (100 μM) or bicuculline (10 μM) was bath applied, and NBQX (10 μM) and AP5 (50 μM) were bath applied to block glutamatergic signaling. Input resistance was monitored throughout the experiment. Any cells in which the resistance change was >20% were discarded. Data were analyzed using ClampFit (Molecular Devices) or Igor Pro 8 with Neuromatic (www.neuromatic. thinkrandom.com) and custom scripts. Drugs: Picrotoxin and biculline were purchased from Sigma. NBQX and AP5 were purchased from Hello Bio.

### Quantification, statistical analysis and reproducibility

All experiments were performed in at least two separate cohorts of mice and all treatment groups in each cohort were performed at least twice to avoid any unspecific day/condition effect. Treatments (i.e., choice of viral compounds within the dHPC and dmNAcSh) were randomly assigned to animals before testing. Statistical significance was taken as */#$p < 0.05$, **/##$p < 0.01$, ***/###$p < 0.001$, and ****/####$p < 0.0001$, as determined by a one-way ANOVA or a two-way repeated-measures ANOVA followed by a Sidak post hoc tests, two-tailed unpaired or paired $t$-test, one sample $t$-test compared to the hypothetical value (50%) or Mann-Whitney for unpaired values as appropriate. All data samples were tested for normality of distribution using Shapiro-Wilk test before being assigned to ANOVAs, two-tailed $t$-test, two-tailed Mann–Whitney for unpaired values, or two-tailed Wilcoxon matched-pairs signed rank analysis. All data are expressed as mean ± SEM. Sample size (n number) always refers to the value obtained from an individual animal when referring to behavioral experiments and anatomical slice analysis. For each experiment detailed statistical analysis and sample size (n number) are carefully reported in the figure legends. Statistical analyses were performed in GraphPad Prism 10.0 and SPSS. Representative images presented in Figs. 1B, 4B, 5B–C, and 7B are observed in all the animals included in the respective experiments. Validations of viral expression and fiber implants for animals in all the relevant experiments are included in Supplementary Fig. 6–9. Images in Fig. 2B, C are representative of all 5 mice used.

### Reporting summary

Further information on research design is available in the Nature Portfolio Reporting Summary linked to this article.

## Data availability

Authors can confirm that all relevant data are included in the paper and its Supplementary Information files. All data generated in this study are provided in the Supplementary Information/Source Data file. Further information and requests for resources and reagents should be directed to and will be fulfilled by the Lead Contact, Jose A. Morón (jmoron-concepcion @wustl.edu). Source data are provided with this paper.

## Code availability

Custom code used to analyze fiber photometry experiments in this paper is available at https://github.com/khairunisa-ibrahim/Ibrahim-Massaly-2023.

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

## Acknowledgements

We would like to thank Dr. Alexxai Kravitz for his critical help in experimental design and contribution to acquiring and analyzing local field potential data. In addition, we would like to thank Dr. Hannah Frye, Ms. Olayinka Idowu, Ms. Basma Daham, and all members of the Moron-Concepcion's lab for their help throughout the completion of the current study. Figures 2A and 7A were created with the help from BioRender.com. This work was supported by US National Institutes of Health (NIH) grant DA042499 (J.A.M.), DA045463 (J.A.M.), DA054900 (J.A.M.), DA055047 (N.M.), NARSAD Independent Investigator Award from the Brain and Behavior Research Foundation (J.A.M.), Philippe Foundation (N.M.), McDonnell Center for Cellular and Molecular Neurobiology (N.M.).

## Author contributions

Conceptualization: K.M.I., N.M., T.L.K. and J.A.M.; Methodology, K.M.I., N.M., A.W., R.W.G. and J.A.M.; Formal analysis: K.M.I., N.M., A.W., B.J.H. and J.A.M.; Investigation: K.M.I., N.M., H.J.Y., R.S., A.W., B.J.H., S.W., W.P., S.P., T.L., A.Z., A.P. and W.Y.; Writing—Original Draft: K.M.I., N.M. and J.A.M.; Writing—Review & Editing: K.M.I., N.M., A.W., B.J.H., R.W.G. and J.A.M.; Funding Acquisition: N.M., T.L.K., R.W.G. and J.A.M.; Resources: T.L.K., R.W.G. and J.A.M.; Supervision: K.M.I., N.M., R.W.G. and J.A.M.

## Competing interests

The authors declare no competing interests.
