## [Peer Review File · Nature Communications]

Dorsal Hippocampus To Nucleus Accumbens Projections Drive Reinforcement Via Activation of Accumbal Dynorphin NeuronsReviewers' comments:

Reviewer #1 (Remarks to the Author):

Massaly et al

This manuscript describes a series of experiments that attempt to establish whether stimulation of CA1 pyramidal neurons supports self-stimulation - what the authors call reinforcement - through its reported connections with the nucleus accumbens. While this is an interesting question there are issues with the task that the authors have used to address it. There are also a significant number of issues with the demonstration that the effects of CA1 manipulations are specific to projections to the accumbens specifically. This arises from the authors' unconvincing mapping of (i) the projection from CA1 to accumbens relative to better described projections to other structures immediately medial and dorsal to the accumbens; (ii) the separation of any effects of stimulation on accumbens to those of other structures, notably the diagonal band and medial septal areas - both of which have significant connections with the CA1 - but also other areas (iii) whether recordings made from accumbens and inactivation of projections from CA1 in the accumbens are localised or are a product of activation (or silencing) in other areas. This includes the final assessment in ENK-Cre and DYN-Cre animals; cells expressing these peptides are located outside the accumbens, again in adjacent areas. Finally, the statistics are inadequate or are not fully reported. Generally, only p values are reported; ANOVAs matching the various experimental designs are not described and interactions are inferred and not tested or reported despite the fact that a reliable interaction is necessary to support the conclusions from most of these experiments. Specifically:

1. The task: The authors claim that the effects they are examining reflect reinforcement. However, they pair a light above a nose poke hole with stimulation of CA1 which provides the opportunity for the animals to learn a Pavlovian light-CA1 stimulation association. The question is: are the animals poking the hole for the stimulation of CA1 as the authors argue or because that stimulation enhances saliency of the light? No attempt is made to assess this and related questions. As a consequence, it is unclear how the nose pokes should be interpreted. The authors rely heavily on the place preference work of Trouche et al as rationale for this study. The latter study did not find evidence of reward processes and instead interpreted their findings in terms of changes in the saliency of spatial stimuli. The same conclusion could be drawn here.
2. I expected to see experiments demonstrating that the effects the authors attribute to the CA1-accumbens projection are specific to that projection and not due to the influence of CA1 on other adjacent areas. This requires experiments comparing the effects of similar manipulations of those adjacent areas with effects in the accumbens, but this was not done.
3. Throughout the study the mapping of the tracing is poor and inadequate. The authors focus on accumbens and do not look more broadly at CA1 projections. But we can see, even in the very limited illustrations provided in the figures, that there is considerable labelling immediately medial and dorsal to the accumbens shell and adjacent to the core; perhaps in medial septum/diagonal band areas (its not clear from these figures). There is likely to be much more of this labelling. The figures are simply not adequate to allow the reader to discern what structures are labelled by YFP or in the cleared brain. Reconstruction of the tracing needs to be much more thorough. This is true of all of the studies assessing CA1 projections but also those in which ChR2 is stimulated in the accumbens region and including the final studies manipulating ENK and DYN expressing neurons because these are not only found in the accumbens but throughout the ventral forebrain.
4. The statistics that have been reported are very limited and inadequate. The authors claim they have conducted ANOVA but report no F statistics, no main effects or interactions are reported; which should be expected when comparing the differences between experimental and control groups. For example, I don't see any comparison between the comparable periods in the light stim in left CPP in ChR2 vs YFP

controls in fig G and H. In fact, visually these groups don't look much different. There also appears to be some transfer after reversal in the YFP controls suggesting that the light accompanying ChR2 stimulation is alone exerting some control over performance. Furthermore, in Fig 3B, the authors state "The persistence of reinforcing behavior stopped when the active nose-poke was not associated with light delivery": but it doesn't stop and still looks to be well elevated to the control. But no direct statistical comparison is reported comparing the difference with light on and light off. This lack of comparison of the differences between groups extends to most of the behavioral data and the GCaMP data as well.

5. For the pharmacology experiments I couldn't find details of the drugs, their concentrations or volumes. How far did the drug infusions spread? How off target could their effects have been?

Reviewer #2 (Remarks to the Author):

In this manuscript, Massaly, Ibrahim and colleagues explore whether activation of the dorsal CA1 (dCA1) to nucleus accumbens (NAc) projection is sufficient to drive reinforcement. This question is of great interest, as the hippocampal-accumbens circuit is thought to be critical for spatially-guided reward learning, yet a direct projection between dorsal hippocampus and NAc has only recently been established and has not been thoroughly characterized. The authors clearly demonstrate that mice will initiate actions to receive optogenetic stimulation of dCA1 and of dCA1 axon terminals. Further, they show that dCA1 stimulation evokes increased activity in the nucleus accumbens (NAc) as measured by LFP and fiber photometry, and that self-stimulation of dCA1 is blunted by glutamatergic blockers in the NAc and inhibition of dynorphin-expressing NAc medium spiny neurons. The behavioral experiments are well-controlled and support evidence in the literature that the dCA1-to-NAc circuit is engaged in reward learning. However, there is limited evidence in the manuscript that the effects seen are driven by the direct dCA1-to-NAc projection as opposed to an indirect projection or by dCA1 projections to other structures, although I believe that a dCA1-to-NAc direct projection exists. I think these caveats could be addressed with additional analyses and experiments before the study is published. I outline these below, in addition to some minor comments.

Major comments:

1. The strongest evidence of a direct projection is from the dCA1 terminal stimulation in NAc (Fig. 1E-H), but there are several caveats even with this experiment:
 - a. The authors only do this for nosepoke self-stimulation. NAc terminal stimulation should be repeated for the conditioned place preference paradigm, to provide evidence that the self-stimulation of the direct projection can be spatially associated.
 - b. Optogenetic axon terminal stimulation can drive antidromic spikes, which could excite other regions that the stimulated neurons project to (see Kim et al. 2017, 10.1038/nrn.2017.15). Because of this, even axon stimulation in the NAc could induce indirect effects by driving spikes in dCA1 that either propagate through the dorsal hippocampal circuit or activate downstream structures directly through branching axons. To truly confirm that only the dCA1-to-NAc projection is responsible, while optogenetically stimulating dCA1 axons in NAc, the authors would need to infuse glutamatergic antagonists into NAc, or infuse muscimol or glutamatergic antagonists into dCA1 to block propagation of activity from dCA1 cell bodies.
 - i. Notably, the reduction in self-stimulation that the authors report with CNQX/AP5 infusion into NAc is not complete – pokes still seem higher than at the inactive ports, though the statistics for this comparison are not reported (at least not in the main figure). This could be because dCA1 cell body stimulation is activating other downstream targets that can drive reinforcement, even if NAc glutamatergic transmission is blocked.

c. The authors should estimate and quantify the spread of light from the optic fiber over NAc, as well as the spread of their drug infusion. dCA1 exhibits strong projections to neighboring structures such as the lateral septum, where reward-related activity has also been observed (Wirtshafter & Wilson, 2020, 10.7554/eLife.55252). Could the effects of stimulation in NAc be driven by accidental activation of the lateral septum projection?

i. This possibility (and the other caveats mentioned) could be also be addressed by retrogradely labeling dCA1 cells that project to the NAc with a retro-AAV-Cre virus injected in NAc, and then injecting Cre-dependent opsin in dCA1 to stimulate projection-specific cell bodies.

2. The direct projection would be most convincing by demonstrating a monosynaptic connection from dCA1 axons to dynorphin-expressing MSNs with slice physiology.

To be clear, I do not think the authors need to do every single one of the additional experiments above, but I would need to see at least 2 of them to be convinced that the results are due to the direct dCA1-to-NAc projection if that is a claim the authors want to make.

3. The authors interpret their results as demonstrating “reward-seeking behavior” in response to dCA1 stimulation, but it can be argued that intra-cranial self-stimulation is not really reward-seeking behavior (Millard et al. 2022, <https://doi.org/10.1101/2022.08.11.503670>).

a. It would be very interesting and informative to know if the dynorphin-mediated circuit that the authors identify is involved in behavior to collect natural rewards. If they pair one side of the CPP chamber with sucrose and the other with water as in Trouche et al. 2019, and inhibit the dynorphin-expressing neurons, is CPP behavior impaired? However, this type of experiment is not necessarily critical if the authors decide instead to soften their language around reward-seeking.

4. The fiber photometry and LFP data both seem to show relatively long latencies to evoked activity after the dCA1 stimulation, suggesting polysynaptic activity, but this is never quantified. Moreover, in Fig 2E and 2H, it looks like stim-evoked activity is only present on a subset of probe channels and trials, respectively.

a. The authors should quantify the latency to evoked stimulation, and ideally compare it to the latency from a known, robust monosynaptic connection assessed with these techniques.

b. For the LFP data, could the authors characterize where those channels were anatomically located, and whether their location corresponds to an area of higher density of dCA1 terminals? If the probe was labeled by a lipophilic dye, they could at least get a rough estimate based on where the active channels are located on the probe.

c. For the photometry data, what was happening on the trials with more evoked activity? Does the level of evoked calcium transients or evoked LFP vary with the animal’s spontaneous locomotor patterns or other behaviors?

Minor comments:

5. Inhibition of dynorphin-expressing neurons reduces self-stimulation, but not to the levels of the inactive ports.

a. It may be too strong on line 199 to say that dynorphin neurons are “necessary” for the response – they are certainly involved, but something else is contributing as well.

b. Had the animals already learned the port associations before the CNO-inhibition sessions? The methods say that CNO started “after training sessions”, but I’m not sure what that means. If the dynorphin neurons were silenced from the very first self-stimulation session without any training with the nose pokes, would self-stimulation be more completely abolished? In other words, is there a learned association of stimulation with the nose pokes that is independent of dynorphin neurons?

6. In Fig. 1 G and H, when the left side of the box is illuminated, the groups don’t seem to have much difference – the between-group difference is only obvious when stimulation switches to the right side.

a. Are there significant differences between the groups when compared within side?

7. I couldn't find a couple relevant methodological details:

- a. What were the tested factors for the 1 and 2-way ANOVAs outside of Fig. 3B?
- b. For the fixed ratio self-stim, how many times did the animals have to nose poke to get the stimulation?

Reviewer #3 (Remarks to the Author):

Review of Massaly et. al.,

Massaly et. al., describe a set of experiments that show mice will perform behaviors that result in optogenetic stimulation of CAMKII expressing neurons in the dorsal CA1 region of the hippocampus. They use nosepoke as a metric for operant responding to earn the stimulation, and place location in a real time place test as a metric to show mice will spend more time in an area where this stimulation is present. They further use the power of optogenetics to stimulate specific terminals of these neurons projecting to the nucleus accumbens, which is known to be a hub of reward/reinforcement and motivation processing. They argue this indicates that reinforcing properties of dCA1 stimulation is at least partially mediated by the release of neurotransmitters from these neurons in the NAc. They use microinfusions of glutamate receptor antagonists into the NAc to show that this manipulation reduces operant responding for dCA1 opto stimulation. They lastly show that operant responding for dCA1 opto-stim is reduced when one subset of NAc neurons (dynorphin producing) are presumably hyperpolarized using DREADD techniques.

Overall, the experiments conducted in this manuscript are laid out in a manner that is fairly easy to follow. My main concern is that these experiments are very much "tip of the iceberg" at this point, and do not offer much to the field beyond optogenetic dissection of a pathway (already known to be involved in motivation, affective, and learning/memory processes) to show mice will modestly perform behavior to have this pathway artificially excited. It appears that the authors are placing too much emphasis on intracranial self-stimulation as a causal metric of behaviorally relevant reward processing, and in this manuscript have not done the experiments to show that this artificial excitation is meaningful for natural reward related behaviors. In other words, intracranial self-stimulation simply shows that somehow, somehow, the population being stimulated gives some sort of signal, somewhere in the brain, that leads the animal to repeat the behavior that lead to the stimulation. This is not evidence that those neurons mediate *anything* specific to reward. They are not causal in implicating the neurons being stimulated in "reward". I am not saying that these studies are not important, they are, and as illustrated by the seminal work of Olds and colleagues, they are the foundation of systems-neuroscience understanding of motivated behaviors; but many regions support ICSS or ICSSA, and this alone does not tell us those neurons do anything in particular during natural reward processing. Therefore, my main concern is that the authors are greatly over-reaching in their conclusions about the role of dCA1->NAc in reinforcement or reward related behaviors. I will lay out specific concerns that should be addressed to actually test the role of these neurons in reinforcement/reward, and add the type of advancement to the field that, it is my understanding, this journal aims for.

Major concerns:

- 1) Define and be consistent throughout the manuscript what optoICSS means as a readout. They primarily use reinforcement but then also use statements like "drive reward seeking" or "trigger light self-stimulation" which they've not shown. I believe the authors are incorrect and stating that their experiments show these neurons "drive reward seeking" (eg. Line 206, the main summary statement). All ICSS can show is that the stimulation reinforces the behavior directly preceding the stimulation. To make this statement, they must show causally that activation of these neurons increases behaviors directly related to obtaining a goal. The best they can do with the present data is say those neurons may be implicated in reward seeking.
- 2) The optoICSS rates are very low. These are several orders of magnitude lower than what is observed for optoICSS of regions that were then directly implicated in reward/reinforcement processes

(e.g. 1000s of lever presses at FR1 in 30 minutes). Also, nosepoke rates are typically higher than lever press rates given same reward, making the low rates observed in this study even less convincing as evidence of "self-stimulation". So, the low self-stimulation rates here is a glaring discrepancy when considered with past studies doing similar experiments. Did the authors test whether the mice will perform the oICSS without the cue-light? The rates reported here suggest that the mice are not particularly engaged in the self-stimulation itself, so I wonder if the stimulation is increasing the saliency of the change in (i.e. turn off) the active cue light. The rates in this report are more akin to how rodents respond to visual stimuli alone. Given the importance of the dHPC in governing environmental saliency, the above is a possible alternate explanation that should be addressed. Other information about the timeseries of the optoICSS can shed light on whether these neurons are involved in reward processing (and therefore reinforcement). How are the mice responding across a session? Do they press more in the beginning? Do they run around or sniff after a stimulation? Classically, rewarding brain stimulation (electrical or optogenetic) will produce sniffing or forward locomotion, and will typically occur in bursts (or even continuously) across a session. While, yes, the active poke rates are greater than the inactive, the ≤ 1 poke per min is not very convincing towards deeming this population of neurons as supporting self-stimulation. The authors at the very least need to discuss this and attempt to reconcile. But even with a strong justification for why this rate is so low yet still indicates reinforcement or drives reward, I doubt many in the field will consider this strong evidence for their overall claim about optoICSS.

3) Similarly, the effects in RTPT are very modest for a "rewarding" manipulation. There seems to be lag in the effect, in that the first 10 minutes of the task, the ChR2 group shows no strong place preference, but it gets more apparent after the 10min mark, and then stronger and stronger as the session goes on. Typically, the affects seen from rewarding brain stimulation are rapidly acquired in this paradigm. It does not take rodents long to form the association of a "room", or even spot in a chamber, with the rewarding stimulation. The significant differences in Figure 1F appear to be driven by 6 or so mice, otherwise the groups appear similar (are there histological observations for why out of 18 mice, only a handful have % time in stim side greater than control?). The control group (Fig 1 H) show similar trends (with almost half as many mice).

With the above point on optoICSS rates, it is hard to reconcile that this manipulation is rewarding – the effects are just so much weaker at these two common tasks (ICSS and RTTP) than seen with similar manipulations of validated/established "reward" systems.

4) There are important details related to the stimulation parameters that are missing. What intensity was the light for operant tasks? How or why did they decide on the parameters they used? Were there any other "doses" of light they tried, i.e. different train lengths, frequency, etc.? Do dCA1 neurons ever respond in this fashion (i.e 20hz for 2 seconds) during natural behaviors?

5) Infusion of AMPA/NMDA antagonists into the accumbens essentially takes off-line this region that is crucial for motivated behavior (MSNs don't do much without glu input). The authors must show that the reduction in dCA1 ICSS is specific to this manipulation and the manipulation does not, rather, simply disrupt all reward seeking behaviors (or locomotion). It is also unclear how this experiment was conducted. The methods say this was a bilateral cannula implant. Why? The optogenetic manipulations are unilateral, so why infuse into each NAc rather than just ipsilateral to the opto-stim. If indeed, both NAc receive glutamate antagonists, I believe many/most motivated behaviors would be affected, which would undermine what they are trying to show – that the glu release in the NAc from the opto stim is implicated in the reinforcing effects of that stimulation. I recommend designing an experiment that shows this microinfusion specifically disrupts your optoICSS, and not general operant responding.

6) The DREADD experiments suffer from the same design issue as cannula infusions above.

7) They have not demonstrated that the optogenetics or chemogenetics do what they are supposed to do. Some sort of validation of these methods is necessary. Do the dCA1 neurons follow a 2 second 20hz train? Do the Gi DREADDs take Dyn and Enk "offline" equally?

8) The discussion lacks any synthesis of how this pathway fits into canonical reward systems. How these neurons work in this neural module to affect motivation would help us understand why stimulation of these neurons modestly reinforces behavior.

9) The authors clearly have the capability to investigate how the neurons they study here, in particular the dCA1->NAc, contribute to natural reward seeking behaviors. They give some examples in the

introduction and discussion that suggest they do, but we already know dCA1 and dCA1->NAc play a role in memory processes related to reward seeking behaviors. As stated before, the ICSS here is artificial and simply suggests somehow these neurons are related to reward seeking (again, we know this), but complimentary experiments to monitor and/or disrupt the normal activity of these neurons during ethologically relevant goal-directed behavior is needed to make this story more than an incremental advancement.

10) Related to the functional connectivity assays. There are no analysis of the latencies regarding dCA1 stimulation onset and NAc LFP or GCaMP dynamics. This is crucial to interpret the extent that this stimulation acts to modulate NAc directly or indirectly. I am also confused by the summary analysis of the LFP. It appears the authors are using absolute value of the LFP before and after stimulation. Figure 2E seems to show many channels recording a decrease in LFP, and seemingly less show increase, while many show no change. Yet, in line 140 they say evoked "activation"?

I cannot seem to find what Pre-Min and Post-Max are referring to in Figure 2F; what is pre and what is post, and compare a minimum and maximum from these different times? They also seem to have recorded NAc LFPs both ipsilateral and contralateral to the dCA1 stimulation, but this is not mentioned or discussed.

It is unclear how fiber photometry of GCaMP "gains both anatomical and temporal resolution of the dynamics of NAc neuronal activity..." (Line 143) when microelectrode recordings have better resolution of both. It also would seem that the results related to this in the main text (FP during dCA1 optoICSS) are confounded by the coincidence of the behavioral task when interpreting whether the opto-stimulation is affecting the FP signal (as opposed to the behavior) – the suppl. data Fig1E-H better support this interpretation.

11) Line 153 – how is this a selective accumbal response? They've not compared signals in surrounding regions. This brings up another important missing control – there are no site controls in this study. They've not investigated other dCA1 projections, nor placed optic fibers in areas adjacent to injection sites to identify spatial specificity. One of the main claims is that while we know electrical excitation of dHPC is reinforcing, they argue this study identifies dCA1 CAMKII, and in particular those dCA1 CAMKII projecting to NAc as mediating that reward – but these neurons project elsewhere and they've not shown these other pathways don't support ICSS.

12) I am not sure what the brain clearing studies are adding here. Firstly, both tracers jRCECO1a and Chr2 are not specific to terminals, so analysis is limited to regions where fibers are present, could be fibers of passage, no specificity to functional connectivity. But, they are not really doing anything with this data set regardless. In fact, this seems like a missed opportunity to identify some projection "site-controls" to test the above point. Likewise, using similar strategies they could identify whether dCA1->NAc projecting neurons send collaterals elsewhere. Optogenetic stimulation of terminals can send antidromic action potentials back to the cell body, and if the neuron sends collaterals, these alternate projections could mediate reinforcing properties. This later concern would be alleviated if the cannula experiments were performed appropriately.

Minor concerns:

1) dHPC and dCA1 seem to be used interchangeably in the manuscript. Please be consistent or make it clear why one used vs another.

2) I would use a term somewhat similar to optoICSS or oICSS to describe operant responding for opto stimulation, rather than "light self stimulation", which is not what the animals are doing (it is more light self-administration if anything).

3) Sometimes they refer to control group as GFP mice, other times control littermates.

4) Try to avoid superlative/embellishing statements, e.g. Line 119 "In sharp contrast". Most of the effects here are not that pronounced. Also, Line 44, I think "cutting-edge" is a bit of a stretch at this point.

5) Line 130, the virus must be incorrectly written, I think Chr2 and CaMKII should be switched.

6) Line 78, what is meant by "recruitment".

7) I don't see a definition of CaMKII before its use.

8) What was the acquisition rate of LFPs and how were they processed?

9) Line 110, I think they mean CaMKII+ neuron activation, not activity of CaMKII. Again, here in the sentence another example of overreach statement in saying these neurons are necessary 'driver' for seeking behavior.

10) Line 353, should this be "HISTOLOGY"?

11) Line 422, this is an important method reporting but is hard to understand. Please revise and flesh it out. I.e. each behavioral experiment was performed in at least two separate cohorts of animals?

12) Figure fonts and some of the cartoons are very small and hard to make out. Also it is very difficult to glean what the groups or comparisons are by looking at the figures. These information are often buried at the end of the legends when looking at the p-value descriptions.

13) I suggest using Active and Inactive rather than Incorrect (eg. Figure 1B-D). Incorrect implies a choice or decision, and convention for operant behaviors with two instrumental stimuli is active and inactive.

We thank the Reviewers and Editor for their helpful and constructive comments. These suggestions greatly assisted us in strengthening the claims and overall dataset presented in the manuscript. The results included in this revised version add a significant number of new experiments and a more thorough discussion to address each of the comments raised by the three Reviewers and the Editor (reviewed point-by-point below). We do hope you will now agree that, in its current state, our manuscript fully satisfies all the reviewers' concerns.

We conducted a significant number of additional experiments and made extensive improvements to address all of the reviewers' comments and concerns. As a summary of the main additional studies conducted in this revision, we:

1. Provide further evidence for the **sufficiency of monosynaptic projections from the dorsal hippocampus (dHPC) to the nucleus accumbens shell (NAcSh) to drive reinforcing behaviors** – using both synaptic terminal stimulation in the NAcSh and a dual virus strategy with Cre-recombinase-encoding retroAAV virus in the NAcSh in conjunction with Cre-dependent AAV vectors in the dHPC.
2. **Provide a more comprehensive mapping of the dHPC-NAcSh input.** Updated images with higher resolution **for terminal expression in the NAcSh** is now provided. We are also including **images of dHPC cell bodies projecting to the NAcSh** using the retroviral approach used in #1 above.
3. **Further provide functional evidence of monosynaptic inputs from the dHPC to both enkephalin and dynorphin containing medium spiny neurons** within the NAcSh – using channelrhodopsin assisted connectivity **in slice physiology**. In addition, this data is complemented with retroviral tracing mapping approaches.
4. Demonstrate the **involvement of dHPC-NAcSh projection in sucrose-seeking, as a model of natural reward seeking behavior** as opposed to the sole optogenetic self-stimulation described in our previous version of the manuscript.
5. Confirm that the **self-stimulation behavior we observe is not driven by enhanced saliency of the light but rather selective activation of the dHPC→NAcSh pathway.**
6. **Provide further evidence for the robustness of the self-stimulation procedure.** As shown in the additional data provided in this revised manuscript, animals self stimulate at rates previously reported when post-stimulation time-out periods and caps on the maximum number of stimulations per session are removed.
7. **Provide full description of the statistical analyses conducted in the figure legends** (these details were included as an excel file attached in the original version).

These added experiments bolster our overall conclusions and clearly demonstrate the dHPC→NAc as a critical pathway involved in driving reinforcing behavior necessary to maintain reward seeking behavior. Together, this expanded dataset and improvements not only fully address the reviewers' concerns but also, yield a manuscript highly suitable for publication in Nature Communications.

All the modifications made to the manuscript and added experiments have been highlighted in blue throughout the text. See below a point-by-point response to each of the concerns/comments raised by the Reviewers:

Reviewer #1

The authors claim that the effects they are examining reflect reinforcement. However, they pair a light above a nose poke hole with stimulation of CA1 which provides the opportunity for the animals to learn a Pavlovian light-CA1 stimulation association. The question is: are the animals poking the hole for the stimulation of CA1 as the authors argue or because that stimulation enhances saliency of the light? No attempt is made to assess this and

related questions. As a consequence, it is unclear how the nose pokes should be interpreted. The authors rely heavily on the place preference work of Trouche et al as rationale for this study. The latter study did not find evidence of reward processes and instead interpreted their findings in terms of changes in the saliency of spatial stimuli. The same conclusion could be drawn here.

In the previous version of the manuscript, we dissociated the conditioned stimulus (CS; cue light) from the unconditioned stimulus (US; laser stimulation) by presenting the cue light in the absence of laser stimulation. Consequently, mice stopped interacting with the active nose inset, suggesting responding was driven specifically by dHPC stimulation. To bolster this claim, the revised manuscript provides additional experiments in which the CS was omitted in the presence of the US and found that self-stimulation was maintained (see Supplementary **Figure 1 A-D**). As such, this evidence indicates that dHPC self-stimulation is solely contingent on laser stimulation rather than enhancing saliency of the paired conditioned stimuli. These results combined with the findings obtained using Real Time Place Test demonstrate that dHPC→NAc is sufficient to drive reinforcing behavior, independent of conditioned cues.

I expected to see experiments demonstrating that the effects the authors attribute to the CA1-accumbens projection are specific to that projection and not due to the influence of CA1 on other adjacent areas. This requires experiments comparing the effects of similar manipulations of those adjacent areas with effects in the accumbens, but this was not done.

This is an excellent point that was also brought by Reviewer #2. To address this, we used a dual viral targeting strategy with a Cre-recombinase-encoding retroAAV virus in the NAcSh in conjunction with Cre-dependent AAV vectors in the dHPC. This allowed us to selectively engage direct projections from the dHPC to the NAcSh during the light self-stimulation studies. We show here that, similar to the stimulation of global CaMKII+ neuronal populations in the dHPC, circuit-specific activation of the dHPC→NAcSh pathway is sufficient to drive reinforcing behavior (please see **Figure 3 A-D**).

Throughout the study the mapping of the tracing is poor and inadequate. The authors focus on accumbens and do not look more broadly at CA1 projections. But we can see, even in the very limited illustrations provided in the figures, that there is considerable labelling immediately medial and dorsal to the accumbens shell and adjacent to the core; perhaps in medial septum/diagonal band areas (its not clear from these figures). There is likely to be much more of this labelling. The figures are simply not adequate to allow the reader to discern what structures are labelled by YFP or in the cleared brain. Reconstruction of the tracing needs to be much more thorough. This is true of all of the studies assessing CA1 projections but also those in which Chr2 is stimulated in the accumbens region and including the final studies manipulating ENK and DYN expressing neurons because these are not only found in the accumbens but throughout the ventral forebrain.

The circuit from the dHPC→NAc is indeed discrete. To fully characterize the pathway and improve its visualization, we conducted additional viral tracing studies and included additional illustrations for the dHPC projections to the NAcSh using dual virus approaches that permit visualization of CA1 cell bodies projecting to the NAcSh. As suggested, we also provide higher magnification images depicting a dorsal vs ventral expression of terminal fibers within the NAcSh. These additional histological representations are shown throughout Figures 1-4. Further, expression of viruses and fiber placements are fully reported in the supplementary material on the regions of interest.

The statistics that have been reported are very limited and inadequate. The authors claim they have conducted ANOVA but report no F statistics, no main effects or interactions are reported; which should be expected when comparing the differences between experimental and control groups. For example, I don't see any comparison between the comparable periods in the light stim in left CPP in Chr2 vs YFP controls in fig G and H. In fact, visually these groups don't look much different. There also appears to be some transfer after reversal in the YFP controls suggesting that the light accompanying Chr2 stimulation is alone exerting some control over performance. Furthermore, in Fig 3B, the authors state "The persistence of reinforcing behavior stopped when the active nose-poke was not associated with light delivery": but it doesn't stop and still looks to be well elevated to the control. But no direct statistical comparison is reported comparing the difference with light on

and light off. This lack of comparison of the differences between groups extends to most of the behavioral data and the GCaMP data as well.

We truly apologize for this oversight. Although a completed and detailed report of all the statistics ran for each experiment was attached as an excel file during the initial submission, we have revised this version of the manuscript to also include detailed statistics in each of the figure legends.

For the pharmacology experiments I couldn't find details of the drugs, their concentrations or volumes. How far did the drug infusions spread? How off target could their effects have been?

Details regarding drug concentrations and volumes have been added to the methods section of the manuscript. The striatum contains densely packed cells with a compactness ratio of ~0.7, indicating that a bolus injection could be expected to be restricted to about 70% of the same volume of striatal tissue (Ramadi et al., 2020). Based on our injected volume of 250nl, we expect that the pharmacological effects were restricted to within 0.25mm of the injector site.

Reviewer #2

The strongest evidence of a direct projection is from the dCA1 terminal stimulation in NAc (Fig. 1E-H), but there are several caveats even with this experiment:

The authors only do this for nosepoke self-stimulation. NAc terminal stimulation should be repeated for the conditioned place preference paradigm, to provide evidence that the self-stimulation of the direct projection can be spatially associated.

While we cannot rule out that the reinforcing effect of dHPC→NAcSh stimulation is spatially associated, the primary focus of this manuscript aimed to characterize instrumental behaviors since the involvement of this circuit using a CPP paradigm has been characterized to a greater extent elsewhere (Trouche et al.). Moreover, it should be noted that long and synchronous stimulation of CaMKII+ neuronal populations in the dHPC has been shown to trigger the development of epileptic seizures (Campbell et al., 1978, Goddard et al., 1969). In the current manuscript, we overcome this potential caveat with the use of a self-stimulation procedure that allows animals to titrate the number of stimulations they obtain. In addition, **Figure 1A-E** clearly demonstrate that despite the lack of visual cues, animals are able to develop a real time place preference. Hence, demonstrating that “the self-stimulation can be spatially associated”.

Optogenetic axon terminal stimulation can drive antidromic spikes, which could excite other regions that the stimulated neurons project to (see Kim et al. 2017, 10.1038/nrn.2017.15). Because of this, even axon stimulation in the NAc could induce indirect effects by driving spikes in dCA1 that either propagate through the dorsal hippocampal circuit or activate downstream structures directly through branching axons. To truly confirm that only the dCA1-to-NAc projection is responsible, while optogenetically stimulating dCA1 axons in NAc, the authors would need to infuse glutamatergic antagonists into NAc, or infuse muscimol or glutamatergic antagonists into dCA1 to block propagation of activity from dCA1 cell bodies. Notably, the reduction in self-stimulation that the authors report with CNQX/AP5 infusion into NAc is not complete – pokes still seem higher than at the inactive ports, though the statistics for this comparison are not reported (at least not in the main figure). This could be because dCA1 cell body stimulation is activating other downstream targets that can drive reinforcement, even if NAc glutamatergic transmission is blocked. The authors should estimate and quantify the spread of light from the optic fiber over NAc, as well as the spread of their drug infusion. dCA1 exhibits strong projections to neighboring structures such as the lateral septum, where reward-related activity has also been observed (Wirtshafter & Wilson, 2020, 10.7554/eLife.55252). Could the effects of stimulation in NAc be driven by accidental activation of the lateral septum projection?

This possibility (and the other caveats mentioned) could be also be addressed by retrogradely labeling dCA1 cells that project to the NAc with a retro-AAV-Cre virus injected in NAc, and then injecting Cre-dependent opsin in dCA1 to stimulate projection-specific cell bodies.

We thank the reviewer for this comment. To answer this comment, we employed the recommended dual virus strategy with Cre-recombinase-encoding retroAAV virus in the NAcSh in conjunction with Cre-dependent AAV vectors in the dHPC. This allowed us to selectively engage direct projections from the

dHPC to the NAcSh during light self-stimulation procedures. We show here that, similarly to the stimulation of the overall CaMKII+ neuronal population in the dHPC, activation of this selective dHPC→NAcSh pathway is sufficient to drive reinforcing behavior (please see **Figure 3 A-D**).

The direct projection would be most convincing by demonstrating a monosynaptic connection from dCA1 axons to dynorphin-expressing MSNs with slice physiology.

We agree with the reviewer that a clear demonstration of monosynaptic connectivity was lacking in the original version of our manuscript. While a previous report has already described a direct monosynaptic connection of dHPC→NAcSh PV+ interneurons and MSNs, we used in this revised version channelrhodopsin assisted connectivity combined with ex vivo slice physiology and demonstrate that dHPC CaMKII+ neurons monosynaptically synapse onto both enkephalin+ and dynorphin+ neurons in the NAcSh (please see **Figure 6**). Furthermore, while dHPC CaMKII+ inputs to enkephalin-containing MSNs could be found throughout the NAcSh, dynorphin-containing MSNs that respond to dHPC CaMKII+ inputs were only found in the dorsal region of the NAcSh (**Figure 6**). Those results are in accordance with a previous reports demonstrating that MSNs can either exhibit responses or not to direct dHPC CaMKII+ neurons.

The authors interpret their results as demonstrating “reward-seeking behavior” in response to dCA1 stimulation, but it can be argued that intra-cranial self-stimulation is not really reward-seeking behavior (Millard et al. 2022, <https://doi.org/10.1101/2022.08.11.503670>).

It would be very interesting and informative to know if the dynorphin-mediated circuit that the authors identify is involved in behavior to collect natural rewards. If they pair one side of the CPP chamber with sucrose and the other with water as in Trouche et al. 2019, and inhibit the dynorphin-expressing neurons, is CPP behavior impaired? However, this type of experiment is not necessarily critical if the authors decide instead to soften their language around reward-seeking.

We thank the reviewer for this insightful comment. In the revised manuscript, rather than using sucrose CPP, we used sucrose self-administration combined with an inhibitory chemogenetic approach to silence the activity of dHPC→NAcSh pathway during sucrose seeking (see **Figure 4**). With this experiment we demonstrate that silencing dHPC→NAcSh significantly decreased goal-directed behavior for sucrose self-administration in the same instrumental operant task used in our self-stimulation studies. This result further strengthens the importance of the dHPC→NAcSh to drive reinforcement and reward seeking.

The fiber photometry and LFP data both seem to show relatively long latencies to evoked activity after the dCA1 stimulation, suggesting polysynaptic activity, but this is never quantified. Moreover, in Fig 2E and 2H, it looks like stim-evoked activity is only present on a subset of probe channels and trials, respectively.

The authors should quantify the latency to evoked stimulation, and ideally compare it to the latency from a known, robust monosynaptic connection assessed with these techniques.

We thank the reviewer for this comment and we now provide quantification of the latency of the evoked stimulation in **Figure 2H** of the revised manuscript.

For the photometry data, what was happening on the trials with more evoked activity? Does the level of evoked calcium transients or evoked LFP vary with the animal’s spontaneous locomotor patterns or other behaviors?

Animals were habituated to the arena before recordings were initiated to account for novelty-induced locomotion. As for fiber photometry experiments (see **Supplementary Figure 2**) responses in the NAc were not associated with locomotor activity, presentation of cue light, or light stimulation when ChR2 was not expressed in hippocampal tissue.

Reviewer#3

Major concerns:

Define and be consistent throughout the manuscript what optoICSS means as a readout. They primarily use reinforcement but then also use statements like “drive reward seeking” or “trigger light self-stimulation” which

they've not shown. I believe the authors are incorrect and stating that their experiments show these neurons "drive reward seeking" (eg. Line 206, the main summary statement). All ICSS can show is that the stimulation reinforces the behavior directly preceding the stimulation. To make this statement, they must show causally that activation of these neurons increases behaviors directly related to obtaining a goal. The best they can do with the present data is say those neurons may be implicated in reward seeking.

We thank the reviewer for this comment that was also brought by reviewer #2. In the revised manuscript, we used sucrose self-administration combined with an inhibitory chemogenetic approach to silence the activity of dHPC→NAcSh pathway and assess its contribution to sucrose seeking behavior (see **Figure 4**). With this experiment we demonstrate that silencing dHPC→NAcSh significantly decreases goal directed behavior for sucrose self-administration. This result further strengthens the importance of the dHPC→NAcSh in mediating reinforcement and reward seeking in addition to its already defined role in contextual/cue memory. We also went carefully through the manuscript to make sure that the appropriate terms were used.

The optoICSS rates are very low. These are several orders of magnitude lower than what is observed for optoICSS of regions that were then directly implicated in reward/reinforcement processes (e.g. 1000s of lever presses at FR1 in 30 minutes). Also, nosepoke rates are typically higher than lever press rates given same reward, making the low rates observed in this study even less convincing as evidence of "self-stimulation". So, the low self-stimulation rates here is a glaring discrepancy when considered with past studies doing similar experiments. Did the authors test whether the mice will perform the oICSS without the cue-light? The rates reported here suggest that the mice are not particularly engaged in the self-stimulation itself, so I wonder if the stimulation is increasing the saliency of the change in (i.e. turn off) the active cue light. The rates in this report are more akin to how rodents respond to visual stimuli alone. Given the importance of the dHPC in governing environmental saliency, the above is a possible alternate explanation that should be addressed. Other information about the timeseries of the optoICSS can shed light on whether these neurons are involved in reward processing (and therefore reinforcement). How are the mice responding across a session? Do they press more in the beginning? Do they run around or sniff after a stimulation? Classically, rewarding brain stimulation (electrical or optogenetic) will produce sniffing or forward locomotion, and will typically occur in bursts (or even continuously) across a session. While, yes, the active poke rates are greater than the inactive, the ≤ 1 poke per min is not very convincing towards deeming this population of neurons as supporting self-stimulation. The authors at the very least need to discuss this and attempt to reconcile. But even with a strong justification for why this rate is so low yet still indicates reinforcement or drives reward, I doubt many in the field will consider this strong evidence for their overall claim about optoICSS.

We performed several additional experiments to address these comments/concerns:

1. Because long and synchronous stimulation of CaMKII+ neuronal population in the dHPC has been shown to trigger the development of epileptic seizures, we first applied restrictive parameters, including a 10sec time out after each dHPC self-stimulation, and a maximum of 100 stimulations per session after which the overall session will end. To answer the first part of the reviewer's comment we removed those restrictive parameters (**Figure 1 K-L**). In those graphs we can see that removing the 10sec time out period increases the number of stimulation with a session (yellow/green bar and single plots), and removing both time outs and 100 maximum self-stimulation per session, further increase the number of active interactions with the nose poke inset to levels reported in other publications assessing reinforcement and using optogenetics as an approach to interrogate a selective circuits. We also depicted in **Figure 1H and 1L**, the continuous distribution of self-stimulation across a session confirming that the animals do seek for dHPC stimulation across the session.
2. To address the comment regarding the saliency of the change, we trained animals to self-stimulate and removed the cue-light indicating the reward availability (please see **Suppl Fig 1**). In this case, and in opposition to when we shut off the laser and leave the cue light on (Please see **Fig 1-I**), removing the cue-light does not impact the rate of self-stimulation, further confirming that dHPC→NAcSh is sufficient to drive reinforcing behavior, independently from the salience of the cue light, or "saliency of change".

Similarly, the effects in RTPT are very modest for a “rewarding” manipulation. There seems to be lag in the effect, in that the first 10 minutes of the task, the Chr2 group shows no strong place preference, but it gets more apparent after the 10min mark, and then stronger and stronger as the session goes on. Typically, the affects seen from rewarding brain stimulation are rapidly acquired in this paradigm. It does not take rodents long to form the association of a “room”, or even spot in a chamber, with the rewarding stimulation. The significant differences in Figure 1F appear to be driven by 6 or so mice, otherwise the groups appear similar (are there histological observations for why out of 18 mice, only a handful have % time in stim side greater than control?). The control group (Fig 1 H) show similar trends (with almost half as many mice)

We appreciate the reviewer’s comment. We would like to point out the fact that all animals that underwent the real time place preference procedure (Figure 1A-E) exhibited a preference for the light stimulation side, suggesting a rewarding effect of the stimulation. We acknowledge that the effects shown in the figure described by the reviewer (now removed to avoid confusion) may have appeared slightly more nuanced. In this procedure the chamber associated with light stimulation was changed in the middle of the 40 minutes session, forcing the animals to adjust their strategy to receive dHPC stimulation. While most of the animals expressing channelrhodopsin exhibited this adjustment, both the length of the procedure (40min vs 30min for classic RTPT) and the adjustment to the change in light stimulated chamber decrease their overall preferences when compared to a more classic RTPT procedure (see Figure 1A-E in which 100% of Chr2 animals spent above 50% of their time in the light associated chambers). Lastly, we would like to point out that in addition to RTPT, the results shown throughout the manuscript using an operant procedure reinforce the conclusion drawn by the RTPT results.

With the above point on optoICSS rates, it is hard to reconcile that this manipulation is rewarding – the effects are just so much weaker at these two common tasks (ICSS and RTPP) than seen with similar manipulations of validated/established “reward” systems.

Please see comments above.

4) There are important details related to the stimulation parameters that are missing. What intensity was the light for operant tasks? How or why did they decide on the parameters they used? Were there any other “doses” of light they tried, i.e. different train lengths, frequency, etc.? Do dCA1 neurons ever respond in this fashion (i.e 20hz for 2 seconds) during natural behaviors?

Those details have been added in the Methods section of the manuscript. We based our stimulation procedures on multiple publications looking at neurocircuits of operant reward and aversion using optogenetics, and more particularly in the NAcSh and the dHPC (PMIDS: 34646022, 34385700, 26335648, 28496380, 34160168).

5) Infusion of AMPA/NMDA antagonists into the accumbens essentially takes off-line this region that is crucial for motivated behavior (MSNs don’t do much without glu input). The authors must show that the reduction in dCA1 ICSS is specific to this manipulation and the manipulation does not, rather, simply disrupt all reward seeking behaviors (or locomotion). It is also unclear how this experiment was conducted. The methods say this was a bilateral cannula implant. Why? The optogenetic manipulations are unilateral, so why infuse into each NAc rather than just ipsilateral to the opto-stim. If indeed, both NAc receive glutamate antagonists, I believe many/most motivated behaviors would be affected, which would undermine what they are trying to show – that the glu release in the NAc from the opto stim is implicated in the reinforcing effects of that stimulation. I recommend designing an experiment that shows this microinfusion specifically disrupts your optoICSS, and not general operant responding.

6) The DREADD experiments suffer from the same design issue as cannula infusions above.

We would like to thank the reviewer for this comment. While sparse, we observed faint labelling in contralateral NAc when infusing Chr2 in dHPC. Thus, we felt the need to provide treatment in both NAc hemispheres to account for this. This has been justified in the methodology section of the manuscript : “Treatments were given bilaterally to ensure that any Chr2-induced recruitment of Enk+ or Dyn+

neurons would be silenced.” and “Treatments were given bilaterally to ensure that any Chr2-induced dHPC→NAc glutamate signaling would be blocked”

7) *They have not demonstrated that the optogenetics or chemogenetics do what they are supposed to do. Some sort of validation of these methods is necessary. Do the dCA1 neurons follow a 2 second 20hz train? Do the Gi DREADDs take Dyn and Enk “offline” equally?*

We greatly appreciate the concern brought by the reviewer. We characterized both our chemogenetic and optogenetic approaches in previous reports and, as mentioned above, these constructs have been used in prior publications to silence both dyn- and enk-neurons within the NAc. Therefore, we followed the procedures that were described in these other reports. The appropriate references have been added in the methods section of manuscript to support this.

8) *The discussion lacks any synthesis of how this pathway fits into canonical reward systems. How these neurons work in this neural module to affect motivation would help us understand why stimulation of these neurons modestly reinforces behavior.*

We expanded the discussion to further integrate this pathway into the current literature on reward circuitry. “While the number of Dyn+ MSNs responding to dHPCCaMKII+ terminal stimulation in the NAcSh remains low (~12.5%), it is plausible that constant stimulation of dHPCCaMKII+ input to NAcSh Dyn+ MSNs, either through experience or training, would strengthen the pathway, making it key to modulate positive reinforcing behaviors (Figure 8). For example, constant exposure to drugs such as cocaine enhances excitatory synaptic connections into NAc MSNs 67. This is also evident in our fiber photometry recordings where calcium transient recorded in NAcSh of mice with prior training to self-stimulation operant task (Figure 2K-M) were higher than those who have no prior exposure to dHPCCaMKII+ neuron stimulation (Suppl Figure 2F-H).

Furthermore, the discrepancy between the selective role of Dyn-MSNs in dHPCCaMKII+-driven reinforcing behavior and the small response in slice electrophysiology experiments could be due to the loss of local circuitry within the NAc during slice preparation. This could lead to the underrepresentation or absence of certain microcircuits in the NAcSh that are critical for the activation of Dyn-MSNs in dHPCCaMKII+-driven reinforcing behavior. The recent report of dHPCCaMKII+ neurons recruiting FSIs, which in turn drives the activity of the MSNs assembly in the NAc 23, suggests the potential recruitment of the local circuitry in the NAcSh to regulate Dyn+ neurons during dHPCCaMKII+ driven reinforcing behavior. Therefore, further investigation is also necessary to fully determine the contribution of local circuitry in the NAcSh in dHPCCaMKII+-driven reinforcing behavior. Additionally, there is high heterogeneity in the anatomy and function of the NAc along the rostral/caudal axis 46, indicating the need for further investigation to fully evaluate the role of NAcSh MSNs in driving reinforcement, rewarding and aversive behaviors.”

9) *The authors clearly have the capability to investigate how the neurons they study here, in particular the dCA1->NAc, contribute to natural reward seeking behaviors. They give some examples in the introduction and discussion that suggest they do, but we already know dCA1 and dCA1->NAc play a role in memory processes related to reward seeking behaviors. As stated before, the ICSS here is artificial and simply suggests somehow these neurons are related to reward seeking (again, we know this), but complimentary experiments to monitor and/or disrupt the normal activity of these neurons during ethologically relevant goal-directed behavior is needed to make this story more than an incremental advancement.*

We thank the reviewer for this insightful comment. In the revised manuscript, we used sucrose self-administration combined with an inhibitory chemogenetic approach to silence the activity of dHPC→NAcSh pathway during sucrose seeking (see **Figure 4**). With this experiment we demonstrate that silencing dHPC→NAcSh significantly decreased self-administration of sucrose pellets seeking under fixed ratio 1 schedule of reinforcement. This result further strengthens the importance of the

dHPC→NAcSh in mediating reinforcement and reward seeking in addition to its already defined role in contextual/cue memory.

10) Related to the functional connectivity assays. There are no analysis of the latencies regarding dCA1 stimulation onset and NAc LFP or GCaMP dynamics. This is crucial to interpret the extent that this stimulation acts to modulate NAc directly or indirectly. I am also confused by the summary analysis of the LFP. It appears the authors are using absolute value of the LFP before and after stimulation. Figure 2E seems to show many channels recording a decrease in LFP, and seemingly less show increase, while many show no change. Yet, in line 140 they say evoked “activation”?

I cannot seem to find what Pre-Min and Post-Max are referring to in Figure 2F; what is pre and what is post, and compare a minimum and maximum from these different times? They also seem to have recorded NAc LFPs both ipsilateral and contralateral to the dCA1 stimulation, but this is not mentioned or discussed. It is unclear how fiber photometry of GCaMP “gains both anatomical and temporal resolution of the dynamics of NAc neuronal activity...” (Line 143) when microelectrode recordings have better resolution of both. It also would seem that the results related to this in the main text (FP during dCA1 optoICSS) are confounded by the coincidence of the behavioral task when interpreting whether the opto-stimulation is affecting the FP signal (as opposed to the behavior) – the suppl. data Fig1E-H better support this interpretation.

We thank the reviewer for this comment. We included additional information in the text and also additional analyses in **Figure 2** to help understand the LFP data (**Figure 2E-H**). Briefly, we included in the text the following information regarding the LFP recordings. Pre- and post-stim amplitude were calculated by subtracting the minimum amplitude from the maximum amplitude within a 0.1s window pre- and post- dHPC stimulation (**Figure 2F**). Stimulation of dHPC^{CaMKII+} neurons evoked a significant increase in overall evoked LFP amplitude recorded in the NAc (**Figure 2G**). By calculating the time between dHPC stimulation onset and the first evoked peak (Latency to peak; **Figure 2F**), we observed a wide variability in latency to peak (**Figure 2H**), suggesting the presence of monosynaptic inputs (latency < 0.01s) and polysynaptic inputs (latency > 0.01 s) from the dHPC. The well-defined role of the NAcSh in processing reward and positive reinforcement, together with our finding of denser dHPC^{CaMKII+} projections into the NAcSh (**Figure 2C**), suggest the possible involvement of this pathway to mediate dHPC^{CaMKII+}-driven reinforcing behavior.

Regarding the reviewer’s comment about the confound between behavioral task and light stimulation, those concerns are addressed by the additional experiments now shown in supplementary **Figure 2 E-H** in which the animals receive a non-contingent light self-stimulation in an open field arena. The response observed in this scenario is tightly coupled to the stimulation of dHPC CaMKII+ neurons administered by the experimenter. These additional data demonstrate that the activation of those neurons is sufficient to trigger increase in calcium transient within the NAcSh, regardless of the behavioral task.

11) Line 153 – how is this a selective accumbal response? They’ve not compared signals in surrounding regions. This brings up another important missing control – there are no site controls in this study. They’ve not investigated other dCA1 projections, nor placed optic fibers in areas adjacent to injection sites to identify spatial specificity. One of the main claims is that while we know electrical excitation of dHPC is reinforcing, they argue this study identifies dCA1 CAMKII, and in particular those dCA1 CAMKII projecting to NAc as mediating that reward – but these neurons project elsewhere and they’ve not shown these other pathways don’t support ICSS.

We thank the reviewer for this comment. To answer this comment, and as suggested by Reviewer #2, instead of widely stimulating adjacent areas we employed the suggested dual virus strategy with Cre-recombinase-encoding retroAAV virus in the NAcSh in conjunction with Cre-dependent AAV vectors in the dHPC. This allowed us to selectively engage direct projections from the dHPC to the NAcSh during light self-stimulation procedures. We show here that, similarly to the stimulation of the overall CaMKII+ neuronal population in the dHPC, activation of this selective dHPC→NAcSh pathway is sufficient to drive reinforcing behavior (please see **Figure 3 A-D**).

12) I am not sure what the brain clearing studies are adding here. Firstly, both tracers jRCECO1a and Chr2 are

not specific to terminals, so analysis is limited to regions where fibers are present, could be fibers of passage, no specificity to functional connectivity. But, they are not really doing anything with this data set regardless. In fact, this seems like a missed opportunity to identify some projection “site-controls” to test the above point. Likewise, using similar strategies they could identify whether dCA1->NAc projecting neurons send collaterals elsewhere. Optogenetic stimulation of terminals can send antidromic action potentials back to the cell body, and if the neuron sends collaterals, these alternate projections could mediate reinforcing properties. This later concern would be alleviated if the cannula experiments were performed appropriately.

We thank the reviewer and agree that those brain clearing analysis were not providing critical data and therefore they were removed from the current version of the manuscript. Instead, we included additional comprehensive analysis of the dHPC-NAcSh input using viral tracing approaches and included updated illustrations of the projections.

In summary, we would like to respectfully request for our manuscript to be reconsidered given the additional studies that we conducted and the revisions to the text that fully address all the reviewers’ concerns.

We would like to thank you for your time and consideration.

Regards,

Dr. Jose A. Moron

Dr. Nicolas Massaly

Dr. Khairunisa Ibrahim

REVIEWER COMMENTS

Reviewer #1 (Remarks to the Author):

The authors have added considerable new data and have made other attempts to add details to the manuscript to address concerns. Generally, these have been helpful. However, I still have significant issues with the way the authors are interpreting their task and considerable reservations concerning their anatomical imaging.

1. In their reply, the authors claim to have established that the unpaired procedure is sufficient to argue that nose poke is driven by the CA1-NacSH pathway stimulation. But they are missing my point. It is the interpretation of the nose-poking that I was commenting on. For some reason the authors are very determined to argue they have found a reinforcement process in this pathway but they use a task that confounds cues and responses as the source of stimulation and so of nose-poking. This is not to say that the unpaired procedure doesn't show the stimulation is required. The question is whether it is the response for the stimulation or the response to the cue paired with the stimulation that is being recorded here. Would the animals continue to nose-poke during the light if it omitted the stimulation? Consider the experimental situation in which the light comes on above the nose poke hole and is paired with stimulation but only if the animals refrain from nose poking. Would they acquire/maintain nose poking in this situation? Something like this is required if the authors are trying to claim the mice are nose-poking for the stimulation rather than the cue is eliciting nose pokes as a conditioned response because the cue is paired with stimulation.

2. Note that demonstrating place preference is not an adequate control for this claim; it is just a Pavlovian pairing between context and stimulation which has exactly the same confound as the nose poke test.

3. The idea of assessing the influence of the pathway on sucrose self-administration is interesting but the effects are pretty modest. In addition, the cue-nose poke hole conjunction is maintained in this setup. Won't the mice nose poke without a cue that is spatially adjacent to the nose poke hole? The authors seem to suggest that the cue is necessary for discrimination. What about if the cue were over the inactive hole? Would they be able to detect the active hole then? What I am trying to make clear is that the confound induced by having the response and cue both predicting stimulation makes it impossible to know whether they are responding for the stimulation or responding because the cue predicts stimulation. Certainly, it appears that the cue is very important for the effects described here. As a consequence, I still do not believe the source of the nose poke response has been adequately characterised and, therefore, the authors' interpretation of their effects, and of the function of this pathway, is very much open to question. Perhaps the authors can consider various alternative explanations for nose poking in the discussion and, to that end, one feels that an explanation that suggests the stimulation causes an increase in spatial saliency attracting the mice to a point in space, is perfectly consistent with these data. Either the authors run experiments to establish which account is accurate or they must consider all alternative explanations and so change the interpretation of these data throughout the manuscript.

4. The retro-cre approach is reasonable but where are the images showing that the retro-AAV infusion is confined to the NacSH? If the AAV infusion extends to areas other than the NacSH then I hope the authors can see that this will mean they are stimulating a pathway that includes more than the NacSH.

5. The imaging of the tracing of the pathway remains limited to one panel in figure 2. The authors want to assure us this is a discrete pathway but the images are very dark and show only one A-P section. It would be useful, given the paper is focussed on a specific pathway, to have the anatomical characterisation of this pathway very clearly and convincingly demonstrated. This would go a long way towards increasing confidence in the overall claims.

6. There is some kind of glitch in the supplementary material with regard to what I assume is Figure S7, which has some issues with the lower panels and no caption.

Reviewer #2 (Remarks to the Author):

The authors have made a very commendable effort to address the reviewers' concerns from the initial submission, adding substantial experimental data and analyses to their manuscript. I am convinced that the manuscript should be reconsidered, and I agree that the authors' claims are now much better supported. In particular, the evidence in Figure 4 nicely shows that a putative direct dCA1->NAc projection substantially contributes to the reinforcement effect, and I thank the authors for taking this suggestion and conducting the experiments with retrogradely-transporting Cre virus. However, in my view there are still a few critical gaps that need to be filled. Hopefully with these revisions the authors can move on to publication.

Major:

1. The slice physiology experiments are a welcome addition, but the evidence for the authors' statement on line 82 that "the dHPC_CaMKII+ connect directly to dynorphin-containing neurons" is slim. Only a single Dyn+ neuron was found to have an excitatory response, and there is no quantification of the evoked response latencies for this neuron in Fig. 6 even though the latencies are nicely quantified for the Enk+ neurons. Did this one neuron at least show a latency consistent with a monosynaptic response? The EPSC looks broader and more likely to have a disynaptic component than the Enk+ EPSC shown. It was also not clear from the figure legend (though it is stated in the text) that this was the only cell showing this response. Is it possible to collect a few more neurons and quantify the latencies? There should be a bimodal distribution of monosynaptic and disynaptic responses to support the circuit diagram shown in Fig. 8.

1a. Also, what is "baseline" in the Dyn+ recordings? Why is NBQX+AP5 not compared directly to picrotoxin as it is for the Enk+ cells?

2. In their response, the authors emphasize that prolonged stimulation of dCA1 can elicit seizures. This is a very valid concern, and now I wonder whether this could drive the conditioned place preference effect in Fig. 1 in the RTPT paradigm? Mice tend to "freeze" during seizures, which would increase the amount of time spent on the stimulated side. Are the mice still moving during the increased time spent on the stimulated side? All the authors would need to show is a quantification of time moving vs. time immobile on both the paired and unpaired sides, to show that the mice are still exploring and not freezing as they express their conditioned place preference. If they are freezing, the authors may want to use an LFP electrode to make sure there are no seizures occurring (they might have this data already from the recordings in Figure 2, as a brain-wide seizure should be visible in NAc as well).

3. The additional histology illustrations in the supplement are helpful, but it is still not clear to me whether there might be neighboring regions receiving dHPC projections that could contribute to the results. For the experiments in Figure 4, could you at least show one representative raw histology image of the labeled dHPC axons in NAc and the surrounding regions, to get a sense of whether there is any labeling of adjacent areas such as septum? Even with the retrograde Cre virus it is possible there was uptake in surrounding areas. I expected to see this histology in addition to the cell body image in Fig. 4B.

Minor comments:

4. I'm not sure it is fully accurate to say that "dHPC-driven reinforcement... is mediated via inputs to the NAcSh" (line 259-260). This is also implied in the abstract. If this were true, then glutamatergic blockers in the NAc should have *completely* blocked the reinforcement effect of dCA1 stim. Instead, you showed that dCA1 stim can drive reinforcement behavior in a way that is *partially* dependent on direct glutamatergic transmission in the direct dCA1->NAc circuit. I suggest adjusting your language to match the results.

5. Relatedly, it would be worth discussing the partial effect of glutamatergic block in the NAc. Possible explanations include: (1) dCA1 subpopulations projecting to other regions also contribute, (2) the dCA1->NAc cells collateralize to other regions that also contribute, (3) there is some memory-dependent association with the stimulation-paired port that does not require active transmission between dCA1 and NAc in the test session.

6. In the abstract, what does "enhanced glutamatergic signaling" mean? Do you mean "relies on glutamatergic signaling"?

7. The title of Sup Fig 3 might have a typo – it says "sucrose self-administration" but the figure is showing optogenetic self-stimulation.

Reviewer #3 (Remarks to the Author):

COMMENTS ON REVISION:

Reviewer Comment:

The authors have addressed most of my concerns, and the main concerns that I originally thought precluded their conclusions were addressed with additional experiments (save point 5). I still see some minor things that should be revised. I think the additional experiments were important to perform and enhance the interpretation/conclusions of the manuscript considerably.

Major concerns:

Define and be consistent throughout the manuscript what optoICSS means as a readout. They primarily use reinforcement but then also use statements like "drive reward seeking" or "trigger light self-stimulation" which they've not shown. I believe the authors are incorrect and stating that their experiments show these neurons "drive reward seeking" (eg. Line 206, the main summary statement). All ICSS can show is that the stimulation reinforces the behavior directly preceding the stimulation. To make this statement, they must show causally that activation of these neurons increases behaviors directly related to obtaining a goal. The best they can do with the present data is say those neurons may be implicated in reward seeking.

We thank the reviewer for this comment that was also brought by reviewer #2. In the revised manuscript, we used sucrose self-administration combined with an inhibitory chemogenetic approach to silence the activity of dHPC \rightarrow NAcSh pathway and assess its contribution to sucrose seeking behavior (see Figure 4). With this experiment we demonstrate that silencing dHPC \rightarrow NAcSh significantly decreases goal directed behavior for sucrose self-administration. This result further strengthens the importance of the dHPC \rightarrow NAcSh in mediating reinforcement and reward seeking in addition to its already defined role in contextual/cue memory. We also went carefully through the manuscript to make sure that the appropriate terms were used.

Reviewer Comment:

Thank you for performing this important additional experiment. It addresses my prior concern.

Note that the figure caption states in (B) that the example image has a fiber, but these are DREADD experiments, so either this is a copy paste error or the animals are used in other experiments which is

unclear. Either way I don't see a fiber tract in that image.

The optoICSS rates are very low. These are several orders of magnitude lower than what is observed for optoICSS of regions that were then directly implicated in reward/reinforcement processes (e.g. 1000s of lever presses at FR1 in 30 minutes). Also, nosepoke rates are typically higher than lever press rates given same reward, making the low rates observed in this study even less convincing as evidence of "self-stimulation". So, the low self-stimulation rates here is a glaring discrepancy when considered with past studies doing similar experiments. Did the authors test whether the mice will perform the oICSS without the cue-light? The rates reported here suggest that the mice are not particularly engaged in the self-stimulation itself, so I wonder if the stimulation is increasing the saliency of the change in (i.e. turn off) the active cue light. The rates in this report are more akin to how rodents respond to visual stimuli alone. Given the importance of the dHPC in governing environmental saliency, the above is a possible alternate explanation that should be addressed. Other information about the timeseries of the optoICSS can shed light on whether these neurons are involved in reward processing (and therefore reinforcement). How are the mice responding across a session? Do they press more in the beginning? Do they run around or sniff after a stimulation? Classically, rewarding brain stimulation (electrical or optogenetic) will produce sniffing or forward locomotion, and will typically occur in bursts (or even continuously) across a session. While, yes, the active poke rates are greater than the inactive, the ≤ 1 poke per min is not very convincing towards deeming this population of neurons as supporting self-stimulation. The authors at the very least need to discuss this and attempt to reconcile. But even with a strong justification for why this rate is so low yet still indicates reinforcement or drives reward, I doubt many in the field will consider this strong evidence for their overall claim about optoICSS.

We performed several additional experiments to address these comments/concerns:

1. Because long and synchronous stimulation of CaMKII+ neuronal population in the dHPC has been shown to trigger the development of epileptic seizures, we first applied restrictive parameters, including a 10sec time out after each dHPC self-stimulation, and a maximum of 100 stimulations per session after which the overall session will end. To answer the first part of the reviewer's comment we removed those restrictive parameters (Figure 1 K-L). In those graphs we can see that removing the 10sec time out period increases the number of stimulation with a session (yellow/green bar and single plots), and removing both time outs and 100 maximum self-stimulation per session, further increase the number of active interactions with the nose poke inset to levels reported in other publications assessing reinforcement and using optogenetics as an approach to interrogate a selective circuits. We also depicted in Figure 1H and 1L, the continuous distribution of self-stimulation across a session confirming that the animals do seek for dHPC stimulation across the session.
2. To address the comment regarding the saliency of the change, we trained animals to self-stimulate and removed the cue-light indicating the reward availability (please see Suppl Fig 1). In this case, and in opposition to when we shut off the laser and leave the cue light on (Please see Fig 1-I), removing the cue-light does not impact the rate of self-stimulation, further confirming that dHPC \rightarrow NAcSh is sufficient to drive reinforcing behavior, independently from the salience of the cue light, or "saliency of change".

Reviewer Comment:

Thank you for the additional optoICSS experiments. It is clearer now that the stimulation is reinforcing the active lever press.

Similarly, the effects in RTPT are very modest for a "rewarding" manipulation. There seems to be lag in the effect, in that the first 10 minutes of the task, the Chr2 group shows no strong place preference, but it gets more apparent after the 10min mark, and then stronger and stronger as the session goes on. Typically, the affects seen from rewarding brain stimulation are rapidly acquired in this paradigm. It does not take rodents long to form the association of a "room", or even spot in a chamber, with the rewarding stimulation. The significant differences in Figure 1F appear to be driven

by 6 or so mice, otherwise the groups appear similar (are there histological observations for why out of 18 mice, only a handful have % time in stim side greater than control?). The control group (Fig 1 H) show similar trends (with almost half as many mice)

We appreciate the reviewer's comment. We would like to point out the fact that all animals that underwent the real time place preference procedure (Figure 1A-E) exhibited a preference for the light stimulation side, suggesting a rewarding effect of the stimulation. We acknowledge that the effects shown in the figure described by the reviewer (now removed to avoid confusion) may have appeared slightly more nuanced. In this procedure the chamber associated with light stimulation was changed in the middle of the 40 minutes session, forcing the animals to adjust their strategy to receive dHPC stimulation. While most of the animals expressing channelrhodopsin exhibited this adjustment, both the length of the procedure (40min vs 30min for classic RTPT) and the adjustment to the change in light stimulated chamber decrease their overall preferences when compared to a more classic RTPT procedure (see Figure 1A-E in which 100% of ChR2 animals spent above 50% of their time in the light associated chambers). Lastly, we would like to point out that in addition to RTPT, the results shown throughout the manuscript using an operant procedure reinforce the conclusion drawn by the RTPT results.

Reviewer Comment:

Thank you for the additional RTPT experiment and clarification. I agree that this supports the optoICSS. Since the authors did not do a true "preference" test, in the sense that they paired on side with stimulation and then test again later without the stimulation to see where they preferred to be, I would suggest slightly revising this text (lines 100-102 or so) to remove terms like showed "strong" preference (also in general these statements are speculative, what is "strong" preference?), to just spent more time on the light paired side, which is really all you can assess from this assay. One other suggestion, I think many in the field see ICSS as the gold standard for the assessments the authors are conducting, and since the RTPT here mainly is just supportive of the optoICSS (i.e. they have not presented much more analysis of this test beyond this), I think the flow of the findings may work better leading with the ICSS and showing the RTPT as support. But, this is up to them.

With the above point on optoICSS rates, it is hard to reconcile that this manipulation is rewarding – the effects are just so much weaker at these two common tasks (ICSS and RTPT) than seen with similar manipulations of validated/established "reward" systems. Please see comments above.

Reviewer Comment:

I think the data on optogenetic reinforcement is clearer now.

4) There are important details related to the stimulation parameters that are missing. What intensity was the light for operant tasks? How or why did they decide on the parameters they used? Were there any other "doses" of light they tried, i.e. different train lengths, frequency, etc.? Do dCA1 neurons ever respond in this fashion (i.e 20hz for 2 seconds) during natural behaviors?

Those details have been added in the Methods section of the manuscript. We based our stimulation procedures on multiple publications looking at neurocircuits of operant reward and aversion using optogenetics, and more particularly in the NAcSh and the dHPC (PMIDS: 34646022, 34385700, 26335648, 28496380, 34160168).

Reviewer Comment:

Make sure because I still don't see details like laser power for the optoICSS. These info are important for reproducibility or folks who want to follow up on these findings.

5) Infusion of AMPA/NMDA antagonists into the accumbens essentially takes off-line this region that is crucial for motivated behavior (MSNs don't do much without glu input). The authors must show that the reduction in dCA1 ICSS is specific to this manipulation and the manipulation does not, rather, simply disrupt all reward seeking behaviors (or locomotion). It is also unclear how this experiment was conducted. The methods say this was a bilateral cannula implant. Why? The optogenetic manipulations are unilateral, so why infuse into each NAc rather than just ipsilateral to the opto-stim. If indeed, both NAc receive glutamate antagonists, I believe many/most motivated behaviors would be affected, which would undermine what they are trying to show – that the glu release in the NAc from the opto stim is implicated in the reinforcing effects of that stimulation. I recommend designing an experiment that shows this microinfusion specifically disrupts your optoICSS, and not general operant responding.

6) The DREADD experiments suffer from the same design issue as cannula infusions above.

We would like to thank the reviewer for this comment. While sparse, we observed faint labelling in contralateral NAc when infusing ChR2 in dHPC. Thus, we felt the need to provide treatment in both NAc hemispheres to account for this. This has been justified in the methodology section of the manuscript : "Treatments were given bilaterally to ensure that any ChR2-induced recruitment of Enk+ or Dyn+ neurons would be silenced." and "Treatments were given bilaterally to ensure that any ChR2-induced dHPC→NAc glutamate signaling would be blocked"

Reviewer Comment:

This is not really addressing the issue with this experiment. As I said before, glu antagonist bilaterally infused into the NAc will disrupt most, if not all, motivated behaviors. So there is no basis to claim that it is specifically attenuating the ICSS here. I do not think the authors can say much from how the experiment was conducted. I do not think it adds much anyway, given the later DREADD experiments and slice physiology. I think the authors should walk back their interpretation of this experiment or remove it. An appropriate claim could be that Glu transmission in the NAc is needed for operant responding for dHPC→NAc optogenetic stimulation, but this again doesn't add anything novel.

7) They have not demonstrated that the optogenetics or chemogenetics do what they are supposed to do. Some sort of validation of these methods is necessary. Do the dCA1 neurons follow a 2 second 20hz train? Do the Gi DREADDs take Dyn and Enk "offline" equally?

We greatly appreciate the concern brought by the reviewer. We characterized both our chemogenetic and optogenetic approaches in previous reports and, as mentioned above, these constructs have been used in prior publications to silence both dyn- and enk-neurons within the NAc. Therefore, we followed the procedures that were described in these other reports. The appropriate references have been added in the methods section of manuscript to support this.

Reviewer Comment:

Thanks for the clarification.

8) The discussion lacks any synthesis of how this pathway fits into canonical reward systems. How these neurons work in this neural module to affect motivation would help us understand why stimulation of these neurons modestly reinforces behavior.

We expanded the discussion to further integrate this pathway into the current literature on reward circuitry. "While the number of Dyn+ MSNs responding to dHPCCaMKII+ terminal stimulation in the NAcSh remains low (~12.5%), it is plausible that constant stimulation of dHPCCaMKII+ input to NAcSh Dyn+ MSNs, either through experience or training, would strengthen the pathway, making it key to modulate positive reinforcing behaviors (Figure 8). For example, constant exposure to drugs such as cocaine enhances excitatory synaptic connections into NAc MSNs 67. This is also evident in our fiber photometry recordings where calcium transient recorded in NAcSh of mice with prior training to self-stimulation operant task (Figure 2K-M) were higher than those who have no prior exposure to dHPCCaMKII+ neuron stimulation (Suppl Figure 2F-H).

Furthermore, the discrepancy between the selective role of Dyn-MSNs in dHPCCaMKII+-driven reinforcing behavior and the small response in slice electrophysiology experiments could be due to the loss of local circuitry within the NAc during slice preparation. This could lead to the underrepresentation or absence of certain microcircuits in the NAcSh that are critical for the activation of Dyn-MSNs in dHPCCaMKII+-driven reinforcing behavior. The recent report of dHPCCaMKII+ neurons recruiting FSIs, which in turn drives the activity of the MSNs assembly in the NAc 23, suggests the potential recruitment of the local circuitry in the NAcSh to regulate Dyn+ neurons during dHPCCaMKII+ driven reinforcing behavior. Therefore, further investigation is also necessary to fully determine the contribution of local circuitry in the NAcSh in dHPCCaMKII+-driven reinforcing behavior. Additionally, there is high heterogeneity in the anatomy and function of the NAc along the rostral/caudal axis 46, indicating the need for further investigation to fully evaluate the role of NAcSh MSNs in driving reinforcement, rewarding and aversive behaviors.”

Reviewer Comment:

These are useful discussion points. The newly added last sentence should be revised. “potential” used twice and it is an awkward way to end the manuscript.

9) The authors clearly have the capability to investigate how the neurons they study here, in particular the dCA1->NAc, contribute to natural reward seeking behaviors. They give some examples in the introduction and discussion that suggest they do, but we already know dCA1 and dCA1->NAc play a role in memory processes related to reward seeking behaviors. As stated before, the ICSS here is artificial and simply suggests somehow these neurons are related to reward seeking (again, we know this), but complimentary experiments to monitor and/or disrupt the normal activity of these neurons during ethologically relevant goal-directed behavior is needed to make this story more than an incremental advancement.

We thank the reviewer for this insightful comment. In the revised manuscript, we used sucrose self-administration combined with an inhibitory chemogenetic approach to silence the activity of dHPC Δ NAcSh pathway during sucrose seeking (see Figure 4). With this experiment we demonstrate that silencing dHPC Δ NAcSh significantly decreased self-administration of sucrose pellets seeking under fixed ratio 1 schedule of reinforcement. This result further strengthens the importance of the dHPC Δ NAcSh in mediating reinforcement and reward seeking in addition to its already defined role in contextual/cue memory.

Reviewer Comment:

Yes this is a good experiment thanks.

10) Related to the functional connectivity assays. There are no analysis of the latencies regarding dCA1 stimulation onset and NAc LFP or GCaMP dynamics. This is crucial to interpret the extent that this stimulation acts to modulate NAc directly or indirectly. I am also confused by the summary analysis of the LFP. It appears the authors are using absolute value of the LFP before and after stimulation. Figure 2E seems to show many channels recording a decrease in LFP, and seemingly less show increase, while many show no change. Yet, in line 140 they say evoked “activation”? I cannot seem to find what Pre-Min and Post-Max are referring to in Figure 2F; what is pre and what is post, and compare a minimum and maximum from these different times? They also seem to have recorded NAc LFPs both ipsilateral and contralateral to the dCA1 stimulation, but this is not mentioned or discussed.

It is unclear how fiber photometry of GCaMP “gains both anatomical and temporal resolution of the dynamics of NAc neuronal activity...” (Line 143) when microelectrode recordings have better resolution of both. It also would seem that the results related to this in the main text (FP during dCA1 optoICSS) are confounded by the coincidence of the behavioral task when interpreting whether the opto-stimulation is affecting the FP signal (as opposed to the behavior) – the suppl. data Fig1E-H better support this interpretation.

We thank the reviewer for this comment. We included additional information in the text and also additional analyses in Figure 2 to help understand the LFP data (Figure 2E-H). Briefly, we included in the text the following information regarding the LFP recordings. Pre- and post-stim amplitude were calculated by subtracting the minimum amplitude from the maximum amplitude within a 0.1s window pre- and post- dHPC stimulation (Figure 2F). Stimulation of dHPCCaMKII+ neurons evoked a significant increase in overall evoked LFP amplitude recorded in the NAc (Figure 2G). By calculating the time between dHPC stimulation onset and the first evoked peak (Latency to peak; Figure 2F), we observed a wide variability in latency to peak (Figure 2H), suggesting the presence of monosynaptic inputs (latency < 0.01s) and polysynaptic inputs (latency > 0.01 s) from the dHPC. The well-defined role of the NAcSh in processing reward and positive reinforcement, together with our finding of denser dHPCCaMKII+ projections into the NAcSh (Figure 2C), suggest the possible involvement of this pathway to mediate dHPCCaMKII+-driven reinforcing behavior.

Reviewer Comment:

Thank you for adding this.

Regarding the reviewer's comment about the confound between behavioral task and light stimulation, those concerns are addressed by the additional experiments now shown in supplementary Figure 2 E-H in which the animals receive a non-contingent light self-stimulation in an open field arena. The response observed in this scenario is tightly coupled to the stimulation of dHPC CaMKII+ neurons administered by the experimenter. These additional data demonstrate that the activation of those neurons is sufficient to trigger increase in calcium transient within the NAcSh, regardless of the behavioral task.

Reviewer Comment:

Good experiment to address the concern here.

11) Line 153 – how is this a selective accumbal response? They've not compared signals in surrounding regions. This brings up another important missing control – there are no site controls in this study. They've not investigated other dCA1 projections, nor placed optic fibers in areas adjacent to injection sites to identify spatial specificity. One of the main claims is that while we know electrical excitation of dHPC is reinforcing, they argue this study identifies dCA1 CaMKII, and in particular those dCA1 CaMKII projecting to NAc as mediating that reward – but these neurons project elsewhere and they've not shown these other pathways don't support ICSS.

We thank the reviewer for this comment. To answer this comment, and as suggested by Reviewer #2, instead of widely stimulating adjacent areas we employed the suggested dual virus strategy with Cre-recombinase-encoding retroAAV virus in the NAcSh in conjunction with Cre-dependent AAV vectors in the dHPC. This allowed us to selectively engage direct projections from the dHPC to the NAcSh during light self-stimulation procedures. We show here that, similarly to the stimulation of the overall CaMKII+ neuronal population in the dHPC, activation of this selective dHPC→NAcSh pathway is sufficient to drive reinforcing behavior (please see Figure 3 A-D).

Reviewer Comment:

The AAV-retro does add more specificity here. Thanks.

12) I am not sure what the brain clearing studies are adding here. Firstly, both tracers jRCECO1a and Chr2 are not specific to terminals, so analysis is limited to regions where fibers are present, could be fibers of passage, no specificity to functional connectivity. But, they are not really doing anything with this data set regardless. In fact, this seems like a missed opportunity to identify some projection "site-controls" to test the above point. Likewise, using similar strategies they could identify whether dCA1→NAc projecting neurons send collaterals elsewhere. Optogenetic stimulation of terminals can send antidromic action potentials back to the cell body, and if the neuron sends collaterals, these alternate

projections could mediate reinforcing properties. This later concern would be alleviated if the cannula experiments were performed appropriately.

We thank the reviewer and agree that those brain clearing analysis were not providing critical data and therefore they were removed from the current version of the manuscript. Instead, we included additional comprehensive analysis of the dHPC-NAcSh input using viral tracing approaches and included updated illustrations of the projections.

Reviewer Comment:

Good idea.

Reviewer Comment:

Other suggestions:

Line 258 and Methods heading line 389– "...light self-stimulation". I do not think this is the correct phrase. I know what they mean, but others not familiar with these paradigms may be confused and interpret this as light – i.e. modest, 'lite', weak, etc. I think optogenetic self-stimulation or just self-stimulation. If anything, that phrase should read 'light self-administration'.

We thank the Reviewers for their helpful and constructive comments. These suggestions greatly assisted us in strengthening the claims and overall dataset presented in the manuscript. The results included in this revised version add new control experiments and a more thorough discussion to address each of the comments raised by the Reviewers (reviewed point-by-point below). We do hope you will now agree that, in its current state, our manuscript fully satisfies all the reviewers' concerns.

All the modifications made to the manuscript and added experiments have been highlighted in blue throughout the text. See below a point-by-point response to each of the concerns/comments raised by the Reviewers:

Reviewer #1

The authors have added considerable new data and have made other attempts to add details to the manuscript to address concerns. Generally, these have been helpful. However, I still have significant issues with the way the authors are interpreting their task and considerable reservations concerning their anatomical imaging.

1. In their reply, the authors claim to have established that the unpaired procedure is sufficient to argue that nose poke is driven by the CA1-NacSH pathway stimulation. But they are missing my point. It is the interpretation of the nose-poking that I was commenting on. For some reason the authors are very determined to argue they have found a reinforcement process in this pathway but they use a task that confounds cues and responses as the source of stimulation and so of nose-poking. This is not to say that the unpaired procedure doesn't show the stimulation is required. The question is whether it is the response for the stimulation or the response to the cue paired with the stimulation that is being recorded here. Would the animals continue to nose-poke during the light if it omitted the stimulation?

Consider the experimental situation in which the light comes on above the nose poke hole and is paired with stimulation but only if the animals refrain from nose poking. Would they acquire/maintain nose poking in this situation? Something like this is required if the authors are trying to claim the mice are nose-poking for the stimulation rather than the cue is eliciting nose pokes as a conditioned response because the cue is paired with stimulation.

We thank the reviewer for this comment and the experimental suggestion.

To fully address the reviewer's comment in this revised version of the manuscript, we conducted a self-stimulation experiment with **no cues throughout the entire experiment** (Suppl. Figure 1F). We demonstrated that mice expressing ChR2 in the dHPC CaMKII⁺ neurons interacted significantly more with the active nose inserts even from session 1. Importantly, control mice which do not express ChR2 in the dHPC do not interact with the active nose insert (Suppl. Figure 1G). Furthermore, to fully investigate the maintenance and robustness of self-stimulation with **no cues**, the active and inactive nose inserts were swapped for sessions 6-8. Despite the change in the nose insert paired with stimulation and the absence of cues to indicate the availability of the stimulation, these mice still maintained the same level of self-stimulation (FR1 schedule; Suppl. Figure 1H).

This additional finding is in line with the results included in our previous manuscript version, in which we used:

- 1) a Real-Time Place Test (RTPT) using an apparatus consisting of two identical black matte compartments with **no external cues or context associated with the "light-associated compartment"**. In this experiment (Figure 1C-E), mice exhibited a significantly higher time spent for the compartment where they received dHPC stimulation, suggesting a preference for the activation of CaMKII⁺ neurons in the dHPC independent of cues or context.
- 2) extinction sessions in Figure 1I, where cue light was still ON in the active nose insert but the lasers were turned OFF, resulting in no stimulation delivered with active nose pokes. We found that the mice stopped interacting with the active nose insert during these sessions. When the active nose insert is paired again with stimulation during the reinstatement session, the self-stimulation behavior is

reinstated. This suggests that the activation of CaMKII⁺ neurons in the dHPC is reinforcing and independent from the cue light being paired with the “reinforcer”.

- 3) “no cue” sessions in **Suppl. Figure 1C-E**, where the cue light for the active nose insert was turned OFF but the lasers were left ON. Mice continue to maintain the number of active nose pokes in these “no cue” sessions compared to their “cue” sessions. This again demonstrates that stimulation in the CaMKII⁺ neurons is reinforcing and independent of the presence of the cue light in the active nose insert.

Altogether, we believe that those evidence demonstrate that animals are seeking for light stimulation independently of the presence of a cue or specific context.

2. Note that demonstrating place preference is not an adequate control for this claim; it is just a Pavlovian pairing between context and stimulation which has exactly the same confound as the nose poke test.

We appreciate the feedback provided by the reviewer. In our Real-Time Place Test (RTPT), none of the compartments had cues or context (**Figure 1A**). While the lack of discrete cues or context may eliminate the possibility of Pavlovian conditioning, it remains possible that other elements might serve as distinctions between those two compartments (such as marking with urine or feces within the 30-minute test). We believe that the additional data provided in **Suppl. Figure 1F-H**, together with our initial results in **Suppl. Figure 1C-E**, provides further evidence that mice engage in self-stimulation of the dHPC CaMKII⁺ neurons independent of cues or context.

3. The idea of assessing the influence of the pathway on sucrose self-administration is interesting but the effects are pretty modest. In addition, the cue-nose poke hole conjunction is maintained in this setup. Won't the mice nose poke without a cue that is spatially adjacent to the nose poke hole? The authors seem to suggest that the cue is necessary for discrimination. What about if the cue were over the inactive hole? Would they be able to detect the active hole then? What I am trying to make clear is that the confound induced by having the response and cue both predicting stimulation makes it impossible to know whether they are responding for the stimulation or responding because the cue predicts stimulation. Certainly, it appears that the cue is very important for the effects described here. As a consequence, I still do not believe the source of the nose poke response has been adequately characterised and, therefore, the authors' interpretation of their effects, and of the function of this pathway, is very much open to question. Perhaps the authors can consider various alternative explanations for nose poking in the discussion and, to that end, one feels that an explanation that suggests the stimulation causes an increase in spatial saliency attracting the mice to a point in space, is perfectly consistent with these data. Either the authors run experiments to establish which account is accurate or they must consider all alternative explanations and so change the interpretation of these data throughout the manuscript.

As per our response to point 1 made by the reviewer, in this revised manuscript, we have conducted self-stimulation experiment with no cues throughout the entire experiment (**Suppl. Figure 1F**). We showed the despite the lack of cues in the nose inserts, mice that expressed Chr2 in the dHPC CaMKII⁺ neurons still nose poke significantly higher for stimulation compared to control mice (**Suppl. Figure 1G-H**).

4. The retro-cre approach is reasonable but where are the images showing that the retro-AAV infusion is confined to the NacSH? If the AAV infusion extends to areas other than the NacSH then I hope the authors can see that this will mean they are stimulating a pathway that includes more than the NacSH.

We thank you for this excellent point which was also brought up by Reviewer #2. In this revised version of the manuscript, we have included a representative image of the dHPC fibers in the NAc with the retro-cre approach (**Figure 5C**). We have also included a figure validating projecting fibers in all the mice used in the retro-AAV + DIO-Gi DREADD approach (**Suppl. Figure 8C**).

5. The imaging of the tracing of the pathway remains limited to one panel in figure 2. The authors want to assure us this is a discrete pathway but the images are very dark and show only one A-P section. It would be useful,

given the paper is focussed on a specific pathway, to have the anatomical characterisation of this pathway very clearly and convincingly demonstrated. This would go a long way towards increasing confidence in the overall claims.

We appreciate the reviewer's feedback. We have included representative images of brain sections that encompass the anterior-to-posterior extent of the NAc region in **Figure 2** and **Suppl. Figure 2**. In this tracing experiment, we expressed AAV-CaMKII-eYFP in the dHPC and tracked the resulting projecting fibers within the NAc region. We observed that the majority of these fibers were concentrated in the rostral segment of the NAc.

6. There is some kind of glitch in the supplementary material with regard to what I assume is Figure S7, which has some issues with the lower panels and no caption.

We apologize for this glitch. We have fixed the figure and ensured that the legend is attached (**Suppl. Figure 8** in the revised version).

Reviewer #2

The authors have made a very commendable effort to address the reviewers' concerns from the initial submission, adding substantial experimental data and analyses to their manuscript. I am convinced that the manuscript should be reconsidered, and I agree that the authors' claims are now much better supported. In particular, the evidence in Figure 4 nicely shows that a putative direct dCA1->NAc projection substantially contributes to the reinforcement effect, and I thank the authors for taking this suggestion and conducting the experiments with retrogradely-transporting Cre virus. However, in my view there are still a few critical gaps that need to be filled. Hopefully with these revisions the authors can move on to publication.

Major:

1. The slice physiology experiments are a welcome addition, but the evidence for the authors' statement on line 82 that "the dHPC_CaMKII+ connect directly to dynorphin-containing neurons" is slim. Only a single Dyn+ neuron was found to have an excitatory response, and there is no quantification of the evoked response latencies for this neuron in Fig. 6 even though the latencies are nicely quantified for the Enk+ neurons. Did this one neuron at least show a latency consistent with a monosynaptic response? The EPSC looks broader and more likely to have a disynaptic component than the Enk+ EPSC shown. It was also not clear from the figure legend (though it is stated in the text) that this was the only cell showing this response. Is it possible to collect a few more neurons and quantify the latencies? There should be a bimodal distribution of monosynaptic and disynaptic responses to support the circuit diagram shown in Fig. 8.

1a. Also, what is "baseline" in the Dyn+ recordings? Why is NBQX+AP5 not compared directly to picrotoxin as it is for the Enk+ cells?

We apologize for the Dyn⁺ slice physiology being confusing. In the previous version of the manuscript, we found 2 Dyn⁺ cells in the dorsal NAc shell that showed small, but consistent responses to light stimulation. In the first cell, bath application of NBQX + AP5 completely blocked the synaptic response, which suggests dependence on glutamatergic signaling and hinders further pharmacological analysis (**Suppl. Figure 5A** in the revised version). In the second cell, bicuculline was bath applied first and the entire response was blocked indicating GABAergic signaling is also involved. As the responses were small (approximately 15pA peak amplitude), determining latencies was too difficult with trace-to-trace noise. The term "Baseline" in the Dyn⁺ recordings refers to evoked response in aCSF bath before GABAergic or glutamatergic blockers were bath applied.

In this revised version, we have included more Dyn⁺ cells which allowed us to quantify the latencies and jitter. In this experiment, we recorded evoked baseline responses in aCSF before applying bicuculline in the bath to block GABAergic signaling. If the cells have any residual evoked response with the presence of GABAergic blocker in the bath, we bath applied NBQX/AP5 to block glutamatergic signaling. The average latency from the 9 cells recorded suggests both monosynaptic and polysynaptic inputs from dHPC

to Dyn⁺ cells in the NAcSh (**Figure 7H-L**) supporting the circuit diagram in **Figure 9** (updated in the revised version).

2. In their response, the authors emphasize that prolonged stimulation of dCA1 can elicit seizures. This is a very valid concern, and now I wonder whether this could drive the conditioned place preference effect in Fig. 1 in the RTPT paradigm? Mice tend to “freeze” during seizures, which would increase the amount of time spent on the stimulated side. Are the mice still moving during the increased time spent on the stimulated side? All the authors would need to show is a quantification of time moving vs. time immobile on both the paired and unpaired sides, to show that the mice are still exploring and not freezing as they express their conditioned place preference. If they are freezing, the authors may want to use an LFP electrode to make sure there are no seizures occurring (they might have this data already from the recordings in Figure 2, as a brain-wide seizure should be visible in NAc as well).

We thank the reviewer for this comment. We analyzed the distance traveled by the mice in the paired vs. unpaired sides in the graph below. We found that the mice moved more in the stimulation paired compartment compared to the unpaired compartment, eliminating the seizures as a confound for the increase in time spent in the paired compartment. Furthermore, to mitigate the risk of seizures during prolonged dHPC stimulation, we used a lower stimulation power of 4mW in contrast to the more commonly used 6-10mW in other studies.

3. The additional histology illustrations in the supplement are helpful, but it is still not clear to me whether there might be neighboring regions receiving dHPC projections that could contribute to the results. For the experiments in Figure 4, could you at least show one representative raw histology image of the labeled dHPC axons in NAc and the surrounding regions, to get a sense of whether there is any labeling of adjacent areas such as septum? Even with the retrograde Cre virus it is possible there was uptake in surrounding areas. I expected to see this histology in addition to the cell body image in Fig. 4B

We appreciate this excellent point raised by the reviewer, which was also raised by Reviewer #1. In the revised manuscript, we have included a representative image of the dHPC fibers in the NAc with the retro-cre viral approach (**Figure 5C** in the revised version). Additionally, we have included a figure that validates projecting fibers in all the mice used in the retro-AAV + DIO-Gi DREAAD experiment (**Suppl. Figure 8C**). We have also included a more comprehensive tracing figure (**Figure 2** and **Suppl. Figure 2**), showing that the dHPC projections are predominantly in the rostral part of the NAc.

Minor comments:

4. I'm not sure it is fully accurate to say that “dHPC-driven reinforcement... is mediated via inputs to the NAcSh” (line 259-260). This is also implied in the abstract. If this were true, then glutamatergic blockers in the NAc should have *completely* blocked the reinforcement effect of dCA1 stim. Instead, you showed that dCA1 stim can drive reinforcement behavior in a way that is *partially* dependent on direct glutamatergic transmission in the direct dCA1->NAc circuit. I suggest adjusting your language to match the results.

We thank the reviewer and would like to apologize for the language used. We have revised our discussion and abstract to be more accurate with our findings (Lines 271-273).

5. Relatedly, it would be worth discussing the partial effect of glutamatergic block in the NAc. Possible explanations include: (1) dCA1 subpopulations projecting to other regions also contribute, (2) the dCA1→NAc cells collateralize to other regions that also contribute, (3) there is some memory-dependent association with the stimulation-paired port that does not require active transmission between dCA1 and NAc in the test session.

We appreciate the reviewer's feedback and have included possible explanations for the partial inhibition of self-stimulation behavior with the glutamatergic antagonist in the NAcSh within our discussion (Lines 298-302).

6. In the abstract, what does "enhanced glutamatergic signaling" mean? Do you mean "relies on glutamatergic signaling"?

We apologize for using vague terms. We have rephrased the sentence to accurately convey our findings, which suggests that the reinforcing behavior induced by stimulation of CaMKII⁺ neurons is, to some extent, reliant on the glutamatergic signaling within the NAcSh (Lines 47-49).

7. The title of Sup Fig 3 might have a typo – it says "sucrose self-administration" but the figure is showing optogenetic self-stimulation.

We apologize for this typo. We have corrected it to optogenetic self-stimulation (updated to **Suppl. Figure 4**).

Reviewer#3

Reviewer Comment:

The authors have addressed most of my concerns, and the main concerns that I originally thought precluded their conclusions were addressed with additional experiments (save point 5). I still see some minor things that should be revised. I think the additional experiments were important to perform and enhance the interpretation/conclusions of the manuscript considerably.

Major concerns:

Define and be consistent throughout the manuscript what optoICSS means as a readout. They primarily use reinforcement but then also use statements like "drive reward seeking" or "trigger light self-stimulation" which they've not shown. I believe the authors are incorrect and stating that their experiments show these neurons "drive reward seeking" (eg. Line 206, the main summary statement). All ICSS can show is that the stimulation reinforces the behavior directly preceding the stimulation. To make this statement, they must show causally that activation of these neurons increases behaviors directly related to obtaining a goal. The best they can do with the present data is say those neurons may be implicated in reward seeking.

We thank the reviewer for this comment that was also brought by reviewer #2. In the revised manuscript, we used sucrose self-administration combined with an inhibitory chemogenetic approach to silence the activity of dHPC→NAcSh pathway and assess its contribution to sucrose seeking behavior (see Figure 4). With this experiment we demonstrate that silencing dHPC→NAcSh significantly decreases goal directed behavior for sucrose self-administration. This result further strengthens the importance of the dHPC→NAcSh in mediating reinforcement and reward seeking in addition to its already defined role in contextual/cue memory. We also went carefully through the manuscript to make sure that the appropriate terms were used.

Reviewer Comment:

Thank you for performing this important additional experiment. It addresses my prior concern.

Note that the figure caption states in (B) that the example image has a fiber, but these are DREADD experiments, so either this is a copy paste error or the animals are used in other experiments which is unclear. Either way I don't see a fiber tract in that image.

We apologize for the error. This figure caption which is now updated to **Figure 5B** has been corrected.

The optoICSS rates are very low. These are several orders of magnitude lower than what is observed for optoICSS of regions that were then directly implicated in reward/reinforcement processes (e.g. 1000s of lever presses at FR1 in 30 minutes). Also, nosepoke rates are typically higher than lever press rates given same reward, making the low rates observed in this study even less convincing as evidence of "self-stimulation". So, the low self-stimulation rates here is a glaring discrepancy when considered with past studies doing similar experiments. Did the authors test whether the mice will perform the oICSS without the cue-light? The rates reported here suggest that the mice are not particularly engaged in the self-stimulation itself, so I wonder if the stimulation is increasing the saliency of the change in (i.e. turn off) the active cue light. The rates in this report are more akin to how rodents respond to visual stimuli alone. Given the importance of the dHPC in governing environmental saliency, the above is a possible alternate explanation that should be addressed. Other information about the timeseries of the optoICSS can shed light on whether these neurons are involved in reward processing (and therefore reinforcement). How are the mice responding across a session? Do they press more in the beginning? Do they run around or sniff after a stimulation? Classically, rewarding brain stimulation (electrical or optogenetic) will produce sniffing or forward locomotion, and will typically occur in bursts (or even continuously) across a session. While, yes, the active poke rates are greater than the inactive, the ≤ 1 poke per min is not very convincing towards deeming this population of neurons as supporting self-stimulation. The authors at the very least need to discuss this and attempt to reconcile. But even with a strong justification for why this rate is so low yet still indicates reinforcement or drives reward, I doubt many in the field will consider this strong evidence for their overall claim about optoICSS.

We performed several additional experiments to address these comments/concerns:

1. Because long and synchronous stimulation of CaMKII+ neuronal population in the dHPC has been shown to trigger the development of epileptic seizures, we first applied restrictive parameters, including a 10sec time out after each dHPC self-stimulation, and a maximum of 100 stimulations per session after which the overall session will end. To answer the first part of the reviewer's comment we removed those restrictive parameters (Figure 1 K-L). In those graphs we can see that removing the 10sec time out period increases the number of stimulation with a session (yellow/green bar and single plots), and removing both time outs and 100 maximum self-stimulation per session, further increase the number of active interactions with the nose poke inset to levels reported in other publications assessing reinforcement and using optogenetics as an approach to interrogate a selective circuits. We also depicted in Figure 1H and 1L, the continuous distribution of self-stimulation across a session confirming that the animals do seek for dHPC stimulation across the session.

2. To address the comment regarding the saliency of the change, we trained animals to self-stimulate and removed the cue-light indicating the reward availability (please see Suppl Fig 1). In this case, and in opposition to when we shut off the laser and leave the cue light on (Please see Fig 1-I), removing the cue-light does not impact the rate of self-stimulation, further confirming that dHPC→NAcSh is sufficient to drive reinforcing behavior, independently from the salience of the cue light, or "saliency of change".

Reviewer Comment:

Thank you for the additional optoICSS experiments. It is clearer now that the stimulation is reinforcing the active lever press.

Similarly, the effects in RTPT are very modest for a "rewarding" manipulation. There seems to be lag in the effect, in that the first 10 minutes of the task, the Chr2 group shows no strong place preference, but it gets more apparent after the 10min mark, and then stronger and stronger as the session goes on. Typically, the affects seen from rewarding brain stimulation are rapidly acquired in this paradigm. It does not take rodents long to form the association of a "room", or even spot in a chamber, with the rewarding stimulation. The significant differences

in Figure 1F appear to be driven by 6 or so mice, otherwise the groups appear similar (are there histological observations for why out of 18 mice, only a handful have % time in stim side greater than control?). The control group (Fig 1 H) show similar trends (with almost half as many mice)

We appreciate the reviewer's comment. We would like to point out the fact that all animals that underwent the real time place preference procedure (Figure 1A-E) exhibited a preference for the light stimulation side, suggesting a rewarding effect of the stimulation. We acknowledge that the effects shown in the figure described by the reviewer (now removed to avoid confusion) may have appeared slightly more nuanced. In this procedure the chamber associated with light stimulation was changed in the middle of the 40 minutes session, forcing the animals to adjust their strategy to receive dHPC stimulation. While most of the animals expressing channelrhodopsin exhibited this adjustment, both the length of the procedure (40min vs 30min for classic RTPT) and the adjustment to the change in light stimulated chamber decrease their overall preferences when compared to a more classic RTPT procedure (see Figure 1A-E in which 100% of ChR2 animals spent above 50% of their time in the light associated chambers). Lastly, we would like to point out that in addition to RTPT, the results shown throughout the manuscript using an operant procedure reinforce the conclusion drawn by the RTPT results.

Reviewer Comment:

Thank you for the additional RTPT experiment and clarification. I agree that this supports the optoICSS. Since the authors did not do a true "preference" test, in the sense that they paired on side with stimulation and then test again later without the stimulation to see where they preferred to be, I would suggest slightly revising this text (lines 100-102 or so) to remove terms like showed "strong" preference (also in general these statements are speculative, what is "strong" preference?), to just spent more time on the light paired side, which is really all you can assess from this assay. One other suggestion, I think many in the field see ICSS as the gold standard for the assessments the authors are conducting, and since the RTPT here mainly is just supportive of the optoICSS (i.e. they have not presented much more analysis of this test beyond this), I think the flow of the findings may work better leading with the ICSS and showing the RTPT as support. But, this is up to them.

We thank the reviewer for this valuable feedback. We have revised the text to remove the term "preference" to avoid any confusion on how the RTPT experiment was being conducted and what it aimed to test (Lines 99-101). Since the RTPT box consists of two identical matte black compartments, we could not do an actual preference test as mentioned by the reviewer. The RTPT experiment aimed to assess the behavioral outcome of stimulating the dHPC, without any association to external cues or context. The observed significant time spent in the stimulation-paired compartment suggests that stimulation of dHPC is sufficient to promote positive reinforcement. This is complemented by the operant self-stimulation procedure as we wanted to assess if the stimulation of dHPC is still sufficient to maintain reinforcement when the mice have to "work" (nose poke) to receive 2 seconds of stimulation.

With the above point on optoICSS rates, it is hard to reconcile that this manipulation is rewarding – the effects are just so much weaker at these two common tasks (ICSS and RTPT) than seen with similar manipulations of validated/established "reward" systems.

Please see comments above.

Reviewer Comment:

I think the data on optogenetic reinforcement is clearer now.

4) There are important details related to the stimulation parameters that are missing. What intensity was the light for operant tasks? How or why did they decide on the parameters they used? Were there any other "doses" of light they tried, i.e. different train lengths, frequency, etc.? Do dCA1 neurons ever respond in this fashion (i.e. 20hz for 2 seconds) during natural behaviors?

Those details have been added in the Methods section of the manuscript. We based our stimulation procedures on multiple publications looking at neurocircuits of operant reward and aversion using optogenetics, and more particularly in the NAcSh and the dHPC (PMIDS: 34646022, 34385700, 26335648, 28496380, 34160168).

Reviewer Comment:

Make sure because I still don't see details like laser power for the optoICSS. These info are important for reproducibility or folks who want to follow up on these findings.

We apologize again for missing that in our previous manuscript version. This has been added together with the other parameters in the Method sections (20 Hz, 15 ms pulses for 2 s, 4 mW; Lines 408-409).

5) Infusion of AMPA/NMDA antagonists into the accumbens essentially takes off-line this region that is crucial for motivated behavior (MSNs don't do much without glu input). The authors must show that the reduction in dCA1 ICSS is specific to this manipulation and the manipulation does not, rather, simply disrupt all reward seeking behaviors (or locomotion). It is also unclear how this experiment was conducted. The methods say this was a bilateral cannula implant. Why? The optogenetic manipulations are unilateral, so why infuse into each NAc rather than just ipsilateral to the opto-stim. If indeed, both NAc receive glutamate antagonists, I believe many/most motivated behaviors would be affected, which would undermine what they are trying to show – that the glu release in the NAc from the opto stim is implicated in the reinforcing effects of that stimulation. I recommend designing an experiment that shows this microinfusion specifically disrupts your optoICSS, and not general operant responding.

6) The DREADD experiments suffer from the same design issue as cannula infusions above.

We would like to thank the reviewer for this comment. While sparse, we observed faint labelling in contralateral NAc when infusing Chr2 in dHPC. Thus, we felt the need to provide treatment in both NAc hemispheres to account for this. This has been justified in the methodology section of the manuscript : “Treatments were given bilaterally to ensure that any Chr2-induced recruitment of Enk+ or Dyn+ neurons would be silenced.” and “Treatments were given bilaterally to ensure that any Chr2-induced dHPC→NAc glutamate signaling would be blocked”

Reviewer Comment:

This is not really addressing the issue with this experiment. As I said before, glu antagonist bilaterally infused into the NAc will disrupt most, if not all, motivated behaviors. So there is no basis to claim that it is specifically attenuating the ICSS here. I do not think the authors can say much from how the experiment was conducted. I do not think it adds much anyway, given the later DREADD experiments and slice physiology. I think the authors should walk back their interpretation of this experiment or remove it. An appropriate claim could be that Glu transmission in the NAc is needed for operant responding for dHPC->NAc optogenetic stimulation, but this again doesn't add anything novel.

We appreciate the feedback provided by the reviewer. We acknowledge that many motivated behaviors are likely to be disrupted with bilateral infusion of glutamate antagonists into the NAc. However, it is important to note that this method is the only available approach in our lab to assess that activation of dHPC CaMKII⁺ neurons induce NAc activation via glutamatergic transmission, subsequently leading to the observed reinforcing behavior. While not entirely novel, this experiment represents an incremental step we have taken in assessing the role of the dHPC^{CaMKII⁺}→NAc pathway in reinforcing behaviors, preceding the DREADD and slice physiology experiments. In response to this and Reviewer #2 feedback, we have adjusted the wording in our conclusion within our revised manuscript to better reflect our findings (Lines 197-198 and 207-208).

7) They have not demonstrated that the optogenetics or chemogenetics do what they are supposed to do. Some sort of validation of these methods is necessary. Do the dCA1 neurons follow a 2 second 20hz train? Do the Gi DREADDs take Dyn and Enk “offline” equally?

We greatly appreciate the concern brought by the reviewer. We characterized both our chemogenetic and optogenetic approaches in previous reports and, as mentioned above, these constructs have been used in prior publications to silence both dyn- and enk-neurons within the NAc. Therefore, we followed the procedures that were described in these other reports. The appropriate references have been added in the methods section of manuscript to support this.

Reviewer Comment:

Thanks for the clarification.

8) The discussion lacks any synthesis of how this pathway fits into canonical reward systems. How these neurons work in this neural module to affect motivation would help us understand why stimulation of these neurons modestly reinforces behavior.

We expanded the discussion to further integrate this pathway into the current literature on reward circuitry. “While the number of Dyn+ MSNs responding to dHPCCaMKII+ terminal stimulation in the NAcSh remains low (~12.5%), it is plausible that constant stimulation of dHPCCaMKII+ input to NAcSh Dyn+ MSNs, either through experience or training, would strengthen the pathway, making it key to modulate positive reinforcing behaviors (Figure 8). For example, constant exposure to drugs such as cocaine enhances excitatory synaptic connections into NAc MSNs 67. This is also evident in our fiber photometry recordings where calcium transient recorded in NAcSh of mice with prior training to self-stimulation operant task (Figure 2K-M) were higher than those who have no prior exposure to dHPCCaMKII+ neuron stimulation (Suppl Figure 2F-H).

Furthermore, the discrepancy between the selective role of Dyn-MSNs in dHPCCaMKII+-driven reinforcing behavior and the small response in slice electrophysiology experiments could be due to the loss of local circuitry within the NAc during slice preparation. This could lead to the underrepresentation or absence of certain microcircuits in the NAcSh that are critical for the activation of Dyn-MSNs in dHPCCaMKII+-driven reinforcing behavior. The recent report of dHPCCaMKII+ neurons recruiting FSIs, which in turn drives the activity of the MSNs assembly in the NAc 23, suggests the potential recruitment of the local circuitry in the NAcSh to regulate Dyn+ neurons during dHPCCaMKII+ driven reinforcing behavior. Therefore, further investigation is also necessary to fully determine the contribution of local circuitry in the NAcSh in dHPCCaMKII+-driven reinforcing behavior. Additionally, there is high heterogeneity in the anatomy and function of the NAc along the rostral/caudal axis 46, indicating the need for further investigation to fully evaluate the role of NAcSh MSNs in driving reinforcement, rewarding and aversive behaviors.”

Reviewer Comment:

These are useful discussion points. The newly added last sentence should be revised. “potential” used twice and it is an awkward way to end the manuscript.

We thank the reviewer for this feedback. This last sentence has been revised in the current version of the manuscript (Lines 345-346).

9) The authors clearly have the capability to investigate how the neurons they study here, in particular the dCA1->NAc, contribute to natural reward seeking behaviors. They give some examples in the introduction and discussion that suggest they do, but we already know dCA1 and dCA1->NAc play a role in memory processes related to reward seeking behaviors. As stated before, the ICSS here is artificial and simply suggests somehow these neurons are related to reward seeking (again, we know this), but complimentary experiments to monitor and/or disrupt the normal activity of these neurons during ethologically relevant goal-directed behavior is needed to make this story more than an incremental advancement.

We thank the reviewer for this insightful comment. In the revised manuscript, we used sucrose self-administration combined with an inhibitory chemogenetic approach to silence the activity of dHPC->NAcSh pathway during sucrose seeking (see Figure 4). With this experiment we demonstrate that silencing dHPC->NAcSh significantly decreased self-administration of sucrose pellets seeking under fixed ratio 1 schedule of reinforcement. This result

further strengthens the importance of the dHPC→NAcSh in mediating reinforcement and reward seeking in addition to its already defined role in contextual/cue memory.

Reviewer Comment:

Yes this is a good experiment thanks.

10) Related to the functional connectivity assays. There are no analysis of the latencies regarding dCA1 stimulation onset and NAc LFP or GCaMP dynamics. This is crucial to interpret the extent that this stimulation acts to modulate NAc directly or indirectly. I am also confused by the summary analysis of the LFP. It appears the authors are using absolute value of the LFP before and after stimulation. Figure 2E seems to show many channels recording a decrease in LFP, and seemingly less show increase, while many show no change. Yet, in line 140 they say evoked “activation”? I cannot seem to find what Pre-Min and Post-Max are referring to in Figure 2F; what is pre and what is post, and compare a minimum and maximum from these different times? They also seem to have recorded NAc LFPs both ipsilateral and contralateral to the dCA1 stimulation, but this is not mentioned or discussed.

It is unclear how fiber photometry of GCaMP “gains both anatomical and temporal resolution of the dynamics of NAc neuronal activity...” (Line 143) when microelectrode recordings have better resolution of both. It also would seem that the results related to this in the main text (FP during dCA1 optoICSS) are confounded by the coincidence of the behavioral task when interpreting whether the opto-stimulation is affecting the FP signal (as opposed to the behavior) – the suppl. data Fig1E-H better support this interpretation.

We thank the reviewer for this comment. We included additional information in the text and also additional analyses in Figure 2 to help understand the LFP data (Figure 2E-H). Briefly, we included in the text the following information regarding the LFP recordings. Pre- and post-stim amplitude were calculated by subtracting the minimum amplitude from the maximum amplitude within a 0.1s window pre- and post- dHPC stimulation (Figure 2F). Stimulation of dHPCCaMKII+ neurons evoked a significant increase in overall evoked LFP amplitude recorded in the NAc (Figure 2G). By calculating the time between dHPC stimulation onset and the first evoked peak (Latency to peak; Figure 2F), we observed a wide variability in latency to peak (Figure 2H), suggesting the presence of monosynaptic inputs (latency < 0.01s) and polysynaptic inputs (latency > 0.01 s) from the dHPC. The well-defined role of the NAcSh in processing reward and positive reinforcement, together with our finding of denser dHPCCaMKII+ projections into the NAcSh (Figure 2C), suggest the possible involvement of this pathway to mediate dHPCCaMKII+-driven reinforcing behavior.

Reviewer Comment:

Thank you for adding this.

Regarding the reviewer’s comment about the confound between behavioral task and light stimulation, those concerns are addressed by the additional experiments now shown in supplementary Figure 2 E-H in which the animals receive a non-contingent light self-stimulation in an open field arena. The response observed in this scenario is tightly coupled to the stimulation of dHPC CaMKII+ neurons administered by the experimenter. These additional data demonstrate that the activation of those neurons is sufficient to trigger increase in calcium transient within the NAcSh, regardless of the behavioral task.

Reviewer Comment:

Good experiment to address the concern here.

11) Line 153 – how is this a selective accumbal response? They’ve not compared signals in surrounding regions. This brings up another important missing control – there are no site controls in this study. They’ve not investigated other dCA1 projections, nor placed optic fibers in areas adjacent to injection sites to identify spatial specificity. One of the main claims is that while we know electrical excitation of dHPC is reinforcing, they argue

this study identifies dCA1 CAMKII, and in particular those dCA1 CAMKII projecting to NAc as mediating that reward – but these neurons project elsewhere and they’ve not shown these other pathways don’t support ICSS.

We thank the reviewer for this comment. To answer this comment, and as suggested by Reviewer #2, instead of widely stimulating adjacent areas we employed the suggested dual virus strategy with Cre-recombinase-encoding retroAAV virus in the NAcSh in conjunction with Cre-dependent AAV vectors in the dHPC. This allowed us to selectively engage direct projections from the dHPC to the NAcSh during light self-stimulation procedures. We show here that, similarly to the stimulation of the overall CaMKII+ neuronal population in the dHPC, activation of this selective dHPC→NAcSh pathway is sufficient to drive reinforcing behavior (please see Figure 3 A-D).

Reviewer Comment:

The AAV-retro does add more specificity here. Thanks.

12) I am not sure what the brain clearing studies are adding here. Firstly, both tracers jRCECO1a and Chr2 are not specific to terminals, so analysis is limited to regions where fibers are present, could be fibers of passage, no specificity to functional connectivity. But, they are not really doing anything with this data set regardless. In fact, this seems like a missed opportunity to identify some projection “site-controls” to test the above point. Likewise, using similar strategies they could identify whether dCA1→NAc projecting neurons send collaterals elsewhere. Optogenetic stimulation of terminals can send antidromic action potentials back to the cell body, and if the neuron sends collaterals, these alternate projections could mediate reinforcing properties. This later concern would be alleviated if the cannula experiments were performed appropriately.

We thank the reviewer and agree that those brain clearing analysis were not providing critical data and therefore they were removed from the current version of the manuscript. Instead, we included additional comprehensive analysis of the dHPC-NAcSh input using viral tracing approaches and included updated illustrations of the projections.

Reviewer Comment:

Good idea.

Other suggestions:

Line 258 and Methods heading line 389– “...light self-stimulation”. I do not think this is the correct phrase. I know what they mean, but others not familiar with these paradigms may be confused and interpret this as light – i.e. modest, ‘lite’, weak, etc. I think optogenetic self-stimulation or just self-stimulation. If anything, that phrase should read ‘light self-administration’.

We thank you for this feedback. We have corrected the phrasing to optogenetic self-stimulation throughout the revised manuscript including the Methods heading in line 405.

In summary, we would like to respectfully request for our manuscript to be reconsidered given the additional studies that we conducted and the revisions to the text that fully address all the reviewers’ concerns.

We would like to thank you for your time and consideration.

Regards,

*Dr. Jose A. Moron
Dr. Khairunisa Ibrahim
Dr. Nicolas Massaly*

REVIEWER COMMENTS

Reviewer #1 (Remarks to the Author):

The authors have generally done a pretty good job of replying to my various comments. However, two serious points remain.

[1] The authors have added still more data and, although these continue to help make the case for their interpretation of the data, they seem not to have taken the important step of actually comparing predictions from a stimulus driven explanation for their responding (the mouse is responding because stimuli in the environment predict stimulation) and a reinforcement driven explanation: the response is performed because it has been strengthened by reinforcing stimulation. I pointed out how to test this explicit claim in my last comments - i.e., using an omission schedule to assess whether the behavioral response would be acquired or performed if it omitted the stimulation (see Holland, 1978: <https://doi.org/10.1037/0097-7403.5.2.178>) – but among the studies added that is not one of them. As a consequence, the authors make a good case for their interpretation, but it remains merely one interpretation of the data.

Notice that merely removing the explicit stimuli doesn't remove all potential stimuli. Nor does not having explicit contextual cues mean there is no context: that would be an impossibility. It is precisely because the potential for Pavlovian conditioning in operant tasks is so strong that alternative tests to assess the impact of those cues is required in order to understand what has been found.

With respect to how this point should be handled in the manuscript, I think it is important that the potential alternative account, i.e., a purely Pavlovian S-O account of the current results, is raised as a clear alternative explanation of the data in the discussion. It cannot be finally dismissed and so needs to be put forward even if the authors favor the alternative.

[2] Now that the authors include additional images showing the pathway, there is one other anomaly that wasn't apparent before. It is very clear from the images in Figures 2 and S2 that the projection from hippocampus is into the LATERAL shell, not the medial shell. However, as is clear from their various schematics summarising their histology in figures S6-8, all of their placements measuring activity and delivering stimulation are in the MEDIAL shell. Its not clear to me what we should take this to mean except that the results may well be quite different were the authors to target the denser region of the HPC-shell projection. How can we know what to make of this anomaly without some kind of study comparing medial vs. lateral shell? There may well be a medial shell part to this projection but, if there is, it isn't made very clear in the current images. Nor is there any obvious justification for the placements being made in the medial rather than the lateral shell provided in text. As things stand the current images seem to completely undermine the authors' choice of target. Some more work clearly needs to be done to tighten up the logic here and, indeed, it would seem to me that the authors might themselves want to be sure that their findings replicate in lateral shell.

Reviewer #2 (Remarks to the Author):

The authors have adequately addressed all of my remaining concerns. The slice physiology is much clearer and more robust now. I thanks the authors for their thoughtful responses and revisions to the paper.

Reviewer #3 (Remarks to the Author):

Thank you to the authors for clear responses during this peer review process. I have nothing more to add as a reviewer of this manuscript.

We thank the Reviewers and the Editor for their constructive comments. We have revised the manuscript, specifically the discussion, to address each of the comments made by Reviewer 1 (reviewed point-by-point below). We do hope that you now agree that our manuscript has fully satisfies all the reviewers' concerns and is suitable for publication in Nature Communications.

All the modifications made to the manuscript have been highlighted in blue throughout the text. See below for our point-by-point response to each of the comments raised by Reviewer 1:

Reviewer #1

The authors have generally done a pretty good job of replying to my various comments. However, two serious points remain.

[1] The authors have added still more data and, although these continue to help make the case for their interpretation of the data, they seem not to have taken the important step of actually comparing predictions from a stimulus driven explanation for their responding (the mouse is responding because stimuli in the environment predict stimulation) and a reinforcement driven explanation: the response is performed because it has been strengthened by reinforcing stimulation. I pointed out how to test this explicit claim in my last comments - i.e., using an omission schedule to assess whether the behavioral response would be acquired or performed if it omitted the stimulation (see Holland, 1978: <https://doi.org/10.1037/0097-7403.5.2.178>) – but among the studies added that is not one of them. As a consequence, the authors make a good case for their interpretation, but it remains merely one interpretation of the data.

Notice that merely removing the explicit stimuli doesn't remove all potential stimuli. Nor does not having explicit contextual cues mean there is no context: that would be an impossibility. It is precisely because the potential for Pavlovian conditioning in operant tasks is so strong that alternative tests to assess the impact of those cues is required in order to understand what has been found.

With respect to how this point should be handled in the manuscript, I think it is important that the potential alternative account, i.e., a purely Pavlovian S-O account of the current results, is raised as a clear alternative explanation of the data in the discussion. It cannot be finally dismissed and so needs to be put forward even if the authors favor the alternative.

We appreciate the feedback provided by the reviewer. In response, we have made adjustments to our discussion to acknowledge that our findings do not negate the possibility that the stimulation of dHPC^{CaMKII+} could potentially enhance the saliency of mice nose-poking behavior. We have explicitly stated that further studies are necessary to fully distinguish and comprehend the role of dHPC^{CaMKII+} stimulation in saliency-driven responses as opposed to reinforcement-based behavioral responses in the operant self-stimulation paradigm (refer to Lines 353-356).

[2] Now that the authors include additional images showing the pathway, there is one other anomaly that wasn't apparent before. It is very clear from the images in Figures 2 and S2 that the projection from hippocampus is into the LATERAL shell, not the medial shell. However, as is clear from their various schematics summarising their histology in figures S6-8, all of their placements measuring activity and delivering stimulation are in the MEDIAL shell. Its not clear to me what we should take this to mean except that

the results may well be quite different were the authors to target the denser region of the HPC-shell projection. How can we know what to make of this anomaly without some kind of study comparing medial vs. lateral shell? There may well be a medial shell part to this projection but, if there is, it isn't made very clear in the current images. Nor is there any obvious justification for the placements being made in the medial rather than the lateral shell provided in text. As things stand the current images seem to completely undermine the authors' choice of target. Some more work clearly needs to be done to tighten up the logic here and, indeed, it would seem to me that the authors might themselves want to be sure that their findings replicate in lateral shell.

We appreciate the reviewer's feedback. In our results section, we have duly recognized the existence of two distinct areas of projections, namely the dorsal-medial NAcSh and the ventro-lateral NAcSh (see Lines 163-164). Projections in the dorsal-medial NAcSh are visibly illustrated in Figure 2C, 2D, and 2E. The primary emphasis of our current study is on the dorsal-medial NAcSh, as supported by previous publications highlighting its significant role in reward and reinforcement (Al-Hasani et al., 2015; Castro & Bruchas, 2019; Massaly et al., 2019).

In response to the reviewer's suggestions, we have made clarifications in the revised manuscript, explicitly stating the primary focus on the dorsal-medial NAcSh (dmNAcSh) in Lines 164-166. Furthermore, we have adjusted the terminology throughout the manuscript, including figures, to consistently refer to dmNAcSh, providing clarity on our region of interest.

While we acknowledge the reviewer's interest in exploring the role of dHPC^{CaMKII+} projection to the ventral NAcSh in reinforcing behavior, we would like to note that this falls outside the scope of our current study. However, we have recognized the importance of such investigations and have highlighted the need for future research specifically addressing the dHPC^{CaMKII+} → ventral lateral subregion of the NAcSh projections (Lines 357-359).

Reviewer #2

The authors have adequately addressed all of my remaining concerns. The slice physiology is much clearer and more robust now. I thank the authors for their thoughtful responses and revisions to the paper.

Reviewer #3

Thank you to the authors for clear responses during this peer review process. I have nothing more to add as a reviewer of this manuscript.

REVIEWERS' COMMENTS

Reviewer #1 (Remarks to the Author):

The authors have now adequately ddressed my remaining comments.